# Brain expression quantitative trait locus and network analyses reveal downstream effects and putative drivers for brain-related diseases

Niek de Klein [1,2,10], Ellen A. Tsai [3,10], Martijn Vochteloo [1,4,5,10], Denis Baird[3], Yunfeng Huang[3], Chia-Yen Chen[3], Sipko van Dam[1,6], Roy Oelen[1,5], Patrick Deelen [1,5], Olivier B. Bakker [1,2], Omar El Garwany [1,2], Zhengyu Ouyang[7], Eric E. Marshall[3], Maria I. Zavodszky[3], Wouter van Rheenen [8], Mark K. Bakker [8], Jan Veldink [8], Tom R. Gaunt [9], Heiko Runz [3,11] ✉, Lude Franke [1,5,11] ✉ & Harm-Jan Westra [1,5,11] ✉

Identification of therapeutic targets from genome-wide association studies (GWAS) requires insights into downstream functional consequences. We harmonized 8,613 RNA-sequencing samples from 14 brain datasets to create the MetaBrain resource and performed *cis*- and *trans*-expression quantitative trait locus (eQTL) meta-analyses in multiple brain region- and ancestry-specific datasets ($n \le 2,759$). Many of the 16,169 cortex *cis*-eQTLs were tissue-dependent when compared with blood *cis*-eQTLs. We inferred brain cell types for 3,549 *cis*-eQTLs by interaction analysis. We prioritized 186 *cis*-eQTLs for 31 brain-related traits using Mendelian randomization and co-localization including 40 *cis*-eQTLs with an inferred cell type, such as a neuron-specific *cis*-eQTL (*CYP24A1*) for multiple sclerosis. We further describe 737 *trans*-eQTLs for 526 unique variants and 108 unique genes. We used brain-specific gene-co-regulation networks to link GWAS loci and prioritize additional genes for five central nervous system diseases. This study represents a valuable resource for post-GWAS research on central nervous system diseases.

Psychiatric and neurological diseases continue to be a massive global health burden: The World Health Organization estimated that in 2019, globally 280 million individuals were affected by depression, 39.5 million by bipolar disorder and 287.4 million by schizophrenia (SCZ)[1]. Similarly, the number of people living with dementia is expected to rise from 50 million today to 152 million by 2050 (ref. [2]), with similar trajectories for other neurodegenerative diseases. Although substantial progress has been made in uncovering the genetic basis

[1]Department of Genetics, University Medical Center Groningen, University of Groningen, Groningen, The Netherlands. [2]Wellcome Sanger Institute, Hinxton, UK. [3]Translational Biology, Research and Development, Biogen Inc., Cambridge, MA, USA. [4]Institute for Life Science and Technology, Hanze University of Applied Sciences, Groningen, The Netherlands. [5]Oncode Institute, Groningen, The Netherlands. [6]Ancora Health, Groningen, The Netherlands. [7]BioInfoRx, Inc., Madison, WI, USA. [8]Department of Neurology, UMC Utrecht Brain Center, University Medical Center Utrecht, Utrecht University, Utrecht, The Netherlands. [9]MRC Integrative Epidemiology Unit, Bristol Medical School, University of Bristol, Bristol, UK. [10]These authors contributed equally: Niek de Klein, Ellen A. Tsai, Martijn Vochteloo. [11]These authors jointly supervised this work: Heiko Runz, Lude Franke, Harm-Jan Westra. ✉e-mail: heiko.runz@gmail.com; l.h.franke@umcg.nl; h.j.westra@umcg.nl

**Fig. 1 | Overview of the study.** We downloaded publicly available RNA-seq and genotype data from 14 different datasets consisting of 8,613 RNA-seq measurements from seven main brain regions and 6,518 genotype samples. We created six eQTL meta-analysis datasets and performed *cis*-, *trans*- and interaction-eQTL analyses, built a brain-specific gene co-regulation network and prioritized genes using MR, co-localization and the co-regulation network. Image of sagittal cut of brain created with BioRender.com. This figure summarizes values from Supplementary Tables 1, 3, 8, 12 and 25–30.

of these diseases through genome-wide association studies (GWAS), much of how the identified genetic variants impact brain function remains unknown.

To translate from genetic signals to mechanisms, associations with gene expression levels or expression quantitative trait loci (eQTL) have shown great potential. *Cis*-eQTLs (nearby) and *trans*-eQTLs (distal) can aid the interpretation of GWAS loci in several ways. *Cis*-eQTLs provide direct links between genes and phenotypes through causal inference approaches such as Mendelian randomization (MR) and genetic co-localization analyses, whereas *trans*-eQTLs expose sets of downstream genes and pathways on which the effects of disease variants converge.

Expression quantitative trait loci are dynamic features and vary with tissue, cell type and additional factors such as response to stimulation. Therefore, eQTLs from disease-relevant tissues are desired for optimal interrogation of GWAS loci[3]. Previous brain eQTL meta-analyses by the PsychENCODE[4] (*n* = 1,866) and AMP-AD[5] (*n* = 1,433) consortia have been published to help interpret neurodegenerative and psychiatric disease GWAS loci. However, results from statistical approaches such as MR and co-localization are improved by robust effect-size estimates from even larger carefully curated eQTL datasets. In addition, large sample sizes are better suited to decompose eQTL effects to specific cell types.

To maximize the potential of eQTL-based analyses of the brain, we combined and rigorously harmonized brain RNA-sequencing (RNA-seq) and genotype data from 14 different cohorts, including 8,613 RNA-seq samples from all major brain eQTL studies, and publicly available samples from the European Nucleotide Archive (ENA). We created a gene co-regulation network based on 8,544 RNA-seq samples covering different brain regions and performed *cis*- and *trans*-eQTL analyses of up to 2,683 individuals of European ancestry (EUR), with replication in up to 319 individuals of African ancestry (AFR). We made inferences on the brain cell types in which eQTLs operate and systematically conducted MR and co-localization analyses to find shared genetic effects between eQTLs and 31 brain-related GWAS

traits. Our analyses prioritize probable causal genes and reveal cell type-dependent eQTLs that may be associated with disease risk (Fig. 1).

To facilitate future studies, we have made all summary statistics and the co-expression network derived from our resource available at www.metabrain.nl.

## Results

### Harmonizing datasets for eQTL and co-regulation analysis

We combined 14 eQTL datasets into the 'MetaBrain' resource to maximize statistical power to detect eQTLs and create a brain-specific gene co-regulation network (Fig. 2, Supplementary Figs. 1–7 and Supplementary Table 1). Previous to quality control (QC), MetaBrain includes 7,604 RNA-seq samples and accompanying genotypes from the AMP-AD consortium (AMP-AD MAYO, ROSMAP and MSBB)[6], Braineac[7], the PsychENCODE consortium[8] (Bipseq[4], BrainGVEX[4], CMC[9], CMC_HBCC and UCLA_ASD[4]), BrainSeq[10], NABEC[11], TargetALS[12] and GTEx[3]. In addition, we carefully selected 1,759 brain RNA-seq samples from the ENA[13], which we subsequently genotyped and imputed (Fig. 2a, Supplementary Note and Supplementary Figs. 1–3). After realignment, removal of duplicate samples and stringent QC, 8,613 RNA-seq samples remained (Methods and Supplementary Figs. 4,5). Using slightly different QC thresholds, we created a gene network using 8,544 samples (Supplementary Note). For both datasets, we corrected the RNA-seq data for technical covariates and defined seven major tissue groups (amygdala, basal ganglia, cerebellum, cortex, hippocampus, hypothalamus and spinal cord): principal component analysis (PCA) on the RNA-seq data showed clear clustering by these major tissue groups, resembling brain physiology (Fig. 2b and Supplementary Fig. 6). The genotype data revealed individuals from different ancestries (Fig. 2c and Supplementary Fig. 2), including 5,138, 805 and 208 samples from EUR, AFR and East Asian (EAS) ancestries, respectively. After QC and deduplication, we created six *cis*-eQTL discovery datasets: Basal ganglia-EUR (*n* = 208), Cerebellum-EUR (*n* = 492), Cortex-EUR (*n* = 2,683), Cortex-AFR (*n* = 319), Hippocampus-EUR (*n* = 168) and Spinal cord-EUR (*n* = 108; Supplementary Table 1 and Fig. 2d). We used Cortex-EAS

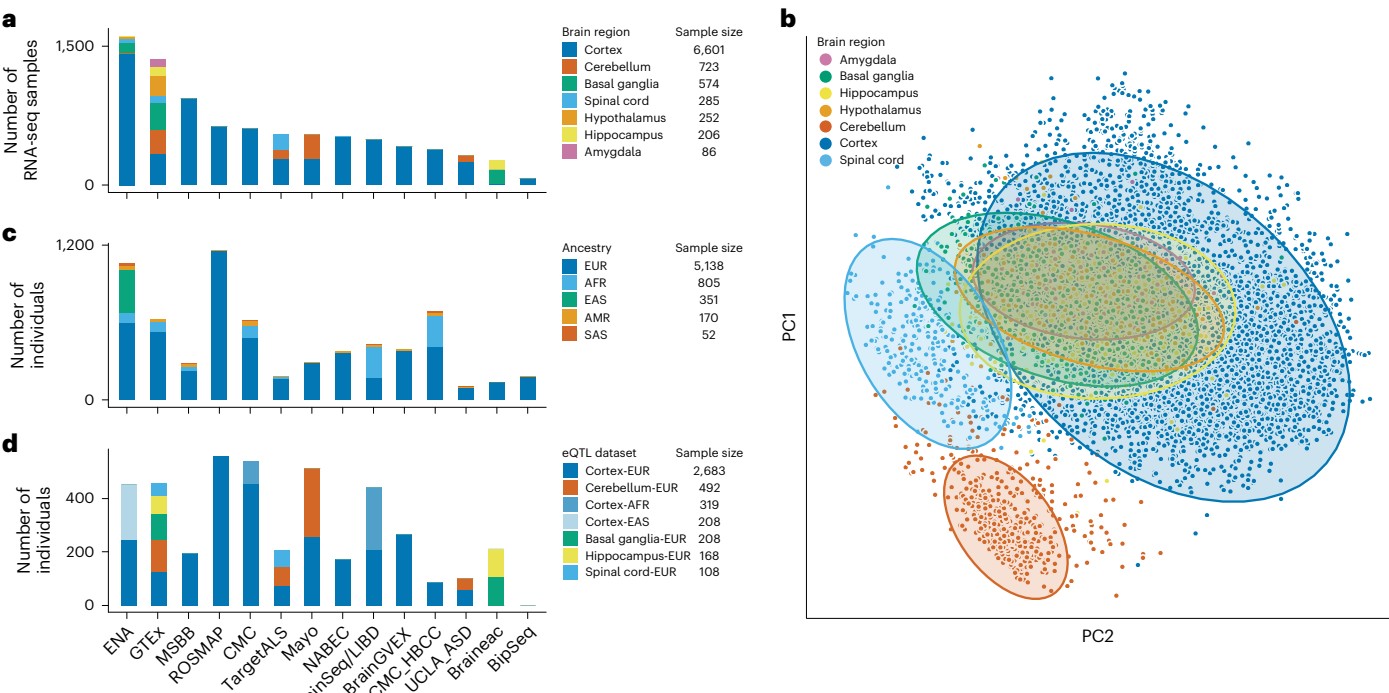

**Fig. 2 | Overview of the datasets. a**, Number of samples per included cohort stratified according to the seven major brain regions. **b**, PCA dimensionality reduction plot of the normalized expression data after covariate correction. Each dot represents an RNA-seq sample and is colored according to the brain region. The figure shows that the samples cluster mainly on the brain region. **c**, Number of genotypes per cohort stratified according to ancestry. AMR, Admixed Americans; SAS, South Asian. **d**, Number of individuals per cohort, with each color representing an eQTL dataset. The number of individuals differ from the intersection between the number of RNA-seq samples and number of genotypes because not all samples with genotypes have RNA-seq samples and vice versa, and some individuals with genotypes have multiple RNA-seq measurements.

($n$ = 208) as a replication dataset. *Cis*-eQTLs were not calculated for the amygdala and hypothalamus tissue groups due to the small sample size ($n$ < 100).

**54% of cortex *cis*-eQTLs have independent associations**

Within each discovery dataset, we performed a sample size-weighted *cis*-eQTL meta-analysis on common variants (minor allele frequency (MAF) > 1%), within 1 megabase (Mb) of the transcription start site (TSS) of a protein-coding gene. We identified 1,880 Basal ganglia-EUR, 10,577 Cerebellum-EUR, 4,797 Cortex-AFR, 16,169 Cortex-EUR, 1,265 Hippocampus-EUR and 998 Spinal cord-EUR *cis*-eQTL genes (*q*-value < 0.05; Fig. 3a and Supplementary Table 2). The observed eQTLs were consistent between datasets (Supplementary Fig. 8) but showed some sensitivity to RNA-seq alignment strategies (Supplementary Note and Supplementary Figs. 9,10). We next performed conditional analysis to identify independent associations in each *cis*-eQTL locus. In Cortex-EUR, 8,815 genes had a significant secondary *cis*-eQTL (54% of *cis*-eQTL genes identified in this dataset), 4,489 genes had tertiary and 2,065 had quaternary *cis*-eQTLs. We also identified secondary associations for the other discovery datasets, albeit to a lesser extent (Fig. 3a and Supplementary Tables 2,3). The properties of the Cortex-EUR *cis*-eQTLs conform to studies performed in blood[14] and brain[15]: primary lead *cis*-eQTL single nucleotide polymorphisms (SNPs) were generally located close to the TSS (median distance, 33.6 kilobases (kb)) and *cis*-eQTL genes had a lower probability for loss-of-function intolerance (pLI > 0.9; $\chi^2 P$ = 2.48 × 10$^{-83}$). Genes with a *cis*-eQTL generally had a higher median expression (Wilcoxon $P$ = 5.5 × 10$^{-174}$; Fig. 3b); the other properties of *cis*-eQTLs were very comparable with earlier reports (Supplementary Note, Supplementary Fig. 11 and Supplementary Table 4).

**High eQTL agreement between ancestries and brain regions**

We investigated ancestry, brain region, dataset and tissue-type differences in *cis*-eQTLs. Agreement between ancestries was high: allelic concordance (AC) and correlation of effect-size ($R_b$) estimates were high when different ancestries were compared ($R_b$ > 0.78, AC > 92.95%; Fig. 3c, Supplementary Fig. 12 and Supplementary Table 5). The proportion of estimated true-positives ($\pi_1$) and correlation of allelic fold change (caFC) estimates between ancestries were lower, potentially due to differences in sample size (for example, Cortex-EUR versus Cortex-EAS, caFC = 0.55 and $\pi_1$ = 0.29; conversely, caFC = 0.85 and $\pi_1$ = 0.95; Supplementary Fig. 12). Similarly, different brain regions showed high overall agreement ($R_b$ > 0.76, caFC > 0.65, and AC > 91%), with $\pi_1$ estimates dependent on the sample size (0.39−0.95). Cerebellum was an exception and showed lower agreement with the cerebral brain regions (Fig. 3d,e and Supplementary Fig. 12). Despite the limited sample size, we identified 477 *cis*-eQTL genes that are significant in Cerebellum-EUR but not in Cortex-EUR (Supplementary Fig. 13), perhaps due to low expression in the cortex or because they are regulated by transcription factors that are more active in the cerebellum (Supplementary Note and Supplementary Table 6). Next, we repeated Cortex-EUR eQTL discovery while excluding GTEx and compared the results with *cis*-eQTLs from different GTEx tissues (Fig. 3e, Supplementary Figs. 12,14 and Supplementary Table 5). There was high agreement between brain-related tissues (cerebral tissues, $R_b$ > 0.8, caFC > 0.71 and AC > 96%; and cerebellar tissues, $R_b$ > 0.76, caFC > 0.71, $\pi_1$ > 0.55 and AC > 92%) compared with other tissue types. The lowest agreement was with tissues such as testis ($R_b$ = 0.51, caFC = 0.48 and AC = 78%) and whole blood ($R_b$ = 0.55, caFC = 0.53 and AC = 80%). The $\pi_1$ scores were not higher for cerebral or cerebellar tissues compared with non-brain tissues. We also compared Cortex-EUR *cis*-eQTLs with eQTLGen[14]

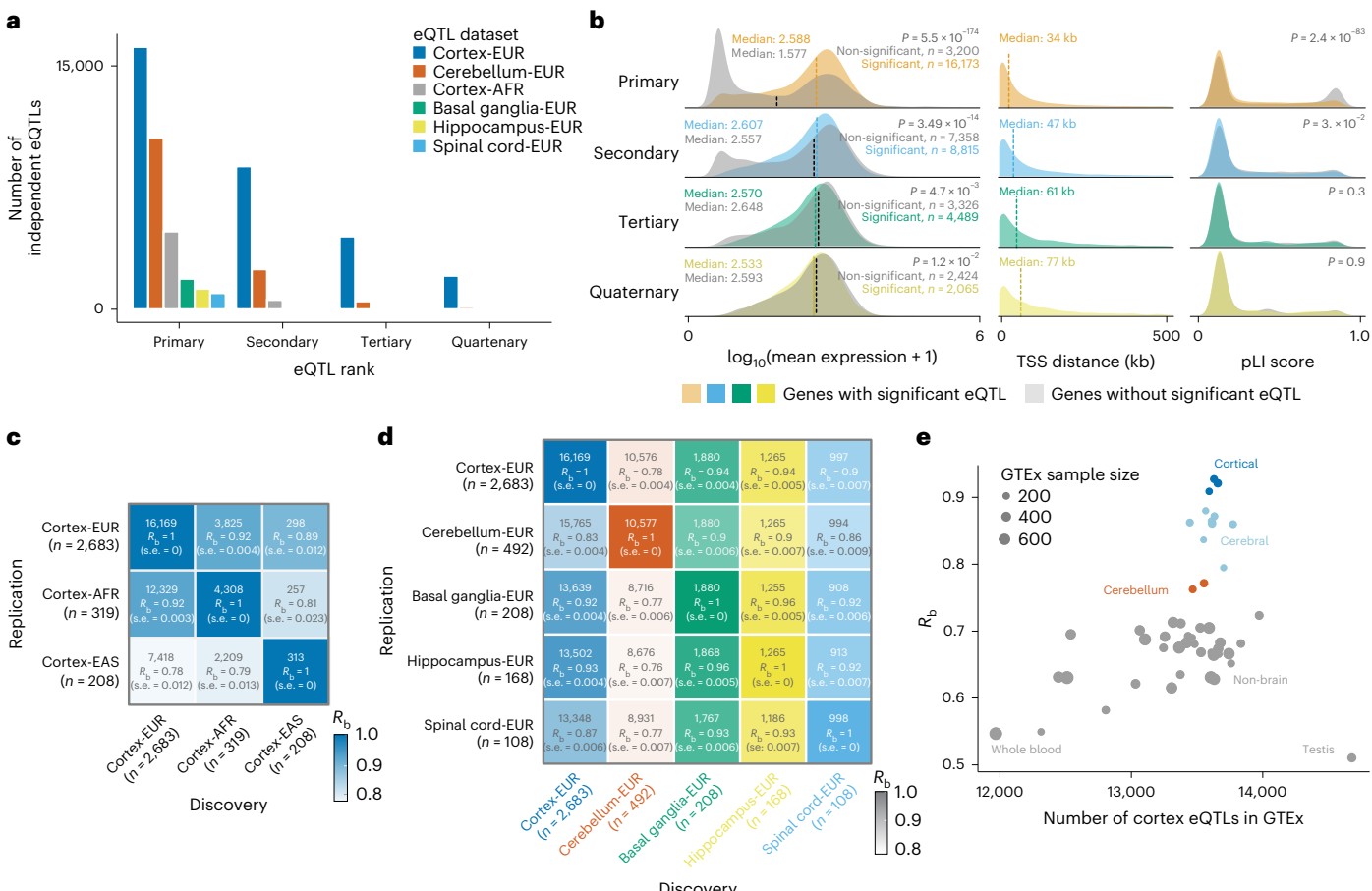

**Fig. 3 | Conditional *cis*-eQTLs. a**, Number of conditional *cis*-eQTLs per eQTL dataset. **b**, Comparison of characteristics between primary and non-primary eQTLs. For the mean expression and pLI score values, each row compares the eQTL genes for that rank with eQTL genes from the previous rank (for example, for tertiary eQTLs, the non-significant (grey) distribution is from genes that have secondary but lack tertiary eQTLs). *P* values were calculated using a Wilcoxon test between significant and non-significant genes. Differences in mean gene expression levels (left), the distance between the most significant SNP–gene combination and the TSS (middle), and pLI scores (right) are shown. For primary, secondary and quaternary eQTLs, non-significant eQTLs have higher pLI scores. Vertical dotted lines indicate median of the distribution for the current rank (coloured) versus the previous rank (black). **c,d**, Number of overlapping eQTLs along with the $R_b$ and standard error (s.e.) values of primary *cis*-eQTLs between the cortex eQTLs of different ancestries (**c**) and the different brain regions for the EUR datasets (**d**). *n*, sample size of the dataset. **e**, Correlation of effect sizes and standard error of primary *cis*-eQTLs of Cortex-EUR (discovery, excluding GTEx) in all of the GTEx tissues (replication). Each dot is a different GTEx tissue; the *x* axis indicates the number of eQTLs that are significant in both discovery and replication.

($n$ = 31,684; blood-based, majority EUR ancestry), which supported the low agreement observed in GTEx blood. Of the overlapping eQTLs, 25% had an opposite allelic effect (AC = 75%, $R_b$ = 0.52 and $\pi_1$ = 0.83; Supplementary Fig. 15 and Supplementary Table 7) (ref. [16]), which represents an increase over GTEx and suggests that many of the eQTLs are tissue-dependent. Combined, these results suggest that additional tissue- or ancestry-specific eQTLs can be identified when sample sizes increase. For instance, opposite effects may happen if two causal variants reside on the same haplotype but are specific for different tissues[17], requiring large sample sizes for disentanglement. By revealing eQTLs with opposite allelic effects, our results highlight the relevance of tissue-dependent eQTL mapping to accurately assess the directionality of eQTLs[17].

### 14% of cortex cis-eQTLs are dependent on the cell-type proportion

We evaluated the extent to which eQTLs are dependent on cell-type proportions by determining cell-type interaction eQTLs (ieQTLs)[3,18,19]. In the Cortex-EUR subset, we predicted cell-type proportions using single-cell RNA-seq-derived signature profiles[4] (Supplementary Note

and Supplementary Fig. 16). The cell-type proportions and reconstruction accuracy of our predictions (87%) were comparable to a previous study that used this reference profile on a subset of the Cortex-EUR samples[4]. We observed low-to-moderate correlations between predicted cell types (0.01 < Pearson's correlation coefficient ($r$) < 0.55; Fig. 4a) and high positive correlations with immunohistochemistry (IHC) counts from the ROSMAP cohort[20] (overall Pearson's $r$ = 0.89 and per cell-type Pearson's $r$ > 0.1; Fig. 4b). However, we note that the exact proportion for each cell type remains uncertain[21,22].

We used Decon-QTL[19] to identify ieQTLs for the 25,497 independent Cortex-EUR *cis*-eQTLs: 3,549 *cis*-eQTLs (13.9%) showed at least one significant ieQTL (4,095 ieQTLs; Benjamini–Hochberg false discovery rate (BH-FDR) < 0.05; Supplementary Table 8). The largest group of interactions were with excitatory, inhibitory and other neurons (1,627; 39.7%), probably because neurons are the most prevalent cell type. The majority of the ieQTLs (3,090; 75.5%) were uniquely mapped to one cell type (Fig. 4c), although we cannot exclude the possibility that these ieQTLs are also present in other cell types.

We replicated these findings in the Cortex-AFR dataset ($n$ = 319) as well as in two independent single-nucleus RNA-seq (snRNA-seq)

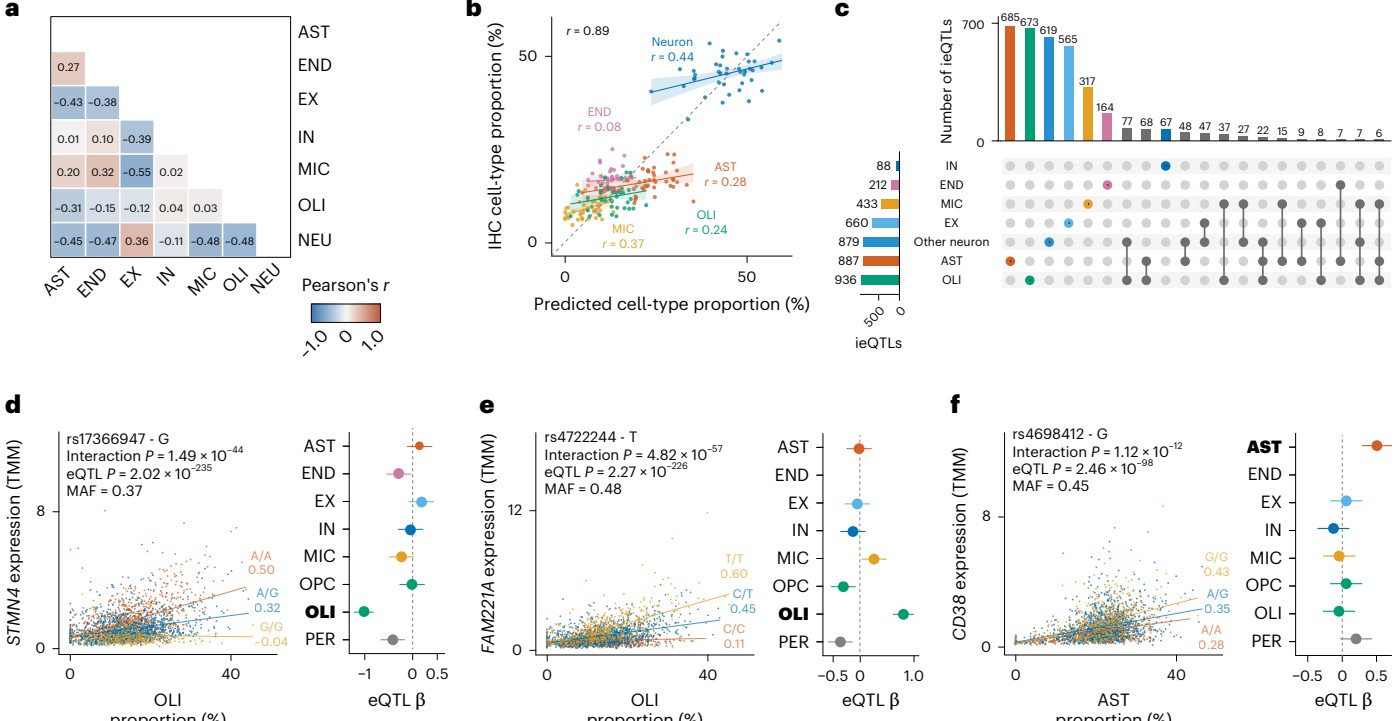

**Fig. 4 | Cell-type ieQTLs. a**, Pearson's correlations between the seven predicted cell-count proportions within cortex samples. **b**, Predicted cell-type proportions compared with cell-type proportions measured using IHC for 42 ROSMAP samples. Pearson's correlation coefficients are provided. The cell-count predictions for most cell types closely approximate actual IHC cell counts. Shaded areas around regression lines indicate 95% confidence interval. **c**, Number of cell-type ieQTLs for Cortex-EUR deconvoluted cell types. The first 20 intersections with the highest overlap are shown. Oligodendrocytes have the most interactions, followed by astrocytes and other neurons. Notably, most interactions are unique for one cell type in 87.1% of the cases. **d–f**, Replication of cell-type ieQTLs for *STMN4* (**d**), *FAM221A* (**e**) and *CD38* (**f**), consisting of the scatterplot of the cell-type ieQTL in MetaBrain Cortex-EUR bulk RNA-seq (left) and a forest plot for the eQTL effect in the ROSMAP snRNA-seq data (right).

Each dot in the scatterplots (left) represents a sample; colors indicate SNP genotype, with yellow being the minor allele; values under the genotypes are the Pearson's correlation coefficients; interaction *P* values were determined using a one-sided *F*-test; eQTL *P* values were derived using the standard normal distribution from meta-analyzed *z*-scores. Forest plot (right): eQTL β values (dots) and standard error (error bars) with effect direction relative to the minor allele when replicating the eQTL effect in ROSMAP single-nucleus data (*n* = 38); each row denotes a cell type-specific dataset; cell types highlighted in bold reflect the equivalent to the cell type used in the ieQTL. Vertical dashed lines indicate an eQTL beta of 0. TMM, trimmed mean of M-values; AST, astrocytes; END, endothelial cells; EX, excitatory neurons; IN, inhibitory neurons; MIC, microglia; OPC, oligodendrocyte precursor cells; OLI, oligodendrocytes; NEU, other neuron; PER, pericytes.

datasets from ROSMAP[23] (*n* = 39) and Bryois et al.[24] (*n* = 196; Supplementary Figs. 17–19 and Supplementary Table 9). Across all replication datasets, we observed moderate-to-high rates of agreement, depending on the cell-type frequency and sample size (Bryois et al.[24]: $0.78 < R_b < 0.86$, median $R_b = 0.84$, $0.43 < π_1 < 0.83$, median $π_1 = 0.69$, $81\% < AC < 90\%$, median AC = 0.9; Supplementary Note). Examples of replicating ieQTLs include the oligodendrocyte ieQTL genes *FAM221A*, *NKAIN1* and *STMN4*, which were previously identified as oligodendrocyte-specific[25], and *AMPD3* and *CD82*, of which the SNPs were previously associated with white-matter microstructure[26], suggesting a role for oligodendrocytes (Fig. 4d,e and Supplementary Fig. 20a–e). The high replication rates indicate that our approach can accurately identify the cell type for a large number of eQTLs. We note that summary statistics were available for only 54% of ieQTLs in a well-powered replication dataset (Bryois et al.[24]), suggesting that our approach had the power to detect ieQTLs that are not yet identified in snRNA-seq datasets.

These ieQTLs can also provide valuable information about the cell types of interest for disease-associated SNPs. For example, the A allele of variant rs4698412, which is associated with increased risk for Parkinson's disease (PD), also increased the expression of *CD38*, for which we identified a replicating astrocyte ieQTL (Fig. 4f and Supplementary Fig. 20f). This gene is an immunomodulatory agent and is mainly expressed in neurons, astrocytes and microglia[27],

and increased levels of *CD38* are observed with neuroinflammation (Supplementary Note).

## Shared genetic effects between *cis*-eQTLs and central nervous system traits

We next linked Cortex-EUR *cis*-eQTLs to variants associated with brain-related traits and diseases. We determined the linkage-disequilibrium (LD) overlap between *cis*-eQTLs and GWAS SNPs, which indicated that primary eQTLs were 2.6-fold more likely to be in LD with a GWAS SNP compared with non-primary eQTLs (Fisher's exact test, $P = 7.4 × 10^{-125}$; Supplementary Note and Supplementary Table 10). To more formally test whether there was evidence for sharing the same genetic effect between *cis*-eQTLs and 31 neurological traits, we conducted MR using the Wald ratio method and co-localization analyses (Supplementary Table 11). Among the 359,763 Wald ratios tested across 11,270 genes, 1,531 Wald ratios for 1,088 genes passed a suggestive *P*-value threshold ($P < 5 × 10^{-5}$; Supplementary Table 12). Of the *cis*-eQTL instruments from these findings, 294 were also cell-type ieQTLs. There were 549 significant Wald ratios that passed Bonferroni's correction ($P < 1.43 × 10^{-7}$), from which 186 also co-localized between the eQTL and GWAS traits when using coloc[28] (posterior probability for co-localization of significant signals PP4 > 0.7; Fig. 5a and Supplementary Fig. 21), confirming that the two traits shared the same causal SNP.

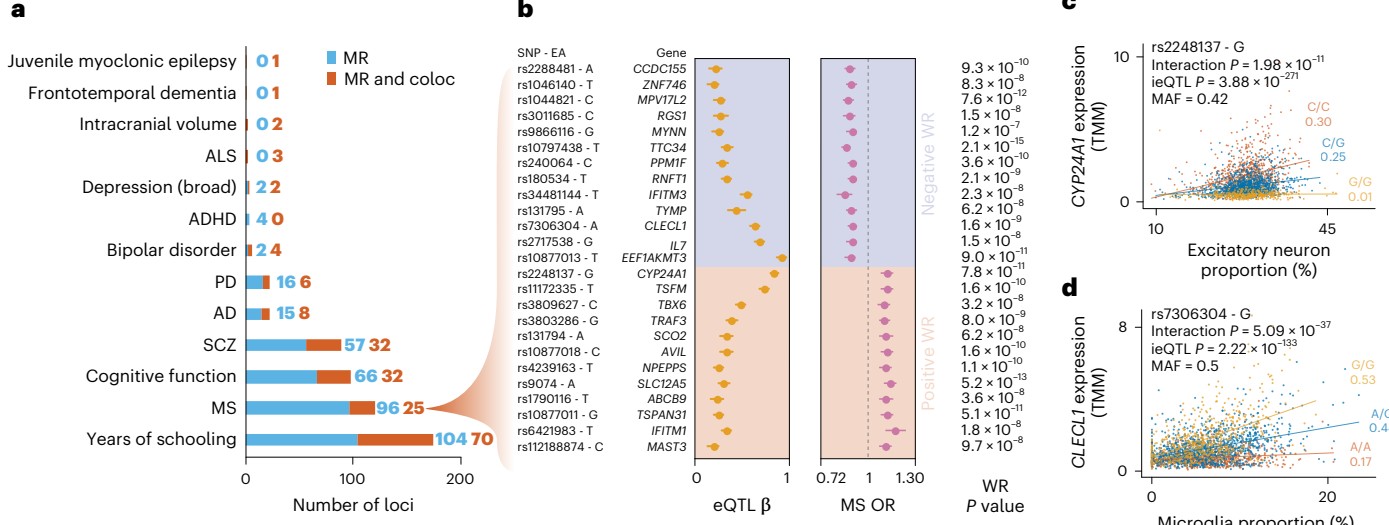

**Fig. 5 | Co-localization and MR analysis of brain-related traits. a**, Number of significant Wald ratio effects (blue) and those with both Wald ratio and co-localization (coloc; red) evidence for 13 brain-related traits. **b**, Forest plots showing the SNP and effect allele (EA), eQTL β and GWAS odds ratio for 20 MS-associated genes that are both MR and co-localization significant as well as their Wald ratio *P* value. The dots indicate the eQTL β or odds ratio, the error bars indicate the 95% confidence interval. WR, Wald ratio; and OR, odds ratio.

Grey dotted line indicates an OR of 1. **c,d**, Cell-type ieQTL for *CYP24A1* (**c**) and *CLECL1* (**d**) showing interactions with predicted excitatory neuron and microglia proportions, respectively. Each dot represents a sample. Colors indicate the SNP genotype, with yellow being the minor allele. Values under the alleles are the Pearson's correlation coefficients. Interaction *P* values were derived using a one-sided *F*-test; eQTL *P* values were derived using the standard normal distribution from meta-analyzed *z*-scores. TMM, trimmed mean of M-values.

Of the prioritized findings, 282 (82 of which co-localized) were associated with the risk for 31 prioritized neurological and neuropsychiatric diseases (Table 1). We focus on multiple sclerosis (MS) and highlight two examples where MR and co-localization point to probable causal GWAS genes. For other traits, see Supplementary Note, Supplementary Fig. 22 and Supplementary Tables 11–16.

**Tissue-specific genetic effects for MS**

Using MR analysis for MS[29], we identified 121 instruments (*cis*-eQTL SNPs) in 99 genes that passed the Bonferroni-adjusted *P*-value threshold of $1.43 \times 10^{-7}$ (Supplementary Table 12); 25 of these instruments passed co-localization (Table 1 and Fig. 5b), of which 13 genes had a positive Wald ratio—indicating that increased gene expression increases disease risk—and the remainder a negative Wald ratio, indicating the opposite. A systematic comparison of the Wald ratio estimates on the 7,748 shared *cis*-eQTL genes between Cortex-EUR and eQTLGen[14] (which instrumented the same genes but potentially with different SNPs) showed opposite effect directions for 3,173 (41.0%) genes (Supplementary Note, Supplementary Figs. 23,24 and Supplementary Table 14a). Although the agreement improved when the same SNP instrument was compared between studies, 2,671 (27.5%) of 9,728 MetaBrain Wald ratios still showed opposite directionality to eQTLGen (Supplementary Table 14b), underscoring the importance of tissue-specific differences when interpreting transcriptomics data.

Of the 172 genes with Wald ratio findings in Cortex-EUR, there were 47 without a significant eQTLGen instrument, including five (*RGS1*, *SCO2*, *SLC12A5*, *CCDC155* and *MYNN*) that passed the MR and co-localization significance thresholds in MetaBrain (Supplementary Note and Supplementary Table 16). In comparisons of the blood and brain expression levels of these genes in GTEx[30], *SLC12A5* and *CCDC155* had almost no expression in blood, whereas expression was comparable between tissues for *RGS1*, *SCO2* and *MYNN* (Supplementary Note and Supplementary Fig. 25). The discrepancy in MR findings observed between Cortex-EUR and eQTLGen suggest the existence of tissue-dependent genetic effects for MS.

**Cell type-specific *cis*-eQTLs linked to MS**

Two MS-associated genes, *CYP24A1* and *CLECL1*, showed cell type-specific *cis*-eQTLs (Fig. 5c,d). Another gene that was previously suggested to be neuron-specific[31], *SLC12A5*, did not show a significant ieQTL in our data. In our analysis, we found that higher *CYP24A1* expression is associated with increased risk for MS (Wald ratio = 0.13, $P = 7.8 \times 10^{-11}$) and that the eQTL and GWAS signals are co-localized (PP4 = 1.00). Furthermore, ieQTL analyses showed increasing expression of *CYP24A1* with increasing excitatory neuron proportions for the risk allele rs2248137-C (interaction β = 1.92, interaction $P = 1.98 \times 10^{-11}$; Fig. 5c), similar to other neurons (Supplementary Table 12). *CYP24A1* encodes for a protein that catalyzes the inactivation of 1,25-dihydroxyvitamin $D_3$ (calcitriol), the active form of vitamin D[32]. Epidemiological studies have proposed vitamin D deficiency as a risk factor for MS[33,34], which has recently been validated through MR[35–37]. Our findings are consistent with a previous report that indicates a shared signal for MS and *CYP24A1 cis*-eQTL in the frontal cortex[38].

Decreased expression of *CLECL1* was significantly associated with increased MS risk (Wald ratio = −0.16, $P = 1.58 \times 10^{-9}$) and showed clear co-localization (PP4 > 0.87). The ieQTL analysis indicated that rs7306304-A increased expression of *CLECL1* with increased proportions of microglia (interaction β = −2.72, interaction $P = 5.09 \times 10^{-37}$; Fig. 5d), confirming a previous finding of a microglia cell type-specific *cis*-eQTL for *CLECL1* at this MS risk locus[29]. This eQTL also replicates in the microglia single-cell analysis by Bryois et al.[24] (eQTL β = −0.62, $P = 3.2 \times 10^{-20}$) with the same direction of effect. The rs7306304 SNP is in strong LD with the MS lead SNP rs7977720 ($r^2 = 0.84$)[29]. *CLECL1* encodes a C-type lectin-like transmembrane protein expressed at high levels in dendritic and B cells, which has been proposed to modulate immune response[39]. It has 20-fold higher expression in a purified microglia dataset[29] than in cortical tissue, suggesting that decreased *CLECL1* increases MS susceptibility through microglia-mediated immune processes in the brain.

**Table 1 | Prioritized genes from the MR analysis on MetaBrain eQTLs for brain-related outcomes**

| Outcome | Gene | SNP | WR (SE) | P | CT | Outcome | Gene | SNP | WR (SE) | P | CT |
|---|---|---|---|---|---|---|---|---|---|---|---|
| AD | CR1 | rs679515 | 0.15 (0.01) | $1.40 \times 10^{-23}$ | OLI | MS | ZNF746 | rs1046140 | -0.57 (0.11) | $8.29 \times 10^{-08}$ | |
| AD | SLC39A13 | rs3740688 | -0.22 (0.03) | $1.13 \times 10^{-10}$ | | MS | MAST3 | rs112188874 | 0.56 (0.10) | $9.74 \times 10^{-08}$ | |
| AD | TSPAN14 | rs1902660 | 0.12 (0.02) | $4.73 \times 10^{-09}$ | | MS | MYNN | rs9866116 | -0.43 (0.08) | $1.22 \times 10^{-07}$ | |
| AD | APH1B | rs117618017 | 0.17 (0.03) | $1.05 \times 10^{-08}$ | | PD | KANSL1 | rs199451 | -0.25 (0.03) | $3.35 \times 10^{-19}$ | EX |
| AD | PRSS36 | rs78924645 | -0.09 (0.02) | $1.63 \times 10^{-08}$ | | PD | CD38 | rs4698412 | -0.24 (0.03) | $6.99 \times 10^{-14}$ | AST |
| AD | INPP5D | rs7569598 | 0.32 (0.06) | $1.74 \times 10^{-08}$ | | PD | HSD3B7 | rs11150600 | -0.46 (0.07) | $1.90 \times 10^{-10}$ | MIC |
| AD | ZNF668 | rs2359612 | 0.26 (0.05) | $1.00 \times 10^{-07}$ | | PD | SETD1A | rs35733741 | -0.67 (0.11) | $2.43 \times 10^{-09}$ | EX |
| AD | ACE | rs4291 | -0.10 (0.02) | $1.39 \times 10^{-07}$ | | PD | RAB29 | rs708723 | 0.21 (0.04) | $1.03 \times 10^{-08}$ | |
| ALS | SCFD1 | rs229243 | 0.17 (0.02) | $5.56 \times 10^{-15}$ | | PD | SCARB2 | rs7697073 | 0.29 (0.05) | $6.54 \times 10^{-08}$ | |
| ALS | G2E3 | rs229244 | -0.24 (0.03) | $4.23 \times 10^{-13}$ | | SCZ | PPP1R18 | rs9265954 | 0.48 (0.05) | $1.11 \times 10^{-19}$ | |
| ALS | MOBP | rs6772037 | 0.29 (0.05) | $1.10 \times 10^{-08}$ | | SCZ | HIST1H4K | rs13217285 | 0.63 (0.07) | $1.40 \times 10^{-18}$ | |
| BD | DCLK3 | rs9834970 | -0.32 (0.04) | $4.79 \times 10^{-14}$ | | SCZ | MICB | rs204999 | -0.37 (0.05) | $1.82 \times 10^{-15}$ | |
| BD | HAPLN4 | rs17216041 | 0.21 (0.04) | $4.44 \times 10^{-09}$ | | SCZ | FTCDNL1 | rs2949006 | -0.16 (0.02) | $7.97 \times 10^{-14}$ | OLI |
| BD | GNL3 | rs7646741 | 0.19 (0.03) | $1.07 \times 10^{-08}$ | | SCZ | GPANK1 | rs7773668 | 0.24 (0.03) | $2.56 \times 10^{-13}$ | |
| BD | LMAN2L | rs58361269 | -0.24 (0.04) | $1.78 \times 10^{-08}$ | | SCZ | FURIN | rs4702 | -0.23 (0.03) | $2.55 \times 10^{-12}$ | |
| MDD | NEGR1 | rs7531118 | 0.03 (0.00) | $4.07 \times 10^{-12}$ | | SCZ | SF3B1 | rs788018 | 0.21 (0.03) | $6.97 \times 10^{-11}$ | |
| MDD | SLC12A5 | rs9074 | 0.02 (0.00) | $8.03 \times 10^{-08}$ | | SCZ | MDK | rs35324223 | -0.29 (0.05) | $2.23 \times 10^{-10}$ | |
| FTD | BTNL2 | rs9268863 | 0.75 (0.12) | $2.20 \times 10^{-10}$ | | SCZ | THOC7 | rs832190 | -0.14 (0.02) | $6.19 \times 10^{-10}$ | NEU |
| JME | HSD3B7 | rs11150600 | 0.04 (0.01) | $3.96 \times 10^{-09}$ | MIC | SCZ | CNTN4 | rs17194427 | 0.26 (0.04) | $9.19 \times 10^{-10}$ | |
| MS | TTC34 | rs10797438 | -0.43 (0.05) | $2.06 \times 10^{-15}$ | | SCZ | TAOK2 | rs4788200 | 0.44 (0.07) | $1.05 \times 10^{-09}$ | |
| MS | SLC12A5 | rs9074 | 0.44 (0.06) | $5.15 \times 10^{-13}$ | | SCZ | PCCB | rs696520 | -0.12 (0.02) | $1.12 \times 10^{-09}$ | OLI |
| MS | MPV17L2 | rs1044821 | -0.51 (0.07) | $7.57 \times 10^{-12}$ | | SCZ | ATG13 | rs12574668 | -0.29 (0.05) | $1.23 \times 10^{-09}$ | |
| MS | TSPAN31 | rs10877011 | 0.48 (0.07) | $5.13 \times 10^{-11}$ | | SCZ | ASPHD1 | rs12919683 | -0.23 (0.04) | $1.33 \times 10^{-09}$ | OLI |
| MS | CYP24A1 | rs2248137 | 0.13 (0.02) | $7.81 \times 10^{-11}$ | EX + NEU | SCZ | TMEM219 | rs9925102 | 0.30 (0.05) | $1.87 \times 10^{-09}$ | |
| MS | EEF1AKMT3 | rs10877013 | -0.13 (0.02) | $8.99 \times 10^{-11}$ | MIC | SCZ | DOC2A | rs12921996 | 0.27 (0.05) | $1.97 \times 10^{-09}$ | |
| MS | NPEPPS | rs4239163 | 0.43 (0.07) | $1.14 \times 10^{-10}$ | | SCZ | PLEKHO1 | rs11577346 | -0.35 (0.06) | $1.97 \times 10^{-09}$ | |
| MS | AVIL | rs10877018 | 0.34 (0.05) | $1.57 \times 10^{-10}$ | | SCZ | INO80E | rs3814880 | 0.11 (0.02) | $2.47 \times 10^{-09}$ | |
| MS | TSFM | rs11172335 | 0.16 (0.02) | $1.57 \times 10^{-10}$ | NEU | SCZ | GNL3 | rs7646741 | 0.15 (0.03) | $3.46 \times 10^{-09}$ | |
| MS | PPM1F | rs240064 | -0.36 (0.06) | $3.58 \times 10^{-10}$ | | SCZ | SNAP91 | rs2022265 | 0.23 (0.04) | $6.34 \times 10^{-09}$ | |
| MS | CCDC155 | rs2288481 | -0.58 (0.09) | $9.33 \times 10^{-10}$ | | SCZ | TM6SF2 | rs2905432 | 0.26 (0.05) | $7.06 \times 10^{-09}$ | |
| MS | CLECL1 | rs7306304 | -0.16 (0.03) | $1.58 \times 10^{-09}$ | MIC | SCZ | KCTD13 | rs11150575 | 0.39 (0.07) | $9.30 \times 10^{-09}$ | |
| MS | RNFT1 | rs180534 | -0.30 (0.05) | $2.14 \times 10^{-09}$ | | SCZ | VPS45 | rs2319280 | -0.27 (0.05) | $1.17 \times 10^{-08}$ | |
| MS | TRAF3 | rs3803286 | 0.26 (0.04) | $8.03 \times 10^{-09}$ | | SCZ | CACNA1I | rs7288420 | 0.32 (0.06) | $1.22 \times 10^{-08}$ | |
| MS | IL7 | rs2717538 | -0.15 (0.03) | $1.46 \times 10^{-08}$ | EX | SCZ | GATAD2A | rs12975119 | -0.12 (0.02) | $1.34 \times 10^{-08}$ | |
| MS | RGS1 | rs3011685 | -0.47 (0.08) | $1.54 \times 10^{-08}$ | | SCZ | KMT2E | rs35601145 | 0.29 (0.05) | $1.64 \times 10^{-08}$ | |
| MS | IFITM1 | rs6421983 | 0.49 (0.09) | $1.78 \times 10^{-08}$ | MIC | SCZ | ZNF823 | rs72986630 | 0.19 (0.03) | $4.14 \times 10^{-08}$ | |
| MS | IFITM3 | rs34481144 | -0.30 (0.05) | $2.34 \times 10^{-08}$ | | SCZ | PTPRU | rs267700 | -0.19 (0.03) | $4.28 \times 10^{-08}$ | EX |
| MS | TBX6 | rs3809627 | 0.20 (0.04) | $3.25 \times 10^{-08}$ | | SCZ | RERE | rs301792 | 0.16 (0.03) | $4.99 \times 10^{-08}$ | |
| MS | ABCB9 | rs1790116 | 0.47 (0.09) | $3.55 \times 10^{-08}$ | | SCZ | GLYCTK | rs6445358 | 0.19 (0.04) | $7.22 \times 10^{-08}$ | |
| MS | TYMP | rs131795 | -0.25 (0.05) | $6.20 \times 10^{-08}$ | | SCZ | ATP13A1 | rs7245672 | 0.30 (0.06) | $1.08 \times 10^{-07}$ | |
| MS | SCO2 | rs131794 | 0.33 (0.06) | $6.20 \times 10^{-08}$ | | SCZ | CLCN3 | rs72696657 | 0.24 (0.05) | $1.14 \times 10^{-07}$ | |

Harmonized eQTL and GWAS SNP effects and single-SNP Wald ratio-effect (WR) estimates are reported for all genes with Wald ratio effects at $P < 1.865 \times 10^{-7}$. Brain-trait outcomes have been abbreviated as follows: BD, bipolar disorder; MDD, major depressive disorder (broad depression category); FTD, frontotemporal dementia; JME, juvenile myoclonic epilepsy. The cell types (CT) with which the eQTL significantly interacts (BH-FDR < 0.05) are abbreviated as follows: AST, astrocytes; EX, excitatory neurons; MIC, microglia; NEU, other neuron; and OLI, oligodendrocytes. This table is a subset of Supplementary Table 12.

## MetaBrain allows for the identification of *trans*-eQTLs

*Trans*-eQTLs can identify the downstream consequences of disease-associated variants but their effects are usually small[14]. To maximize power, we combined the Cortex-EUR and -AFR datasets (*n* = 2,759, excluding the ENA). We reduced the multiple-testing burden by focusing on 228,819 unique genetic variants, including GWAS and *cis*-eQTL variants.

When correcting for an increasing number of principal components (PCs), we observed a decrease in the number of *trans*-eQTLs

(Fig. 6a, Supplementary Note, Supplementary Fig. 7 and Supplementary Table 17) as well as in heterogeneity (Supplementary Fig. 26). The majority (85%) of the *trans*-eQTLs observed without PC correction were located in a 7p21.3 locus previously associated with frontotemporal lobar degeneration[40], Alzheimer's disease (AD)[41] as well as changes in neuron proportions[42] and gene expression levels[43,44]. We did not find evidence that these *trans*-eQTLs were dependent on AD status or neuron proportions and they were not significant when correcting for 100 PCs (Supplementary Note, Supplementary Figs. 26–29 and Supplementary Tables 17–24).

We therefore concentrated on 737 *trans*-eQTLs, detected after correcting for 100 PCs, which reflect 526 unique SNPs and 108 unique genes; 127 SNPs had *trans*-eQTL effects on multiple genes and 461 (88%) of the *trans*-eQTL SNPs were associated with a significant *cis*-eQTL in Cortex-EUR. We observed that 150 (33%) of the 461 significant *trans*-eQTL SNPs overlapping a *cis*-eQTL SNP were also the *cis*-eQTL index SNP, which represents an enrichment (Fisher's exact test, $P = 1.2 \times 10^{-28}$; Supplementary Note); 29 were also *cis*-eQTL SNPs in tissues other than the cortex (Supplementary Table 17). This indicates that *cis*-eQTL index SNPs yield *trans*-eQTL effects more often in the brain in comparison to other *cis*-eQTL variants.

Significant interactions with predicted cell-type proportions (BH-FDR < 0.05; Supplementary Table 22)—oligodendrocytes ($n = 27$), other neurons ($n = 7$), astrocytes ($n = 7$) and microglia ($n = 2$)—were also observed for 41 *trans*-eQTLs (5.9%). Four eQTLs—all influencing *DTX4* and dependent on oligodendrocyte proportion—replicated significantly in the ROSMAP snRNA-seq dataset with the same direction of effect (Supplementary Fig. 30 and Supplementary Table 23).

We observed *trans*-eQTLs from multiple independent genomic loci for seven genes, suggesting convergent *trans*-eQTL effects (*ARRDC4*, *HBG2*, *POP1*, *COX7A1*, *RFPL2*, *ZNF311* and *ZNF404*; Supplementary Table 17). This includes a convergent *trans*-eQTL on hemoglobin subunit γ-2 (*HBG2*; 11p15.4) that was previously identified in blood. *HBG2* was affected in *trans* by two independent variants (rs1427407 on 2p16.1 and rs4895441 on 6q23.3; Fig. 6b), which have previously been associated with fetal hemoglobin levels[45–47]. We also found converging effects that were not identified in blood. For example, the *ZNF311* gene (6p22.1) was affected by the rs1150668 variant in *cis* and the rs8106871 variant in *trans* (19q13.2), both of which have been previously associated with smoking[48] and risk tolerance[49]. For both associations, the risk allele also increased *ZNF311* expression. Furthermore, the risk allele rs1150668-G increased the expression of *S100A5* in *trans*, and rs8106871-T decreased the expression of *POU2F2* and increased expression of *DEDD2* in *cis* (Fig. 6b). *ZNF311* has been suggested to be a tumor-suppressor gene[50] potentially involved in gliomas[51], *S100A5* is used as a biomarker for astrocytomas[52] and *POU2F2* has previously been associated with glioblastoma[53]. This example shows how multiple variants associated with smoking may alter multiple genes involved in cancer.

**Brain co-regulation networks aid in GWAS interpretation**

We generated brain region-specific co-regulation networks based on the RNA-seq data from 8,544 samples (Supplementary Note and Supplementary Figs. 31–33) using a similar approach to our previously developed multi-tissue GeneNetwork ($n = 31,499$)[54,55]. We applied Downstreamer[56] to SCZ[57], PD[58], MS[29], AD[59] and amyotrophic lateral sclerosis (ALS) GWAS summary statistics[60], using these networks to prioritize genes that are co-regulated with genes in their GWAS loci (Supplementary Note, Supplementary Fig. 34 and Supplementary Tables 25–30). For MS and AD, these were mostly immunity genes, whereas for PD, ALS and SCZ, these were genes that are specifically expressed in the brain (Supplementary Tables 25–30). For ALS and MS, we additionally created smaller networks for the cerebellum ($n = 715$) and cortex ($n = 6,526$) to identify brain region-specific effects.

For ALS, we applied Downstreamer to summary statistics from individuals with EUR ancestry (Supplementary Table 30) and a

trans-ancestry meta-analysis including individuals with EUR and Asian ancestry[60] (EUR + ASN; Supplementary Table 25). In contrast, whereas Downstreamer did not identify genes using GeneNetwork[55] ($n = 31,499$), we identified a set of 27 unique co-regulated genes when applied to the smaller brain co-regulation networks (EUR + ASN summary statistics; Fig. 7a and Supplementary Table 25). Of the identified genes, *HUWE1* was shared between the results from all brain regions and separate results from cortex, whereas *UBR4* was shared between the cortex and cerebellum results. *UBR4* encodes a ubiquitin ligase protein expressed throughout the body, which interacts with calmodulin, a protein regulating $Ca^{2+}$—a process which has been linked to ALS disease-associated genes and motor-neuron vulnerability[61]. Furthermore, a previously discovered private mutation in *UBR4* implicates its role in muscle coordination[62]. Many of the genes prioritized by Downstreamer are co-regulated with each other (Fig. 7b) and were enriched for genes implicated in causing gait disturbances (Fig. 7c). Our analysis identified genes that show strong co-regulation with positional candidate genes inside ALS-associated loci, suggesting that they must have a shared biological function.

For MS, the GeneNetwork[55] identified 257 unique genes that showed significant co-regulation with genes inside MS-associated loci (Fig. 7d and Supplementary Table 29), many of which were immunity genes, which is also expected for this disease. However, when we used the brain co-regulation networks, we identified a much smaller set of genes that showed strong enrichment for the neurotrophin signaling pathway (Fig. 7e,f). Neurotrophins are polypeptides secreted by immunological cell types and promote the survival and proliferation of neurons as well as synaptic transmission (Supplementary Note). The identified genes showed high expression in immunity-related tissues when using the GeneNetwork[55] (Supplementary Fig. 31a) and high expression in the spinal cord but low expression in cortex samples (Supplementary Fig. 31b). This could implicate specific brain regions with MS development: for instance, the cerebellum is involved in muscle coordination and ataxia occurs in approximately 80% of patients with symptoms of MS[63]. But this could also implicate the 'outside-in hypothesis' that suggests the immune system may be a potential trigger for MS[29,64].

## Discussion

We describe an integrated analysis of the effects of genetic variation on gene expression levels in the brain with a sufficient sample size to identify robust *cis*-eQTLs and cell-type ieQTLs to compare *cis*-eQTLs between ancestries and identify brain *trans*-eQTLs that emanate from SNPs that have been previously linked to brain-related diseases. We have released harmonized results of the individual datasets to help others determine the robustness of eQTL effects.

We showed that eQTL-effect directions are generally shared between datasets, tissues and ancestries, but note that opposite allelic effects exist, which became apparent especially when we compared our results with a large blood eQTL dataset. We also identified non-primary *cis*-eQTLs, some of which reflect SNPs that are also the index variants for brain-related disorders, making them particularly interesting for subsequent follow-up.

We predicted cell-type proportions in the cortex and identified 3,549 cell-type ieQTLs. We compared the ieQTLs with snRNA-seq eQTLs and observed that the $\pi_1$ estimates were low due to the low sample size or other limitations in the snRNA-seq datasets[24,65,66]. As we observed good $R_b$ and AC values, we expect that the overlap and replication rates will improve once the sample sizes of snRNA-seq studies increase, highlighting the potential of ieQTL analysis in bulk RNA-seq.

This is a well-powered MR and co-localization analysis using brain *cis*-eQTLs as instruments for bipolar disease, epilepsy, frontotemporal dementia, MS, cognitive function and years of schooling GWAS outcomes. Interestingly, for SCZ, three signals for *CILP2*, *MAU2* and *TM6SF2* met our criteria that had not been reported in a recent

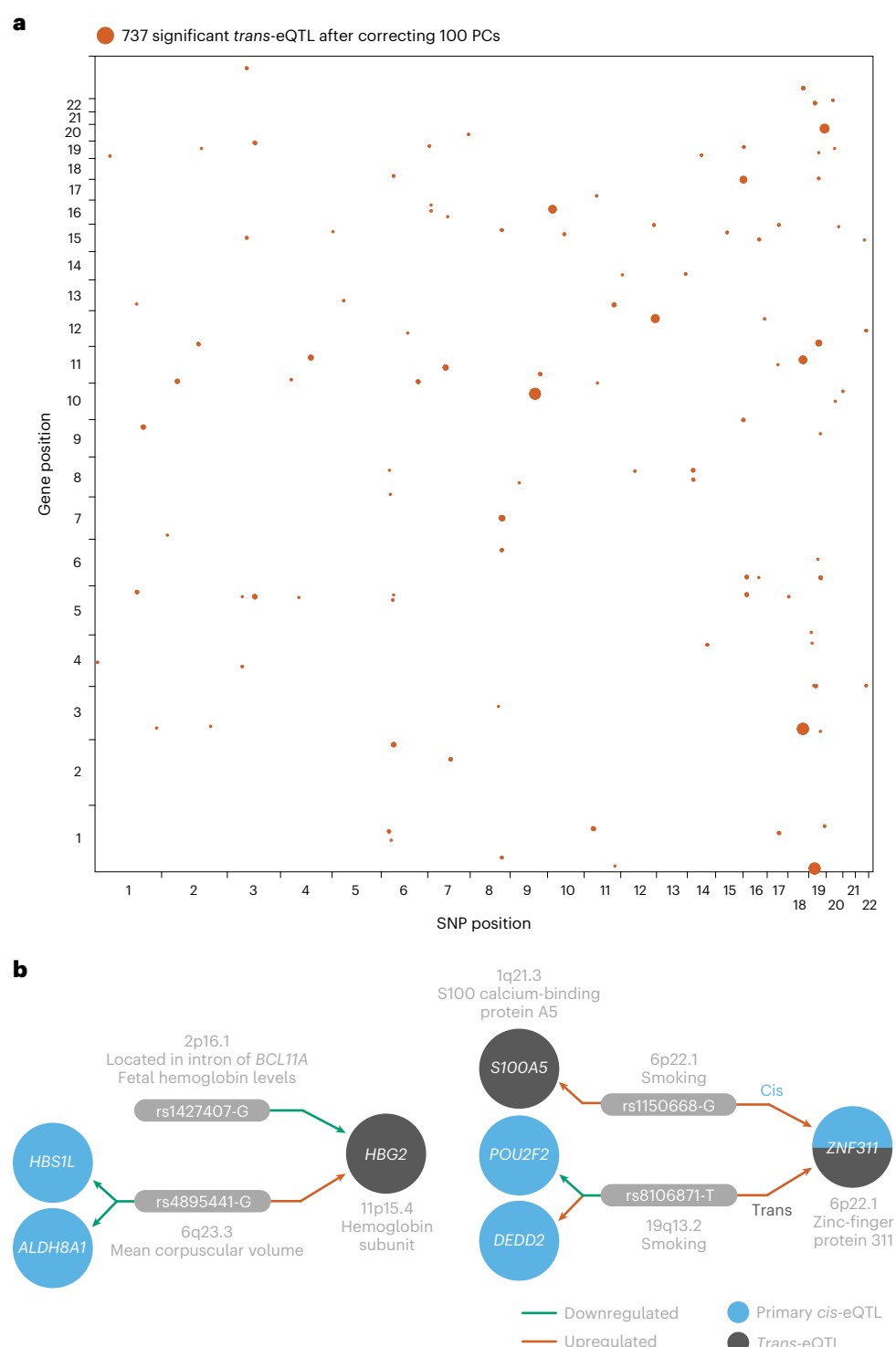

**Fig. 6 | *Trans*-eQTLs in the brain. a**, Location of the identified *trans*-eQTLs (SNP and gene positions) in the genome. The size of the dots indicates the *P* value of the *trans*-eQTL (larger is more significant). **b**, Two examples of convergent effects, where multiple independent SNPs affect the same genes in *trans*.

*Trans*-eQTLs of rs1427407 and rs4895441 on *HBG2* (top). *Trans*-eQTL of rs1150668 and rs106871 on *ZNF31* and *S100A5* (bottom). Both panels are derived from Supplementary Table 17.

psychiatric genomics consortium study[67], further emphasizing the value of our large eQTL dataset (Supplementary Note). Our results also identify increased *CYP24A1* expression to be associated with MS risk and propose excitatory neurons as the cell type most susceptible to *CYP24A1* expression changes and probably active vitamin D levels. The potentially novel role of *CYP24A1* in the brain could play an important role in MS etiology, as may lowered expression of *CLECL1* in microglia.

The analyses identified *trans*-eQTLs in the brain cortex for many variants, some of which are brain-specific. Similar to blood, the *trans*-eQTL-effect sizes in the brain were usually small, emphasizing the importance of increasing the sample size of brain eQTL studies. The identified *trans*-eQTLs allowed us to gain insights into the functional effect of several disease-associated variants. We observed that *trans*-eQTL SNPs were enriched for *cis*-eQTL index SNPs, indicating

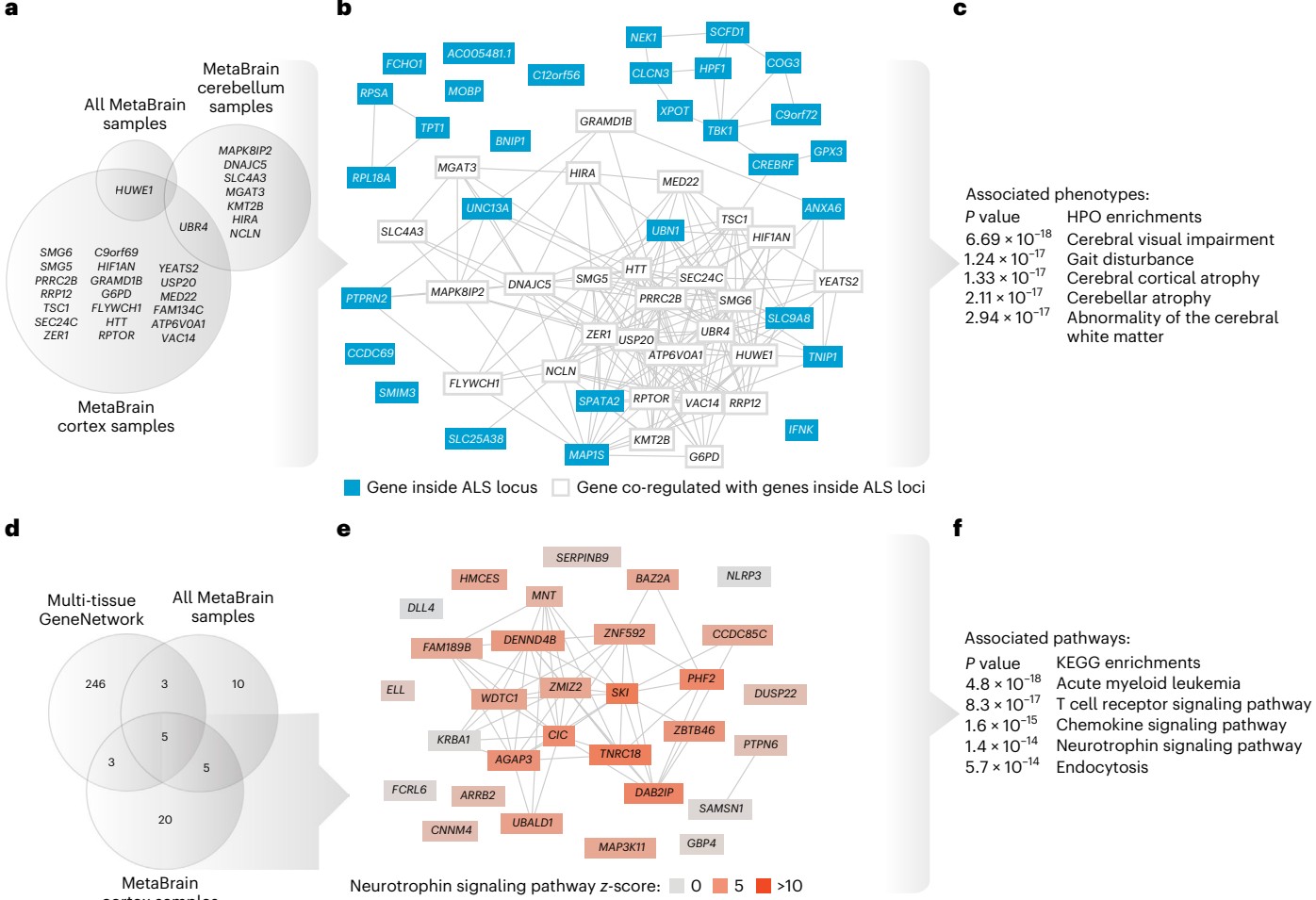

**Fig. 7 | Gene co-regulation. a**, Genes that are co-regulated with genes that are within ALS-associated loci. Co-regulation scores between genes were calculated using the MetaBrain cerebellum and MetaBrain cortex samples as well as the combined (all) MetaBrain samples. Except for *URB4,* cortex and cerebellum networks find different co-regulated genes for ALS. **b**, Co-regulation network using all MetaBrain samples for all genes prioritized for ALS by Downstreamer. **c**, Top five Human Phenotype Ontology (HPO) enrichments for the Downstreamer prioritized ALS-associated genes. **d**, Genes that are co-regulated with genes that are within MS-associated loci. Co-regulation scores between genes were calculated using a heterogeneous multi-tissue network, MetaBrain cerebellum samples and MetaBrain cortex samples. Most genes were found using a large heterogenous co-regulation network. **e**, Co-regulation network of all MetaBrain samples for 33 genes prioritized by Downstreamer in cortex. **f**, Top five Kyoto Encyclopedia of Genes and Genomes (KEGG) enrichments for the Downstreamer prioritized MS genes in the cortex highlight neurotrophin signaling pathway enrichment (red). **c**,**f**, Enrichment *P* values were calculated using a two-sided $\chi^2$ test. Panels **a** and **d** are derived from Supplementary Tables 25, 29 and 30.

consequences beyond the gene regulated in *cis*. Our *trans*-eQTL analysis focused on a single brain region and was limited to trait-associated and *cis*-eQTL SNPs. We expect that a genome-wide approach will identify many more *trans*-eQTLs. Furthermore, we note that true separation of *cis*- and *trans*-eQTL effects would require investigation of the allelic function of the associated SNPs (Supplementary Note).

We used brain-specific co-regulation networks to study several brain-related GWAS studies and prioritized genes that show significantly enriched co-regulation with genes inside GWAS loci. For ALS, this revealed a limited but significant set of genes located outside of currently known ALS loci, which might lead to a better understanding of the poorly understood pathways that cause ALS.

This study has several limitations. First, our eQTL analyses were limited to single tissues and excluded replicate RNA-seq measurements. A joint analysis with random-effects models[68,69] could increase the effective sample size, which would be especially useful for *trans*-eQTL identification. Second, our GWAS overlap analysis may have failed to identify previously identified genes due to differences in sample size, effect size, variant density, LD structure and imputation quality. For example, our results did not include the *MAPT* gene for AD because

the H1/H2 haplotype separating SNP rs8070723 had an eQTL *P* value of $1.8 \times 10^{-5}$ due to our alignment strategy (Supplementary Note). This might have been an issue for other genes as well. Graph-based alignment tools or long-read sequencing methods are required to ultimately determine the true effects on such genes. Third, the GWAS overlap methods we used have known limitations (for example, Supplementary Note). For the MR analysis, we opted to perform single-SNP MR instead of multi-SNP MR (such as inverse-variance weighted[70]), which requires multiple independent associations per gene. As this was the case for only a limited proportion of the tested *cis*-eQTLs and there were no genes with more than five independent eQTLs, we reasoned this would not provide for reliable inverse-variance-weighted estimation. Inverse-variance-weighted estimation could potentially be applied on a genome-wide *trans*-eQTL analysis, resulting in many more independent instruments per gene. However, such an approach would be more susceptible to confounding because of horizontal pleiotropy[71], where a gene is affected by multiple indirect effects. Finally, our co-localization approach was based on the single-causal-variant assumption, which is not applicable to *cis*-eQTL genes with multiple independent associations (for example, *TREM2*; Supplementary Note),

and therefore we may have failed to detect co-localizing signals in such loci. Recently published co-localization methods[72] do not have this assumption, which may improve future co-localization results. Finally, it is possible that bulk RNA-seq eQTL studies generally capture eQTL effects for genes that are not dosage sensitive and do not cause disruptive downstream consequences[73]. Furthermore, many eQTLs can only be detected in certain contexts[74] for which single-cell experiments are best suited.

We expect that this resource will prove valuable for post-GWAS brain research. Our dataset can be utilized to disambiguate GWAS loci, point to causal pathways and prioritize targets for drug discovery. We expect that through future integration with single-cell eQTL studies that have higher resolution but still lower power, our results will help to pinpoint transcriptional effects in specific brain cell types for many disease-associated genetic variants.

## Online content

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

## Methods

### Dataset collection and description

We collected published bulk brain RNA-seq samples from the AMP-AD consortium (AMP-AD MAYO, ROSMAP and MSBB)[6], Braineac[7] and the PsychENCODE consortium[8] (Bipseq[4], BrainGVEX[4], CMC[9], CMC_HBCC and UCLA_ASD[4]) from Synapse.org using the Python 2.7.5 package synapseclient v1.9.2 (ref. [75]). The NABEC and GTEx datasets were retrieved from NCBI dbGaP and the TargetALS data were provided directly by the investigators. In addition, we collected brain bulk RNA-seq samples from the ENA and performed rigorous QC, resulting in 1,759 samples (Supplementary Note, Supplementary Fig. 1 and Supplementary Table 28). Combined with the other datasets, this yielded a total of 9,363 samples (Supplementary Table 1).

### RNA-seq alignment and QC

RNA-seq data were processed using a pipeline built with molgenis-compute[76]. FASTQ files were aligned against the GRCh38. p13 primary assembly using the GENCODE[77] v32 annotation with STAR[78] (v2.6.1c), while excluding patch sequences (Supplementary Note) with the following parameter settings: outFilterMultimapNmax = 1, twopassMode Basic and outFilterMismatchNmax = 8 for paired-end sequences; and outFilterMismatchNmax = 4 for single-end sequences. Gene quantification was performed using STAR, similar to gene quantification using HTSeq[79], with default settings. The gene counts were then TMM-normalized[80] per cohort using edgeR[81] (v3.20.9) with R[82] (v3.5.1). Quantification for the GeneNetwork was done using Kallisto[83] v0.43.1 (Supplementary Note).

To measure the FASTQ and alignment quality we used FastQC[84] (v0.11.3), STAR metrics and Picard Tools[85] metrics (v2.18.26; MultipleMetrics and RNAseqMetrics). Samples were filtered out if aligned reads had <10% coding bases (Supplementary Fig. 4a), <60% reads aligned (Supplementary Fig. 4b) or <60% unique mapping. Among the RNA-seq samples, 117 did not pass this filter, mostly from GTEx[81]. The other quality measurements were visually inspected but contained no outliers. To identify outliers that had not been captured by these statistics, we performed a PCA-based filtering approach, after which 8,868 samples remained (Supplementary Note and Supplementary Fig. 5a–c). To adjust for between-dataset differences observed in the data (Supplementary Fig. 6a), we correlated the RNA-seq data with 77 covariates from the different QC tools and regressed-out the top-20 correlated covariates using ordinary least squares (OLS; Supplementary Note), after which clustering of datasets in PC1 and PC2 were no longer present (Supplementary Fig. 6b).

Our collection of RNA-seq samples consisted of 36 different tissue labels, many of which were represented by only a few samples. Therefore, we next defined major brain regions present in our dataset, including samples from the amygdala, basal ganglia, cerebellum, cortex, hippocampus and spinal cord. We noted that some samples (especially from the ENA) were not annotated with a specific major brain region. To resolve this, we performed PCA over the sample correlation matrix and then performed $k$-nearest neighbors on the first two PCs ($k$ = 7) to classify samples to the major brain regions. Using this approach, we defined a set of 86 amygdala, 574 basal ganglia, 723 cerebellum, 6,601 cortex, 206 hippocampus, 252 hypothalamus and 285 spinal cord samples (Fig. 2a and Supplementary Table 1).

### Genotype QC and definition of eQTL datasets

The genotype data for the included datasets were generated using different platforms, including genotypes called from whole-genome sequencing (AMP-AD, TargetALS[12] and GTEx[3]), genotyping arrays (NABEC[11] and Braineac[7]) and haplotype reference consortium[86]-imputed genotypes (PsychENCODE datasets), or were called from RNA-seq directly (ENA dataset; Supplementary Note). A total of 22 different genotyping datasets were available, reflecting 6,658 genotype samples (Supplementary Table 1). We performed QC on each

dataset separately, using slightly different approaches per platform (Supplementary Note) and used a PCA-based approach to assign ancestries to each individual sample. Most of the included samples were of EUR ancestry: 5,138 samples had an EUR assignment, 805 samples had an AFR assignment and 573 samples were assigned to the other ancestries (Fig. 2b and Supplementary Table 1). We next assessed links between RNA-seq samples and genotyped individuals, and were able to identify 7,644 links (Supplementary Table 1). For eQTL discovery, we grouped these links based on brain region and ancestry, in which we required at least 30 samples spanning at least two cohorts. We next removed sample mix-ups and duplicate samples (Supplementary Note), resulting in the following final eQTL discovery datasets: Basal ganglia-EUR ($n$ = 208), Cerebellum-EUR ($n$ = 492), Cortex-EUR ($n$ = 2,683), Cortex-AFR ($n$ = 319), Hippocampus-EUR ($n$ = 208) and Spinal cord-EUR ($n$ = 108; Fig. 2c and Supplementary Table 1).

### eQTL analysis

We performed cis-eQTL analysis in each of the eQTL discovery datasets by calculating Spearman correlations within each cohort, followed by a sample size-weighted $z$-score meta-analysis approach, as described previously[14]. We opted for this approach to minimize the influence of potential heterogeneity between cohorts and showed that it performs comparably to FastQTL/QTLTools[87] (Supplementary Note and Supplementary Fig. 35). To correct for multiple testing, we used an approach similar to FastQTL/QTLTools[87], where we used 1,000 permutations of the sample labels to fit a β-distribution per gene and, after adjustment using this distribution, calculated the $q$-values[88] over the top association per gene to determine significance (Supplementary Note). Genes with $q$-value < 0.05 were deemed significant. We limited these analyses to 19,373 protein-coding genes and to SNPs located within 1 Mb of the TSS, with MAF > 1% and Hardy–Weinberg $P$ > 0.0001. The RNA-seq data were corrected for up to 20 technical covariates, dataset indicator variables and four multidimensional scaling components derived from the genotype data using OLS. In addition, we evaluated the impact of regressing out increasing numbers of PCs and defined the optimal numbers of PCs to remove for each eQTL discovery dataset (Supplementary Note and Supplementary Fig. 7). To identify secondary, tertiary, quaternary and other non-primary cis-eQTLs, we repeated the procedure in an iterative conditional approach, where in each subsequent iteration, we regressed out the cis-eQTL effect of the previous iterations using OLS and identified cis-eQTLs using the residuals, followed by an LD pruning step to circumvent SNP missingness between included cohorts (Supplementary Note).

To identify trans-eQTLs, we performed a limited analysis to reduce the multiple-testing burden by focusing on 228,819 variants with a known interpretation. This set constituted variants that were either previously associated with traits, having a GWAS $P$ < $5 \times 10^{-8}$ in the IEU OpenGWAS database[89] or EBI GWAS catalog[90] on 3 May 2020, and additional neurological traits (Supplementary Table 17) or that showed an association with $q$-value < 0.05 in any of our discovery cis-eQTL analyses (including non-primary associations identified in the iterative conditional analysis). To maximize power, we combined the Cortex-EUR and Cortex-AFR datasets but excluded the ENA cohort due to the potential for genotypes of poorer quality ($n$ = 2,759; Supplementary Note). For this dataset, we also repeated the cis-eQTL analysis (Supplementary Table 2) and normalization approach, including the selection of optimal number of 100 PCs to regress out (Supplementary Fig. 7). We assessed those combinations of SNPs and genes where the SNP–TSS distance was >5 Mb or where the gene and SNP were on different chromosomes as trans-eQTLs. To determine significance, we employed a previously used FDR estimation method[91] using ten genome-wide permutations (Supplementary Note) and deemed trans-eQTLs with an FDR < 0.05 significant. We finally used an alignment-based approach to detect potential crossmapping artifacts, after which the FDR estimates were recalculated (Supplementary Note).

## eQTL agreement

We used four different measurements of agreement of eQTL effects when comparing different brain regions or tissues: AC, $\pi_1$, $R_b$ and caFC[92]. Each of these measures evaluates different aspects of replication: AC is an indication of the proportion of effects that have a shared direction of effect within the set of eQTLs that is significant in both the discovery and replication datasets and is expected to be 50% for random eQTL effects, $\pi_1$[93] estimates the proportion of eQTL effects that are true positive in the replication cohort but does not take into account the effect direction and can be dependent on the replication dataset sample size, $R_b$[94] effectively estimates the correlation between the eQTL-effect slopes (for example, β values from linear regression) while controlling for potential covariance in standard errors of those slopes and caFC measures the correlation between estimates of the fold change in expression values between alleles[92]. We note that whereas AC, $\pi_1$ and $R_b$ can be calculated from summary statistics, caFC requires access to genotype and expression data. We therefore limited the caFC analysis to comparisons within the MetaBrain datasets and comparisons with the GTEx tissues. Details on how we calculated these measures of agreement are in the Supplementary Note. For AC, $\pi_1$ and $R_b$ comparisons with GTEx, we used the summary statistics for GTEx-v8 that were downloaded from the GTEx portal website. Calculations of the aFC per eQTL were performed with the original aFC script by Mohammadi and colleagues[92] using the settings –log_xform 1 and –log_base 2, after which Pearson correlation was used to calculate caFC across shared eQTLs. For all comparisons with GTEx tissues, we performed discovery in Cortex-EUR while excluding the GTEx cohort.

## Identification of cell type-dependent eQTL effects in bulk RNA-seq

We predicted cell-type proportions of the MetaBrain Cortex-EUR and -AFR datasets using the method and single-cell profiles previously published by the PsychENCODE consortium[4] (Supplementary Note). We decided to discard the developmental cell types as we expected that these cell types are very rare or not present in adult human brain and because their signatures were obtained from fetal cells. The remaining cell types included all major cell types in the brain: neurons (excitatory, inhibitory and other), oligodendrocytes, astrocytes, microglia and endothelial cells. We then predicted the cell-type proportions as previously described[4] (Supplementary Note). However, to enable the joint analysis of samples, we chose to correct the $\log_2$-transformed transcript-per-million gene counts for 20 RNA-seq quality metrics using OLS as we observed that this removed dataset biases in the predictions. To maintain the information captured by relative expression differences between genes required for deconvolution, we rescaled the residuals to the original $\log_2$-transformed mean and standard deviation, and replaced negative values with zero. For the deconvolution step, we used the non-negative least squares[95] implementation in SciPy (v1.4.1)[96]. Given that the average proportions of cell subtypes were often very low (that is, <1%; Supplementary Fig. 16), we opted to sum all the subtypes of cells for excitatory neurons, inhibitory neurons and oligodendrocytes (oligodendrocyte precursor cells and oligodendrocytes). We observed a high correlation between the predicted cell proportions and PCs, indicating that cell-proportion differences contribute to a substantial variance in bulk gene expression levels (Supplementary Fig. 36).

Using the predicted cell-type proportions, we aimed to identify cell type-dependent eQTLs. To increase the robustness of our results, we excluded 50 samples with a cell-proportion $z$-score > 4 on one or more cell type and limited the analysis to eQTLs with <95% missingness per dataset, a joint MAF > 5% and a joint Hardy–Weinberg $P$ < 0.0001. With the remaining 25,497 eQTLs and 2,633 samples, we used Decon-QTL[19], which employs a non-negative least-squares model to identify cell-type interaction effects. For this analysis, we used the steps as described in the Decon-QTL manuscript[19]. For the pre-processing of the TMM

expression counts, we corrected for dataset indicator variables, 20 RNA-seq alignment metrics and four genotype multidimensional scaling components using OLS. As an additional step, we forced the data to the normal distribution per gene to reduce outliers. Finally, we evaluated whether the multiple-testing correction applied by Decon-QTL properly reflects the null distribution by comparing a permutation-based method to the default BH-FDR multiple-testing correction. We found that the vast majority (87.76%) of FDR significant interactions were also significant using permutations (Supplementary Note and Supplementary Figs. 37,38).

To confirm cell type-specific eQTL effects identified in Cortex-EUR, we used three replication datasets: Cortex-AFR and the snRNA-seq datasets Bryois et al.[24] and ROSMAP[4]. For the Cortex-AFR replication, we applied the same cell-type prediction and Decon-QTL interaction analysis as for Cortex-EUR. Over the ieQTLs significant in Cortex-EUR, we calculated BH-FDR estimates and deemed ieQTLs with a BH-FDR < 0.05 as significant. Given that Decon-QTL does not return any standard errors, we predicted β and standard errors using the sample size, MAF, interaction β and interaction $P$ value[97] to calculate the $R_b$ metrics. For the ROSMAP dataset, encompassing 80,660 single-nucleus transcriptomes from the prefrontal cortex of 48 individuals with varying degrees of AD pathology[23], we re-processed the expression matrix to create a pseudo-bulk expression matrix for each broad cell type and subsequently mapped *cis*-eQTLs using the same procedure as the *trans*-eQTL analysis in bulk data (Supplementary Note and Supplementary Figs. 39,40). To correct for multiple testing, we confined the analysis to only test for primary *cis*- or *trans*-eQTLs that had a significant interaction with one or more cell types in MetaBrain Cortex-EUR (BH-FDR < 0.05), while also permuting the sample labels 100 times. Finally, we attempted replication of our findings using the snRNA-seq eQTL summary statistics of the recent preprint by Bryois and colleagues[24]. We overlapped their summary statistics with the Cortex-EUR *cis*-eQTLs and found (depending on the cell type) that between 9,402 and 13,764 overlapped. We calculated a BH-FDR on the $P$ values of the ieQTLs that were significant in Cortex-EUR in the respective cell type. Given that the summary statistics did not include standard errors or MAF values, we predicted β and standard errors using the MetaBrain Cortex-EUR MAF together with the eQTL sample size, β and $P$ value[97] from Bryois et al. to calculate the $R_b$ metrics.

## Single-SNP MR analysis

We conducted MR between the Cortex-EUR eQTLs and 31 neurological traits (21 neurological disease outcomes, two quantitative traits and eight brain-volume outcomes; Supplementary Table 11). For this purpose, we used the Wald ratio method, which computes the change in disease risk per s.d. change in gene expression, explained through the *cis*-eQTL instrument(s) for that gene. To obtain our instruments, Cortex-EUR eQTLs at a genome-wide significant $P$-value threshold ($P$ < $5 \times 10^{-8}$) were selected and then LD-clumped using the ld_clump() function in the ieugwasr package v0.1.4 (ref. [98]) with the default settings (10,000 kb clumping window with $r^2$ cut-off of 0.001 using the 1000 Genomes EUR reference panel). SNP associations for each of the eQTL instruments were then looked up in the outcome GWAS. If the SNP could not be found in the outcome GWAS using the dbSNP rsid, then a proxy search was performed to extract the next closest SNP available in terms of pairwise LD, providing a minimum $r^2$ threshold of 0.8 with the eQTL. These steps were performed using the associations() function in the ieugwasr package. To ensure correct orientation of effect alleles between the eQTL and outcome associations, the SNP effects were harmonized using the harmonise_data() function in TwoSampleMR[70] selecting Action 2, which assumes that the alleles are forward-stranded in the GWASs (so no filtering or re-orientation of alleles according to frequency was conducted on the palindromic SNPs). Single-SNP MR was then performed using the mr_singlesnp() function in TwoSampleMR. We reported all of the MR findings that passed a $P$-value threshold of

$5 \times 10^{-5}$ but note that the Bonferroni-corrected $P = 0.05$ threshold for multiple-testing correction is $P = 1.43 \times 10^{-7}$.

## Co-localization

Following the MR analysis, co-localization analysis was performed on the MR findings that passed the suggestive threshold to determine whether the eQTL and trait shared the same underlying signal. We ran co-localization[28] using both the default parameters (p1 = p2 = $10^{-4}$ and p12 = $10^{-5}$) and parameters based on the number of SNPs in the region (p1 = p2 = 1/(number of SNPs in the region) and p12 = p1/10). We considered the traits to co-localize if either of the parameter runs yielded PP4 > 0.7. In addition, a systematic co-localization analysis was performed to compare findings between Cortex-EUR eQTLs and other existing cortex eQTL datasets (Supplementary Note).

## Ethical compliance

All cohorts included in this study enrolled participants with informed consent and collected and analyzed data in accordance with ethical and institutional regulations. Information about individual institutional review board approvals is available in the original publications for each cohort. Where applicable, data access agreements were signed by the investigators previous to acquisition of the data, either to the UMCG (AMP-AD, CMC, GTEx, CMC_HBCC, BrainSeq, UCLA_ASD, BrainGVEX, BipSeq and NABEC) or Biogen (TargetALS and Braineac), which state the data usage terms. To protect the privacy of the participants, data access was restricted to the investigators of this study, as defined in those data access agreements. Per data use agreements, only summary level data is made publicly available and strictly mentioned in the disclaimer that they cannot be used to re-identify study participants.

## Reporting summary

Further information on research design is available in the Nature Portfolio Reporting Summary linked to this article.

## Data availability

Our study is comprised of previously published human brain eQTL datasets. The majority of these datasets are available on request or through online repositories after signing data access agreements. Summary statistics for the performed *cis*-eQTL analyses are available through the MetaBrain website (https://www.metabrain.nl), after registering name, institute and e-mail address. The mode of access for each of the included datasets was as follows. TargetALS[12] data were pushed directly from the New York Genome center (https://www.targetals.org/) to our SFTP server. CMC[99] (accession code: syn2759792), CMC_HBCC (accession code: syn10623034), AMP-AD[5] (accession code: syn2580853; the snRNA-seq data were collected using the Synapse accession code syn18485175 and the IHC data were from https://github.com/ellispatrick/CortexCellDeconv/tree/master/CellTypeDeconvAnalysis/Data), BrainSeq (accession code: syn12299750), UCLA_ASD data (accession code: syn4587609), BrainGVEx (accession code: syn4590909) and BipSeq (accession code: syn5844980) data were downloaded from Synapse (https://www.synapse.org/) using synapseclient (https://python-docs.synapse.org/build/html/index.html). GTEx[69] was downloaded from the Sequence Read Archive (SRA) using fastq-dump of the SRA toolkit (http://www.ncbi.nlm.nih.gov/Traces/sra/sra.cgi?cmd=show&f=software&m=software&s=software). Access was requested and granted through dbGaP (accession code: phs000424.v7.p2). Braineac[7] data were pushed to our SFTP server by Biogen. The identifiers of the 76 included studies and 2,021 brain samples downloaded from the ENA[13] are listed in Supplementary Table 31. NABEC data (accession code: phs001301.v1.p1) were downloaded from dbgap. The Bryois et al. eQTL summary statistics were downloaded from Zenodo in January 2022 from https://doi.org/10.5281/zenodo.5543734. Other databases or datasets that we have used: SkyMap[100]. Source data are provided with this paper.

## Code availability

Code is available at Zenodo (https://doi.org/10.5281/zenodo.7376855)[101]. The eQTL mapping software (*cis*) is available at https://github.com/molgenis/systemsgenetics/tree/master/mbQTL. The eQTL mapping software (*trans*) is available at https://github.com/molgenis/systemsgenetics/tree/master/eqtl-mapping-pipeline/. Other custom scripts used in this manuscript are available at https://github.com/molgenis/metabrain. The allelic fold change code is available at https://github.com/secastel/aFC.

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

## Acknowledgements

We thank the donors of the brain tissues underlying the RNA-seq data used for this study and their families for their willingness to donate samples for research. We also thank all researchers involved with the included cohorts for making their data available for use. We have acknowledged each of the included cohorts and datasets in the Supplementary Note. We thank the Center for Information Technology of the University of Groningen for their support and for providing access to the Peregrine high-performance computing cluster, as well as the UMCG Genomics Coordination Center, the UG Center for Information Technology and their sponsors BBMRI-NL and TarGet for storage and computing infrastructure. D.B. and T.R.G. are supported by funding from the UK Medical Research Council (MRC Integrative Epidemiology Unit at the University of Bristol, grant no. MC_UU_00011/4) and a sponsored research collaboration with Biogen. L.F. is supported by a grant from the Dutch Research Council (grant no. ZonMW-VICI 09150182010019), an ERC Starting Grant (grant agreement 637640 (ImmRisk)), an Oncode Senior Investigator grant and a sponsored research collaboration with Biogen. This project has received funding from the European Research Council (ERC) under the European Union's Horizon 2020 research and innovation program (grant agreement no. 772376–EScORIAL to J.V.). The authors thank the Biogen CellMap team (Z. Ouyang, N. Bourgeois, E. Lyashenko, P. Cundiff, K. Li, X. Zhang, F. Casey, S. Engle, R. Kleiman, B. Zhang and M. Zavodszky) for the advice they provided towards deriving the cell type-specific expression profiles.

## Author contributions

N.d.K., O.E.G. and H.-J.W. processed the RNA-seq and genotype data. N.d.K. and H.-J.W. were responsible for data management as well as for the *cis*-eQTL analysis. H.-J.W. was responsible for the *trans*-eQTL analysis. M.V. was responsible for, with assistance from Z.O. and M.I.Z., the cell-type proportion prediction. M.V. and R.O. were responsible for the ROSMAP single-nucleus data processing. N.d.K. and M.V. were responsible for the cell-type interaction analysis. S.v.D. was responsible for the selection of brain samples from the ENA. D.B., Y.H., C.-Y.C., E.E.M., T.R.G. and E.A.T. were responsible for the MR and co-localization analysis and interpretation. J.V., M.K.B. and W.v.R. supplied ALS GWAS summary statistics and aided in the interpretation of ALS-related results. P.D., O.B.B. and L.F. were responsible for the Downstreamer analysis. L.F., E.A.T. and H.R. acquired funding and supervised the study. N.d.K., E.A.T., M.V., D.B., Y.H., C.-Y.C., O.B.B., H.R., L.F. and H.-J.W. drafted the manuscript. All authors have proof-read the manuscript.

## Competing interests

E.A.T., D.B., Y.H., C.-Y.C., Z.O., E.E.M., M.I.Z. and H.R. are full-time employees and hold stock options at Biogen. The remaining authors declare no competing interests.

## Additional information

**Correspondence and requests for materials** should be addressed to Heiko Runz, Lude Franke or Harm-Jan Westra.

# Reporting Summary

## Statistics

For all statistical analyses, confirm that the following items are present in the figure legend, table legend, main text, or Methods section.

| n/a | Confirmed | |
|---|---|---|
| ☐ | ☒ | The exact sample size (*n*) for each experimental group/condition, given as a discrete number and unit of measurement |
| ☐ | ☒ | A statement on whether measurements were taken from distinct samples or whether the same sample was measured repeatedly |
| ☐ | ☒ | The statistical test(s) used AND whether they are one- or two-sided<br>*Only common tests should be described solely by name; describe more complex techniques in the Methods section.* |
| ☐ | ☒ | A description of all covariates tested |
| ☐ | ☒ | A description of any assumptions or corrections, such as tests of normality and adjustment for multiple comparisons |
| ☐ | ☒ | A full description of the statistical parameters including central tendency (e.g. means) or other basic estimates (e.g. regression coefficient) AND variation (e.g. standard deviation) or associated estimates of uncertainty (e.g. confidence intervals) |
| ☐ | ☒ | For null hypothesis testing, the test statistic (e.g. *F*, *t*, *r*) with confidence intervals, effect sizes, degrees of freedom and *P* value noted<br>*Give P values as exact values whenever suitable.* |
| ☐ | ☒ | For Bayesian analysis, information on the choice of priors and Markov chain Monte Carlo settings |
| ☒ | ☐ | For hierarchical and complex designs, identification of the appropriate level for tests and full reporting of outcomes |
| ☐ | ☒ | Estimates of effect sizes (e.g. Cohen's *d*, Pearson's *r*), indicating how they were calculated |

*Our web collection on statistics for biologists contains articles on many of the points above.*

## Software and code

Policy information about availability of computer code

| Data collection | A number of datasets (AMP-AD datasets, CMC, CMC_HBCC, BrainSeq, UCLA_ASD, BrainGVEx, BipSeq) were collected from Synapse.org using the Pythong synapseclient (version 1.9.2) packakge. Other datasets (GTEx, NABEC) were acquired from dbGAP using NCBI's SRA-tools. Two datasets (TargetALS and Braineac) were made available through personal communication with investigators of those studies. |
|---|---|
| Data analysis | Alignment for eQTL analysis was performed using STAR v2.6.1c and TMM normalization was performed using edgeR v3.20.9. Alignment for creation of gene networks was performed using Kallisto v0.43.1. RNA-seq genotype calling was performed using GATK v4.0.9.1. RNA-seq QC was performed using FastQC v0.11.3, STAR metrics, and Picard tools v2.18.26. Genotype processing was performed using PLINK v2.0 and bcftools v1.7. Where applicable genotypes were harmonized using GenotypeHarmonizer (https://github.com/molgenis/systemsgenetics/tree/master/Genotype-Harmonizer), and imputed to HRC v1.1 using the Michigan imputation server (https://imputationserver.sph.umich.edu/index.html). Additional custom scripts are available at https://github.com/molgenis/metabrain. This repository includes code for prediction of cell type proportions using the NNLS implementation of SciPy v1.4.1. eQTL analysis software (v1.4.9) is available at https://github.com/molgenis/systemsgenetics/tree/master/eqtl-mapping-pipeline. and https://github.com/molgenis/systemsgenetics/tree/master/mbQTL. Mendelian randomization was performerd using the ieugwasr and TwoSampleMR R packages. Colocalization was performed using COLOC. Downstreamer is available at https://github.com/molgenis/systemsgenetics/tree/master/Downstreamer. Allelic fold-change analysis (aFC was performed using the aFC.py script in this github repository: https://github.com/secastel/aFC/. Single cell data was analyzed using Seurat v3.2.2119. |

For manuscripts utilizing custom algorithms or software that are central to the research but not yet described in published literature, software must be made available to editors and reviewers. We strongly encourage code deposition in a community repository (e.g. GitHub). See the Nature Portfolio guidelines for submitting code & software for further information.

# Data

Policy information about availability of data

All manuscripts must include a data availability statement. This statement should provide the following information, where applicable:

- Accession codes, unique identifiers, or web links for publicly available datasets
- A description of any restrictions on data availability
- For clinical datasets or third party data, please ensure that the statement adheres to our policy

Our study is comprised of previously published human brain eQTL datasets. The majority of these datasets are available upon request, or through online repositories after signing data acces agreements. We have listed the mode of access of each of the included datasets below. Summary statistics for the performed eQTL analyses will be available through https://www.metabrain.nl. TargetALS data was pushed directly from the NY Genome center to our sftp server.
CMC data was downloaded from https://www.synapse.org/ using synapse client (https://python-docs.synapse.org/build/html/index.html). Accession code: syn2759792
GTEx was downloaded from SRA using fastq-dump of the SRA toolkit (http://www.ncbi.nlm.nih.gov/Traces/sra/sra.cgi?cmd=show&f=software&m=software&s=software). Access has been requested and granted through dbGaP, accession code: phs000424.v7.p2.
Braineac data has been pushed to our sftp server by Biogen.
AMP-AD data has been downloaded from synapse13. Accession code: syn2580853. snRNA-seq was collected using Synapse accession code: syn18485175. IHC data: https://github.com/ellispatrick/CortexCellDeconv/tree/master/CellTypeDeconvAnalysis/Data
ENA data has been downloaded from the European Nucleotide Archive. The identifiers of the 76 included studies and 2021 brain samples are listed in Supplementary Table 31.
CMC_HBCC data was downloaded from https://www.synapse.org/ using synapse client (https://python-docs.synapse.org/build/html/index.html). Accession code: syn10623034.
BrainSeq data was downloaded from https://www.synapse.org/ using synapse client (https://python-docs.synapse.org/build/html/index.html). Accession code: syn12299750
UCLA_ASD data was downloaded from https://www.synapse.org/ using synapse client (https://python-docs.synapse.org/build/html/index.html). Accession code: syn4587609
BrainGVEx data was downloaded from https://www.synapse.org/ using synapse client (https://python-docs.synapse.org/build/html/index.html). Accession code: syn4590909
BipSeq data was downloaded from https://www.synapse.org/ using synapse client (https://python-docs.synapse.org/build/html/index.html). Accession code: syn5844980
NABEC data was downloaded from dbgap. Accession code: phs001301.v1.p1
Bryois et al. 2022. Bryois et al. 2022 eQTL summary statistics where downloaded from Zenodo on January 2022 from https://doi.org/10.5281/zenodo.5543734.
Other databaes/datasets that we have used: SkyMap (https://github.com/brianyiktaktsui/Skymap).

# Human research participants

Policy information about studies involving human research participants and Sex and Gender in Research.

| | |
|---|---|
| Reporting on sex and gender | We do not report on sex or gender. |
| Population characteristics | Our study includes 13 previously published brain eQTL datasets, and one dataset created from publicly available RNA-seq samples. In toal, we collected 8,613 RNA-seq samples and 6,518 genotype samples. Depending on the dataset, different clinical covariates were either available or unavailable, including age, gender and ancestry. Depending on the dataset, disease status was available (Alzheimer's disease for AMP-AD, several psychiatric diseases for Psychencode datasets, amyotrophic lateral sclerosis for TargetALS). Due to the high missingness within these clinical covariates, we did not explicitly correct for these covariates, but used principal components, which capture variation in gene expression caused by these covariates |
| Recruitment | Participants were not recruited by the authors of this study. |
| Ethics oversight | All included datasets recruited participants with informed consent, and data was collected in accordance with ethical and institutional regulations, according to their institutional review boards (IRBs). All data was anlayzed in compliance with signed data access agreements, which have been reviewed by the IRBs or data access committees of each of the included datasets. |

Note that full information on the approval of the study protocol must also be provided in the manuscript.

# Field-specific reporting

Please select the one below that is the best fit for your research. If you are not sure, read the appropriate sections before making your selection.

☒ Life sciences       ☐ Behavioural & social sciences       ☐ Ecological, evolutionary & environmental sciences

For a reference copy of the document with all sections, see nature.com/documents/nr-reporting-summary-flat.pdf

# Life sciences study design

All studies must disclose on these points even when the disclosure is negative.

| | |
|---|---|
| Sample size | Our study includes 13 previously published brain eQTL datasets, and one dataset created from publicly available RNA-seq samples. In total, we collected 8,613 RNA-seq samples and 6,518 genotype samples. We did not calculate sample size, instead we aimed to include all (at that time) available bulk brain RNA-seq+genotype samples. eQTL analysis was performed in distinct tissue groups: Basal ganglia-EUR (n=208), Cerebellum-EUR (n=492), Cortex-EUR (n=2,683), Cortex-AFR (n=319), Hippocampus-EUR (n=168) and Spinal cord-EUR (n=108). For each of these analyses, we required that each tested eQTL was present in at least 2 cohorts. As eQTL analysis was performed previously on each of the individual cohorts separately, we reasoned that this would provide sufficient statistical power for our analysis. |
| Data exclusions | A number of samples were excluded during the quality control steps outlined in the methods section of our manuscript (i.e. outliers in terms of alignment metrics, or outliers identified through principal component analysis). For eQTL analysis, if an individual had multiple RNA-seq measurements for the same tissue, we randomly selected a single sample for inclusion, and excluded the others. Similarly, related genotype samples were excluded from analysis. Some samples, such as those for the amygdala and hypothalamus tissues, or samples from SAS or AMR datasets ancestry were excluded from eQTL analysis because their group sizes were deemed too small (n<30), or there were less than 2 available datsets for that particular group. Group definitions and sample inclusion per group are listed in Supplementary Table 1. |
| Replication | All tested eQTLs were required to be present in at least 2 of the included datasets. Cis-eQTLs were replicated in GTEx tissues by excluding the GTEx cohort from the discovery analysis. We osberved high concordance in allelic direction, correlation of genetic effect sizes, and pi1 replication rates in cortical tissues. Interaction eQTLs detected in Cortex-EUR were replicated in the Cortex-AFR dataset, a single nucleus RNA-seq dataset and using summary statistics of an independent single nucleus RNA-seq study. Trans-eQTLs were replicated in a single nucleus RNA-seq dataset. We observed high agreement for both these types of eQTLs. |
| Randomization | RNA-seq samples were assigned to different groups according to (predicted) brain region. The datset consisted of many different tissues from similar brain regions. These tissues were merged into larger brain tissue groups (i.e. amygdala, basal ganglia, cortex, cerebellum, hypothalamus, hippocampus, spinalcord). Tissue grouping is defined in Supplementary Table 1. Genotype samples were grouped according to ancestry, including African (AFR), European (EUR), East Asian (EAS), South Asian (SAS), and admixed American (AMR). To define eQTL datasets, tissue and ancestry groups were overlapped, but analysis was limited to groups having samples from two or more datasets. This created Basal ganglia-EUR, Cerebellum-EUR, Cortex-EUR, Cortex-AFR, Hippocampsu-EUR, and Spinalcord-EUR eQTL datasets. For trans-eQTL analysis an additional dataset was defined, by merging Cortex-EUR and Cortex-AFR, but excluding the ENA dataset. To correct for dataset or batch effects we included several covariates, including dataset indicator variables, principal components, several RNA-seq alignment QC metrics, and 4 MDS components derived from the genotype data to further account for potential population stratification. eQTL analysis tests associations etween genotype (groups) and gene expression levels. Assignment of each individual to a genotype group for a given random can be considered randomized when accounting for the mentioned covariates. |
| Blinding | Since our study uses previously published datasets, the authors were not involved in group allocation during data collection. Furthemore, this study does not represent a clinical trial or an interventional study, so blinding was not required. As described from above, assignment of an individual to a genotype group of a given variant can be considered random. |

# Behavioural & social sciences study design

All studies must disclose on these points even when the disclosure is negative.

| | |
|---|---|
| Study description | *Briefly describe the study type including whether data are quantitative, qualitative, or mixed-methods (e.g. qualitative cross-sectional, quantitative experimental, mixed-methods case study).* |
| Research sample | *State the research sample (e.g. Harvard university undergraduates, villagers in rural India) and provide relevant demographic information (e.g. age, sex) and indicate whether the sample is representative. Provide a rationale for the study sample chosen. For studies involving existing datasets, please describe the dataset and source.* |
| Sampling strategy | *Describe the sampling procedure (e.g. random, snowball, stratified, convenience). Describe the statistical methods that were used to predetermine sample size OR if no sample-size calculation was performed, describe how sample sizes were chosen and provide a rationale for why these sample sizes are sufficient. For qualitative data, please indicate whether data saturation was considered, and what criteria were used to decide that no further sampling was needed.* |
| Data collection | *Provide details about the data collection procedure, including the instruments or devices used to record the data (e.g. pen and paper, computer, eye tracker, video or audio equipment) whether anyone was present besides the participant(s) and the researcher, and whether the researcher was blind to experimental condition and/or the study hypothesis during data collection.* |
| Timing | *Indicate the start and stop dates of data collection. If there is a gap between collection periods, state the dates for each sample cohort.* |
| Data exclusions | *If no data were excluded from the analyses, state so OR if data were excluded, provide the exact number of exclusions and the rationale behind them, indicating whether exclusion criteria were pre-established.* |
| Non-participation | *State how many participants dropped out/declined participation and the reason(s) given OR provide response rate OR state that no participants dropped out/declined participation.* |

| Randomization | *If participants were not allocated into experimental groups, state so OR describe how participants were allocated to groups, and if allocation was not random, describe how covariates were controlled.* |

# Ecological, evolutionary & environmental sciences study design

All studies must disclose on these points even when the disclosure is negative.

| Study description | *Briefly describe the study. For quantitative data include treatment factors and interactions, design structure (e.g. factorial, nested, hierarchical), nature and number of experimental units and replicates.* |
| Research sample | *Describe the research sample (e.g. a group of tagged Passer domesticus, all Stenocereus thurberi within Organ Pipe Cactus National Monument), and provide a rationale for the sample choice. When relevant, describe the organism taxa, source, sex, age range and any manipulations. State what population the sample is meant to represent when applicable. For studies involving existing datasets, describe the data and its source.* |
| Sampling strategy | *Note the sampling procedure. Describe the statistical methods that were used to predetermine sample size OR if no sample-size calculation was performed, describe how sample sizes were chosen and provide a rationale for why these sample sizes are sufficient.* |
| Data collection | *Describe the data collection procedure, including who recorded the data and how.* |
| Timing and spatial scale | *Indicate the start and stop dates of data collection, noting the frequency and periodicity of sampling and providing a rationale for these choices. If there is a gap between collection periods, state the dates for each sample cohort. Specify the spatial scale from which the data are taken* |
| Data exclusions | *If no data were excluded from the analyses, state so OR if data were excluded, describe the exclusions and the rationale behind them, indicating whether exclusion criteria were pre-established.* |
| Reproducibility | *Describe the measures taken to verify the reproducibility of experimental findings. For each experiment, note whether any attempts to repeat the experiment failed OR state that all attempts to repeat the experiment were successful.* |
| Randomization | *Describe how samples/organisms/participants were allocated into groups. If allocation was not random, describe how covariates were controlled. If this is not relevant to your study, explain why.* |
| Blinding | *Describe the extent of blinding used during data acquisition and analysis. If blinding was not possible, describe why OR explain why blinding was not relevant to your study.* |

Did the study involve field work? ☐ Yes ☐ No

## Field work, collection and transport

| Field conditions | *Describe the study conditions for field work, providing relevant parameters (e.g. temperature, rainfall).* |
| Location | *State the location of the sampling or experiment, providing relevant parameters (e.g. latitude and longitude, elevation, water depth).* |
| Access & import/export | *Describe the efforts you have made to access habitats and to collect and import/export your samples in a responsible manner and in compliance with local, national and international laws, noting any permits that were obtained (give the name of the issuing authority, the date of issue, and any identifying information).* |
| Disturbance | *Describe any disturbance caused by the study and how it was minimized.* |

# Reporting for specific materials, systems and methods

We require information from authors about some types of materials, experimental systems and methods used in many studies. Here, indicate whether each material, system or method listed is relevant to your study. If you are not sure if a list item applies to your research, read the appropriate section before selecting a response.

## Materials & experimental systems

| n/a | Involved in the study |
|---|---|
| ☒ ☐ | Antibodies |
| ☒ ☐ | Eukaryotic cell lines |
| ☒ ☐ | Palaeontology and archaeology |
| ☒ ☐ | Animals and other organisms |
| ☒ ☐ | Clinical data |
| ☒ ☐ | Dual use research of concern |

## Methods

| n/a | Involved in the study |
|---|---|
| ☒ ☐ | ChIP-seq |
| ☒ ☐ | Flow cytometry |
| ☐ ☒ | MRI-based neuroimaging |

# Antibodies

Antibodies used — *Describe all antibodies used in the study; as applicable, provide supplier name, catalog number, clone name, and lot number.*

Validation — *Describe the validation of each primary antibody for the species and application, noting any validation statements on the manufacturer's website, relevant citations, antibody profiles in online databases, or data provided in the manuscript.*

# Eukaryotic cell lines

Policy information about cell lines and Sex and Gender in Research

Cell line source(s) — *State the source of each cell line used and the sex of all primary cell lines and cells derived from human participants or vertebrate models.*

Authentication — *Describe the authentication procedures for each cell line used OR declare that none of the cell lines used were authenticated.*

Mycoplasma contamination — *Confirm that all cell lines tested negative for mycoplasma contamination OR describe the results of the testing for mycoplasma contamination OR declare that the cell lines were not tested for mycoplasma contamination.*

Commonly misidentified lines (See ICLAC register) — *Name any commonly misidentified cell lines used in the study and provide a rationale for their use.*

# Palaeontology and Archaeology

Specimen provenance — *Provide provenance information for specimens and describe permits that were obtained for the work (including the name of the issuing authority, the date of issue, and any identifying information). Permits should encompass collection and, where applicable, export.*

Specimen deposition — *Indicate where the specimens have been deposited to permit free access by other researchers.*

Dating methods — *If new dates are provided, describe how they were obtained (e.g. collection, storage, sample pretreatment and measurement), where they were obtained (i.e. lab name), the calibration program and the protocol for quality assurance OR state that no new dates are provided.*

☐ Tick this box to confirm that the raw and calibrated dates are available in the paper or in Supplementary Information.

Ethics oversight — *Identify the organization(s) that approved or provided guidance on the study protocol, OR state that no ethical approval or guidance was required and explain why not.*

Note that full information on the approval of the study protocol must also be provided in the manuscript.

# Animals and other research organisms

Policy information about studies involving animals; ARRIVE guidelines recommended for reporting animal research, and Sex and Gender in Research

Laboratory animals — *For laboratory animals, report species, strain and age OR state that the study did not involve laboratory animals.*

Wild animals — *Provide details on animals observed in or captured in the field; report species and age where possible. Describe how animals were caught and transported and what happened to captive animals after the study (if killed, explain why and describe method; if released, say where and when) OR state that the study did not involve wild animals.*

Reporting on sex — *Indicate if findings apply to only one sex; describe whether sex was considered in study design, methods used for assigning sex. Provide data disaggregated for sex where this information has been collected in the source data as appropriate; provide overall numbers in this Reporting Summary. Please state if this information has not been collected. Report sex-based analyses where performed, justify reasons for lack of sex-based analysis.*

| Field-collected samples | *For laboratory work with field-collected samples, describe all relevant parameters such as housing, maintenance, temperature, photoperiod and end-of-experiment protocol OR state that the study did not involve samples collected from the field.* |
|---|---|
| Ethics oversight | *Identify the organization(s) that approved or provided guidance on the study protocol, OR state that no ethical approval or guidance was required and explain why not.* |

Note that full information on the approval of the study protocol must also be provided in the manuscript.

# Clinical data

Policy information about clinical studies

All manuscripts should comply with the ICMJE guidelines for publication of clinical research and a completed CONSORT checklist must be included with all submissions.

| Clinical trial registration | *Provide the trial registration number from ClinicalTrials.gov or an equivalent agency.* |
|---|---|
| Study protocol | *Note where the full trial protocol can be accessed OR if not available, explain why.* |
| Data collection | *Describe the settings and locales of data collection, noting the time periods of recruitment and data collection.* |
| Outcomes | *Describe how you pre-defined primary and secondary outcome measures and how you assessed these measures.* |

# Dual use research of concern

Policy information about dual use research of concern

## Hazards

Could the accidental, deliberate or reckless misuse of agents or technologies generated in the work, or the application of information presented in the manuscript, pose a threat to:

No | Yes
- ☐ ☐ Public health
- ☐ ☐ National security
- ☐ ☐ Crops and/or livestock
- ☐ ☐ Ecosystems
- ☐ ☐ Any other significant area

## Experiments of concern

Does the work involve any of these experiments of concern:

No | Yes
- ☐ ☐ Demonstrate how to render a vaccine ineffective
- ☐ ☐ Confer resistance to therapeutically useful antibiotics or antiviral agents
- ☐ ☐ Enhance the virulence of a pathogen or render a nonpathogen virulent
- ☐ ☐ Increase transmissibility of a pathogen
- ☐ ☐ Alter the host range of a pathogen
- ☐ ☐ Enable evasion of diagnostic/detection modalities
- ☐ ☐ Enable the weaponization of a biological agent or toxin
- ☐ ☐ Any other potentially harmful combination of experiments and agents

# ChIP-seq

## Data deposition

☐ Confirm that both raw and final processed data have been deposited in a public database such as GEO.

☐ Confirm that you have deposited or provided access to graph files (e.g. BED files) for the called peaks.

| Data access links<br>*May remain private before publication.* | *For "Initial submission" or "Revised version" documents, provide reviewer access links.  For your "Final submission" document, provide a link to the deposited data.* |
|---|---|
| Files in database submission | *Provide a list of all files available in the database submission.* |

Genome browser session
(e.g. UCSC)

*Provide a link to an anonymized genome browser session for "Initial submission" and "Revised version" documents only, to enable peer review. Write "no longer applicable" for "Final submission" documents.*

## Methodology

| | |
|---|---|
| Replicates | *Describe the experimental replicates, specifying number, type and replicate agreement.* |
| Sequencing depth | *Describe the sequencing depth for each experiment, providing the total number of reads, uniquely mapped reads, length of reads and whether they were paired- or single-end.* |
| Antibodies | *Describe the antibodies used for the ChIP-seq experiments; as applicable, provide supplier name, catalog number, clone name, and lot number.* |
| Peak calling parameters | *Specify the command line program and parameters used for read mapping and peak calling, including the ChIP, control and index files used.* |
| Data quality | *Describe the methods used to ensure data quality in full detail, including how many peaks are at FDR 5% and above 5-fold enrichment.* |
| Software | *Describe the software used to collect and analyze the ChIP-seq data. For custom code that has been deposited into a community repository, provide accession details.* |

# Flow Cytometry

## Plots

Confirm that:

☐ The axis labels state the marker and fluorochrome used (e.g. CD4-FITC).

☐ The axis scales are clearly visible. Include numbers along axes only for bottom left plot of group (a 'group' is an analysis of identical markers).

☐ All plots are contour plots with outliers or pseudocolor plots.

☐ A numerical value for number of cells or percentage (with statistics) is provided.

## Methodology

| | |
|---|---|
| Sample preparation | *Describe the sample preparation, detailing the biological source of the cells and any tissue processing steps used.* |
| Instrument | *Identify the instrument used for data collection, specifying make and model number.* |
| Software | *Describe the software used to collect and analyze the flow cytometry data. For custom code that has been deposited into a community repository, provide accession details.* |
| Cell population abundance | *Describe the abundance of the relevant cell populations within post-sort fractions, providing details on the purity of the samples and how it was determined.* |
| Gating strategy | *Describe the gating strategy used for all relevant experiments, specifying the preliminary FSC/SSC gates of the starting cell population, indicating where boundaries between "positive" and "negative" staining cell populations are defined.* |

☐ Tick this box to confirm that a figure exemplifying the gating strategy is provided in the Supplementary Information.

# Magnetic resonance imaging

## Experimental design

| | |
|---|---|
| Design type | *Indicate task or resting state; event-related or block design.* |
| Design specifications | *Specify the number of blocks, trials or experimental units per session and/or subject, and specify the length of each trial or block (if trials are blocked) and interval between trials.* |
| Behavioral performance measures | *State number and/or type of variables recorded (e.g. correct button press, response time) and what statistics were used to establish that the subjects were performing the task as expected (e.g. mean, range, and/or standard deviation across subjects).* |

## Acquisition

| | |
|---|---|
| Imaging type(s) | *Specify: functional, structural, diffusion, perfusion.* |
| Field strength | *Specify in Tesla* |
| Sequence & imaging parameters | *Specify the pulse sequence type (gradient echo, spin echo, etc.), imaging type (EPI, spiral, etc.), field of view, matrix size, slice thickness, orientation and TE/TR/flip angle.* |
| Area of acquisition | *State whether a whole brain scan was used OR define the area of acquisition, describing how the region was determined.* |

Diffusion MRI ☐ Used ☐ Not used

## Preprocessing

| | |
|---|---|
| Preprocessing software | *Provide detail on software version and revision number and on specific parameters (model/functions, brain extraction, segmentation, smoothing kernel size, etc.).* |
| Normalization | *If data were normalized/standardized, describe the approach(es): specify linear or non-linear and define image types used for transformation OR indicate that data were not normalized and explain rationale for lack of normalization.* |
| Normalization template | *Describe the template used for normalization/transformation, specifying subject space or group standardized space (e.g. original Talairach, MNI305, ICBM152) OR indicate that the data were not normalized.* |
| Noise and artifact removal | *Describe your procedure(s) for artifact and structured noise removal, specifying motion parameters, tissue signals and physiological signals (heart rate, respiration).* |
| Volume censoring | *Define your software and/or method and criteria for volume censoring, and state the extent of such censoring.* |

## Statistical modeling & inference

| | |
|---|---|
| Model type and settings | *Specify type (mass univariate, multivariate, RSA, predictive, etc.) and describe essential details of the model at the first and second levels (e.g. fixed, random or mixed effects; drift or auto-correlation).* |
| Effect(s) tested | *Define precise effect in terms of the task or stimulus conditions instead of psychological concepts and indicate whether ANOVA or factorial designs were used.* |

Specify type of analysis: ☐ Whole brain ☐ ROI-based ☐ Both

| | |
|---|---|
| Statistic type for inference<br>(See Eklund et al. 2016) | *Specify voxel-wise or cluster-wise and report all relevant parameters for cluster-wise methods.* |
| Correction | *Describe the type of correction and how it is obtained for multiple comparisons (e.g. FWE, FDR, permutation or Monte Carlo).* |

## Models & analysis

| n/a | Involved in the study | |
|---|---|---|
| ☐ | ☐ | Functional and/or effective connectivity |
| ☐ | ☐ | Graph analysis |
| ☐ | ☐ | Multivariate modeling or predictive analysis |

| | |
|---|---|
| Functional and/or effective connectivity | *Report the measures of dependence used and the model details (e.g. Pearson correlation, partial correlation, mutual information).* |
| Graph analysis | *Report the dependent variable and connectivity measure, specifying weighted graph or binarized graph, subject- or group-level, and the global and/or node summaries used (e.g. clustering coefficient, efficiency, etc.).* |
| Multivariate modeling and predictive analysis | *Specify independent variables, features extraction and dimension reduction, model, training and evaluation metrics.* |

