## [Peer Review File · Nature Genetics]

Peer Review Information

Manuscript Title: Brain expression quantitative trait locus and network analysis reveals downstream effects and putative drivers for brain-related diseases

Corresponding author name(s): Dr Harm-Jan Westra, Dr Heiko Runz, and Professor Lude Franke

Reviewer Comments & Decisions:

Decision Letter, initial version:
--

12th May 2021

Dear Dr Westra,

Your Article, "Brain expression quantitative trait locus and network analysis reveals downstream effects and putative drivers for brain-related diseases" has now been seen by 2 referees. You will see from their comments copied below that while they find your work of considerable potential interest, they have raised quite substantial concerns that must be addressed. In light of these comments, we cannot accept the manuscript for publication, but would be very interested in considering a revised version that addresses these serious concerns.

We hope you will find the referees' comments useful as you decide how to proceed. If you wish to submit a substantially revised manuscript, please bear in mind that we will be reluctant to approach the referees again in the absence of major revisions.

As you will see from these comments, both reviewers have identified aspects of the analyses and the presentation that should be improved. In particular, reviewer #1 has serious concerns regarding the approaches used in this study, the robustness or comprehensiveness of the analyses and results. Please address all referees' points as thoroughly as you can.

If you choose to revise your manuscript taking into account all reviewer and editor comments, please highlight all changes in the manuscript text file. At this stage we will need you to upload a copy of the manuscript in MS Word .docx or similar editable format.

*2) If you have not done so already please begin to revise your manuscript so that it conforms to our Article format instructions, available [here](http://www.nature.com/ng/authors/article_types/index.html). Refer also to any guidelines provided in this letter.

[removed]

If you wish to submit a suitably revised manuscript we would hope to receive it within 6 months. If you cannot send it within this time, please let us know. We will be happy to consider your revision so long as nothing similar has been accepted for publication at Nature Genetics or published elsewhere. Should your manuscript be substantially delayed without notifying us in advance and your article is eventually published, the received date would be that of the revised, not the original, version.

Thank you for the opportunity to review your work.

Sincerely,

Wei

Wei Li, PhD
Senior Editor
Nature Genetics
One New York Plaza, 47th Fl.
New York, NY 10004, USA
www.nature.com/ng

Reviewers' Comments:

Reviewer #1:

Remarks to the Author:

Brain expression quantitative trait locus and network analysis reveals downstream effects and putative drivers for brain-related diseases

De Klein et al performed meta-analysis of 14 published human brain gene expression cohorts to identify cis and trans eQTLs active across 7 major tissue groups in human brain (amygdala, basal ganglia, cerebellum, cortex, hippocampus, hypothalamus, and spinal cord). They perform a traditional set of cis/trans-QTL analyses and disease GWAS enrichments (e.g., MR and coloc) to prioritize candidate risk genes and mechanisms underlying complex brain-relevant traits. They perform comparisons of QTL architecture across brain regions, ancestries, and tissues, as well as generate co-expression networks in brain. Applications to neurological traits like MS are described. Overall, the group has substantial expertise with these types of analyses which is why I find it all the more puzzling why they use several less-standard approaches that give results which in many cases are quite divergent from those in the published literature (e.g., much higher concordance rates, see below). Furthermore, despite the size of this paper and the extremely lengthy supplement, in many cases I don't find explicit comparisons to these more widely used methods and results, which make it all the more difficult to evaluate. As such, aside from the size of the dataset, I don't find the advances presented here to be substantial either methodologically or analytically. Furthermore and perhaps more concerning, I am surprised by the lack of clarity, robustness, or comprehensiveness of the analyses and results described from a group that is otherwise positioned to understand the current state-of-the-art pipelines for functional genomics. The webportal is well done and much appreciated.

1. Cis-QTL mapping -- The authors identify ~12k cis-eQTL in cortex-EUR, which is the most well powered group in their analysis. Despite the large sample size, this is actually substantially smaller I believe compared with results from previously published studies with smaller sample sizes (e.g., CommonMind, BrainSeq, PsychENCODE) at similar FDR thresholds. This is very likely due to analytic differences in how datasets are processed and how QTLs are calculated. This should be explicitly addressed. This manuscript uses a genome-wide permutation approach ($n=10$) to generate a null distribution. While it is claimed that this converges/stabilizes faster than locus-specific approaches,

this is not standard in the field and should at least be compared with the beta approximated null approach as implemented in FastQTL/QTLtools and used by groups like GTEx.

2. Concordance -- The assessment of cis-eQTL concordance rates across regions/tissues/ancestries does not pass the 'smell' test. I've never seen cis-eQTL concordance rates that high (95-100%), which is very likely due to the non-standard method used (correlation of Z-score sign). At a minimum, this needs to be compared to much more standard methods like Pi-hat. Likewise, for comparison of effects across ancestries, there needs to be consideration for both standardized and allelic effects, such as implemented in methods like 'popcorn'. That cisQTLs from EAS-Cortex with a sample size of ~200 show >95% concordance with Cortex EUR with a sample size of ~3000 in both directions seems to undercut the entire premise of this study that 'bigger is better' for discovering gene regulatory effects.

3. Cell-type interacting QTLs -- The cell-type specificity or predominance of QTLs is of interest to the field. However, the implementation in this manuscript is quite bizarre. They use a supervised method (NNLS) with gene expression signatures derived from the CellMap project -- an unpublished resource that can therefore not be fully evaluated. There are now literally dozens of well powered human brain sc/sn-RNAseq datasets defining cell-type-specific gene expression signatures that could have been used (many of which are actually cited in this paper) but were not for unknown reasons. Furthermore, the same signatures were used across brain regions that are known to harbor VERY different cell types -- For example, the cerebellum is comprised of very distinct cell types (Purkinje cells, dentate granule cells, etc) that are not present in cortex and were not included in the reference panel. The proportions estimated are not physiologically reliable -- e.g., the number of astrocytes are much smaller than they should be, the number of macrophages are much larger than they should be (even if substituting microglia for macrophage). Other brain regions here also have very different cell-types (e.g, basal ganglia -- medium spiny neurons; hippocampus has CA1/CA2/CA3 pyramidal cells, etc...). As such, I don't consider any of the cell type analyses to be particularly reliable. There doesn't appear to be much consideration for excitatory vs interneuron neuron effects even though these are very distinct lineages. Finally, the proportions as estimated add up to 1 and therefore have one less degree of freedom than number of cell-types assessed. It doesn't appear that this is accounted for in the deconvolution method, and if so, this should be explicitly addressed in the paper as a limitation.

4. Trans-QTLs. The trans-QTL analysis also does not pass the 'smell' test. That ~80% of trans-QTLs are derived from the single 7p21 locus is VERY concerning for an underlying confound or hidden bias in the data. I think this most likely represents a cell proportion shift -- it seems that all the trans-eGenes are very specific to neuronal (rather than glial) lineages. As such, any SNP that is associated with even slight loss of neurons would then be confoundingly associated with top marker genes of that cell-type, irrespective of whether there was any actual genomic regulatory relationship. Given that a large number of the samples included here come from individuals with a neurodegenerative disorder like Alzheimer's, and these samples do not appear to be excluded, it is not inconceivable that a neurodegenerative process is inducing such trans-QTLs. Other possibilities that need to be excluded but were not addressed including CNVs local to the 7p21 region and potential population stratification effects. QQ plots for nominal p-values for this locus need to be shown and evaluated for genome-wide inflation as I suspect is the case. Finally, the lack of inclusion of expression PCs as covariates in the trans-QTL analysis is problematic. The authors write that trans-QTLs may be associated with cell-proportion which is not true. While trans-QTLs may have very cell-type specific roles, such a relationship would be elucidated via the cell-type INTERACTING QTLs, rather than omitting cell type proportion covariates (as captured by the 10 gene expression PCs).

5. Along the lines as in #4, it is very much unclear why standard factor analysis approaches such as PEER were not used to calculate QTLs -- These have the benefit of removing hidden confounds like batch and other technical artifacts, as well as controlling for large variance component effect such as cell-type proportional differences across samples. Nearly all standard QTL papers that I have encountered perform some sort of search for optimal number of PEER factors or expression PCs to be used as covariates. Here, the use of 10 is both insufficient in terms of number (GTEx uses many more with a much smaller sample size) and in terms of documentation of how this was chosen. At a minimum, some search for the optimal number needs to be shown and described in more detail. These PCs that capture cell proportions should also be included as covariates for trans-QTL analyses or a lack of genome-wide QQ plot inflation needs to be shown.

MINOR

- Star alignment cites Gencode v32 as a reference but Gencode is an annotation not a reference. I assume this is hg38 but that should be explicitly documented

Reviewer #2:

Remarks to the Author:

The authors present a large adult brain bulk eQTL meta-analysis of 14 publicly available datasets. Though multiple brain regions are sampled, the results are largely driven by samples from the cortex in European individuals (n=2970). The dataset was used to identify 27k primary cis (local) eQTLs as well as secondary or more signals from an additional 16k loci. In addition, trans (distal) eQTLs and interaction (cell type proportion) eQTLs were also discovered. The dataset was then used for subsequent co-localization, Mendelian randomization, and novel co-regulation work to help explain the mechanism of brain-related GWAS loci. This is a fantastic study that makes available a large dataset that will be immensely useful to the community and explains new mechanisms underlying common variant risk for multiple disorders. Nevertheless, I have some comments that I hope will be helpful in revising the manuscript.

1. From the supplementary note, only 10% of trait associated GWAS loci are in LD with an eQTL in this the most powerful brain eQTL study to date. Given a recent review asking where are the disease-associated eQTLs (PMID: 32912663/), this seems like something especially worthy of mentioning in the main text and not hidden in supp note. This may suggest that further increases in sample size of "standard" eQTLs will not lead to an increase in explanation of disease associated loci. I would be curious to hear the authors' thoughts on this idea.
2. The dataset was largely composed of cortical samples but it was not clear from which cortical regions the samples were derived. The authors use 10 PCs to control for global differences in transcriptomics during the QTL, nevertheless it would be beneficial to explicitly control for cortical region or minimally show that the PCs directly adjusts for this.
3. It would be beneficial to use more precise definitions of cis/trans vs local/distal (PMID: 25707927).
4. This statement doesn't seem to match with colocalization or pLI results (Figure 3B): "Thus, secondary, tertiary or quaternary cis-eQTL SNPs could potentially be even more interesting to follow-up than certain primary cis-eQTL SNPs to link association signals to function."
5. It's not clear if summary statistics from the whole genome (not just significant loci) from each of the different eQTL analyses run here are available on the metabrain.nl website. There is a "Download summary stats" link but not sure what it does yet. This will be extremely valuable to the research

community.

6. Were sample swap checks/matching of RNA with DNA performed with tools like verifyBAMID?

7. The methods did not provide enough description for me to understand how conditional tests were conducted in a meta-analysis framework? Was this done within each site and then meta-analyzed across sites?

8. It wasn't clear to me if the multiple corrections permutations based method used controlled for multiple SNPs tested for each gene and multiple genes separately. This is especially a question for the cell type proportions QTLs, where there was no description of a permutation approach. Without such a hierarchical multiple testing correction method, there can be an increase in false positive findings (PMID: 30189032).

9. Using only one SNP (one instrument, one point) is in my opinion not sufficient to confidently call MR. Instead, allelic heterogeneity is needed where multiple independent loci are associated with both the gene expression and the downstream trait. Without this they are effectively fitting a line through the origin and one point. I would minimally request that the number of independent loci for each MR analysis be stated.

10. Downstreamer: "genes within loci" seem to be implicated by proximity alone, not using the eQTL resource generated in this paper. This seems like a wasted opportunity to appropriately assign GWAS locus to gene.

11. I did not see a description in methods of concordance calculation (Figures 3C-E; assuming it is π_1 or is it sign of effect concordance)? It wasn't clear to me if Figure 3E, Supp Fig 11 comparing concordance with GTEx (across tissues including brain) excludes GTEx? Otherwise it could be overfit.

12. It would be interesting to see concordance/sharing from fetal brain to adult brain to assess temporal specificity of effects (PMIDs: 31626773, 32268104, 30419947)

13. The discussion on lack of eQTL discovered at the MAPT/chr17 inversion region locus was interesting. I was also confused because chr17 does show an MR for intracranial volume rs199441 in the chr17 inversion region (Supp Fig 19). Can the authors discuss why the contig/multiple annotation issue only affects MAPT and not other genes within the region?

14. Was any kind of heterogeneity test applied to the meta-analysis results (from either cis or trans analyses)? In the forest plots in Supp Figure 25 most trans-eQTLs seems to be driven by ROS-MAP, which does seem strange in that CMC is almost of equivalent sample size yet not driving any of the results.

15. For ENA samples, were the data checked against multiple samples from the same donor within ENA and across all the other cohorts participating by comparing genotypes to ensure donors were not double counted? For ENA samples, heterozygous sites with strong allelic bias in expression may falsely call genotypes. How is this corrected? How are intergenic variants far away from genes called?

Typos/small things:

1. "Thus, there is a level of uncertainty for the expected proportion fr each cell type"

2. "The majority 700 of included samples was of EUR descent" -> "were"

3. GTEx appears twice in data availability section.

4. Supplementary Figure 1D missing

5. Supplementary Figure 9, can you please label the expression bins from highest to lowest.

6. Supp Figure 10 legend for part B doesn't seem to make sense. The y-axis is density not cerebellum expression and not sure what the dotted line represents.

7. Supplementary Figure 12, would be useful to have correlation coefficients and p-values

8. In Figure 1 bottom left hand cartoon, I think it's more useful to give N in terms of number of individuals vs number of samples as this is most important for the eQTL analysis.

9. Figure 2D looks overplotted to me, consider making dot sizes smaller so that points are not occluded.

Author Rebuttal to Initial comments

Summary of changes:

We would like to thank both reviewers for their valuable comments and suggestions. We here enclose a substantially revised version that should address all major concerns. We have summarized the major changes below:

- We have developed a meta-analysis version of FastQTL to be more in line with the way the GTEx study estimates false discovery rate (FDR), as suggested by Reviewer 1. All downstream analyses have been reperformed with this new FDR approach. A detailed comparison of these methods has been added to the Supplementary Note. This, together with more pervasive correction for principal components, has yielded additional significant *cis*-eQTLs in each brain region. We show that identified eQTLs generally have low heterogeneity across datasets highlighting the robustness of our approach.
- We now perform comparisons between eQTL results using the π_1 statistic and genetic correlation (R_b) in addition to allelic concordance. This has not changed our observation that the majority of *cis*-eQTLs are consistent across brain regions and ethnicities (as indicated by R_b and allelic concordance) and that observed differences are mostly due to differences in sample size (as indicated by π_1).
- We have updated the cell type proportion prediction using a more robust cell type reference panel to distinguish between specific neuron subtypes and have removed cell type predictions for non-cortex regions. Furthermore, we have compared our cell type interaction results to two additional datasets, including the Cortex-AFR subset and a large independent single cell RNA-seq dataset, which suggest that our interaction eQTLs are robust and consistent, but also that our analysis finds cell type interactions that have so far not been detected in single cell analyses.
- We have calculated *trans*-eQTLs by including and excluding AMP-AD datasets and tested dependence of *trans*-eQTLs on neuronal proportions and Alzheimer's disease status. This indicates that *trans*-eQTL and neuron proportion associations in the 7p21.3 locus that we described in our initial submission are not likely to be driven by disease status. We show that after correcting for these effects, hundreds of *trans*-eQTLs remain significant with low heterogeneity across datasets.
- We have also clarified our description of single SNP MR and explained reasons for only using a single SNP. The updated *cis*-eQTL results caused a slight increase in the number of loci with significant MR and colocalization results. We have summarized the major changes in a section after the Response to referees.

Questions from reviewers are displayed in **bold**, responses in plain text, and changes to the manuscript in *italics*. Changes in the main text and Supplementary Note are highlighted in yellow.

Response to referees

Reviewer #1

Brain expression quantitative trait locus and network analysis reveals downstream effects and putative drivers for brain-related diseases

De Klein et al performed meta-analysis of 14 published human brain gene expression cohorts to identify cis and trans eQTLs active across 7 major tissue groups in human brain (amygdala, basal ganglia, cerebellum, cortex, hippocampus, hypothalamus, and spinal cord). They perform a traditional set of cis/trans-QTL analyses and disease GWAS enrichments (e.g., MR and coloc) to prioritize candidate risk genes and mechanisms underlying complex brain-relevant traits. They perform comparisons of QTL architecture across brain regions, ancestries, and tissues, as well as generate co-expression networks in brain. Applications to neurological traits like MS are described.

Overall, the group has substantial expertise with these types of analyses which is why I find it all the more puzzling why they use several less-standard approaches that give results which in many cases are quite divergent from those in the published literature (e.g., much higher concordance rates, see below). Furthermore, despite the size of this paper and the extremely lengthy supplement, in many cases I don't find explicit comparisons to these more widely used methods and results, which make it all the more difficult to evaluate. As such, aside from the size of the dataset, I don't find the advances presented here to be substantial either methodologically or analytically. Furthermore and perhaps more concerning, I am surprised by the lack of clarity, robustness, or comprehensiveness of the analyses and results described from a group that is otherwise positioned to understand the current state-of-the-art pipelines for functional genomics. The webportal is well done and much appreciated.

We thank the reviewer for their comments. As outlined above we have now addressed all points to improve the clarity, robustness and comprehensiveness of our work. We believe that our work presents substantial advancements over previous work due to our harmonization of datasets, *trans*-eQTL analysis, cell type interaction eQTLs, analysis of eQTLs across different brain regions and ethnicities, and integrated MR and colocalization analysis. Additionally, by releasing harmonized results of individual level cohorts we allow users to assess the heterogeneity of observed effects, which makes it easier to identify when significant findings have conflicting results in different cohorts.

1. Cis-QTL mapping -- The authors identify ~12k cis-eQTL in cortex-EUR, which is the most well powered group in their analysis. Despite the large sample size, this is actually substantially smaller I believe compared with results from previously published studies with smaller sample sizes (e.g, CommonMind, BrainSeq, PsychENCODE) at similar FDR thresholds. This is very likely due to analytic differences in how datasets are processed and how QTLs are calculated. This should be explicitly addressed. This manuscript uses a genome-wide permutation approach (n=10) to generate a null distribution. While it is claimed that this converges/stabilizes faster than locus-specific approaches,

this is not standard in the field and should at least be compared with the beta approximated null approach as implemented in FastQTL/QTLtools and used by groups like GTEx.

In our initial submission, we performed eQTL analysis using our eQTL mapping pipeline (EMP). We have used this pipeline previously in several highly cited eQTL meta-analyses on bulk data (Fehrmann *et al.*, PLoS Genet., 2011, Westra *et al.*, Nat. Genet., 2013, Zhernakova *et al.*, Nat. Genet., 2017, Bonder *et al.*, Nat. Genet., 2017), single cell data (van der Wijst *et al.*, Nat. Genet., 2018) and more recently in the largest so-far published eQTL meta-analysis in blood (eQTLgen consortium, Vosa *et al.*, Nat Genet., 2021). We have now implemented a multiple testing correction method that uses exactly the same principles as FastQTL but then in a meta-analysis context. We call this software mbQTL, and have compared it with EMP and FastQTL (**Supplementary Note** and new **Supplementary Figure 35**). This comparison shows that EMP provides more stringent FDR cutoffs than FastQTL and mbQTL, the latter of which produce more comparable results. We have now updated all *cis*-eQTL results using this mbQTL approach while applying 1,000 permutations, which has increased the number of *cis*-eQTLs to 16,169, which is much more in line with previous publications, such as the PsychENCODE publication.

We have reperformed downstream analyses, and have updated the relevant display items. The main text now reads:

We identified 1,880 (Basal ganglia-EUR), 10,577 (Cerebellum-EUR), 4,797 (Cortex-AFR), 16,169 (Cortex-EUR), 1,265 (Hippocampus-EUR), and 998 (Spinal cord-EUR) cis-eQTL genes (q-value<0.05; Figure 3A; Supplementary Table 2).

Furthermore, we have adjusted the methods text as follows:

*To correct for multiple testing, we reperformed each of these meta-analyses, while permuting the sample labels. For each gene, we retained the p-value of the SNP with the lowest nominal association p-value in both the unpermuted and permuted results. Then, we fitted a beta distribution using the permuted p-values per gene, which we used to adjust the unpermuted nominal p-value, comparable to the approach implemented in FastQTL/QTLTools⁹⁷. To determine significance, we then applied the q-values package in R⁹⁸ on the beta-distribution adjusted p-values per gene, using a lambda of 0.85, and considered genes with a q-value < 0.05 as significant. [...] We then repeated the cis-eQTL analyses in each of the eQTL datasets after correcting the optimal number of PCs using 1,000 permutations. A comparison of different permutation based multiple testing correction strategies can be found in the **Supplementary Note** and **Supplementary Figure 35**.*

While the FastQTL-based permutation strategy can be applied for *cis*-eQTL analysis, it cannot easily be applied when performing *trans*-eQTL analysis. Furthermore, the number of published *trans*-eQTL analyses with large sample sizes have so-far been limited, and consequently we argue that no clear consensus exists about the proper permutation strategy for *trans*-eQTLs. For this reason, we decided not to change the permutation strategy for the *trans*-eQTL analysis, and adhere to the permutation strategy we also used for *trans*-eQTLs in the eQTLgen manuscript.

2. Concordance -- The assessment of cis-eQTL concordance rates across regions/tissues/ancestries does not pass the 'smell' test. I've never seen cis-eQTL concordance rates that high (95-100%), which is

very likely due to the non-standard method used (correlation of Z-score sign). At a minimum, this needs to be compared to much more standard methods like Pi-hat. Likewise, for comparison of effects across ancestries, there needs to be consideration for both standardized and allelic effects, such as implemented in methods like 'popcorn'. That cisQTLs from EAS-Cortex with a sample size of ~200 show >95% concordance with Cortex EUR with a sample size of ~3000 in both directions seems to undercut the entire premise of this study that 'bigger is better' for discovering gene regulatory effects.

We apologize for having been unclear here. We realize that concordance and replication can mean different things to different researchers. We have changed the word 'concordance' to 'agreement' in the text when talking about comparing eQTL effects. We acknowledge that there are different methods to determine agreement. We have now updated our manuscript to include three measures for agreement, including allelic concordance to measure sharing of effect directions, the π_1 statistic to measure the proportion of shared significant effects, and the R_b measure to estimate the correlation in effect sizes. In the methods section we now explain what the differences between these methods are as follows:

We have used three different measurements of agreement of eQTL effects when comparing between different brain regions or tissues: allelic concordance, π_1 , and R_b . Each of these measures evaluates different aspects of replication: allelic concordance is an indication of the proportion of effects that have a shared direction of effect within the set of eQTLs that is significant in both discovery and replication dataset and is expected to be 50% for random eQTL effects, π_1 ¹¹² estimates the proportion of eQTL effects that are true positive in the replication cohort, but does not take into account effect direction and can be dependent on replication dataset sample size, and R_b ¹¹³ evaluates the correlation of effect sizes between two datasets, taking into account the effect direction, size and standard error in both datasets. To calculate allelic concordance, we took the SNP-gene combination with the lowest p-value in the discovery cohort for each gene if it was significant and matched these with the same SNP-gene pairs in the replication cohort if it was also significant. We then determined the percentage of those significant SNP-gene pairs had the same allelic direction of effect compared to the discovery cohort. To calculate π_1 we took the SNP-gene combination with the lowest p-value in the discovery cohort for each gene if it was significant, ordered the SNP-gene combinations from lowest to highest p-value, and matched these with the same SNP-gene combination in the replication cohort. We then calculated the proportion of true null p-values (π_0) with the pi0est function of R, with parameter p = the p-values of the match SNP-gene combinations in the replication cohort, and calculated π_1 with $1 - \pi_0$. Similarly, we calculated R_b by taking the SNP-gene combination with the lowest p-value in the discovery cohort for each gene if it was significant, matching those to the replication dataset, regardless of significance in the replication dataset.

Furthermore, we have updated the main text to explicitly compare these different measures of agreement:

We investigated differences in cis-eQTLs due to ancestry, brain region, data sets and tissue type. We used three metrics for eQTL agreement. We calculated allelic concordance as the proportion of shared significant eQTLs that have the same direction of effect in both discovery and replication dataset, π_1 as

the proportion of eQTLs that are true positives in the replication dataset and R_b as a measure of the correlation of effect sizes between two datasets. We first compared Cortex-EUR, Cortex-AFR and a smaller, East Asian cortex dataset (Cortex-EAS; $n=208$, limited to the ENA dataset; **Figure 2C**). We observed high allelic concordance and R_b estimates between the different ethnicities ($R_b > 0.78$, **Figure 3C**; allelic concordance $> 92.95\%$, **Supplementary Figure 12**). Interestingly, π_1 estimates between ethnicities were low when the discovery set had more samples than the replication (e.g. Cortex-EUR vs Cortex-EAS π_1 : 0.29, **Supplementary Figure 12**, but not vice versa (e.g. Cortex-EAS vs Cortex-EUR π_1 : 0.95, **Supplementary Figure 12**). This indicates that while the likelihood of replicating eQTLs depends on the sample size of both datasets, the effect sizes and directions are generally similar, especially if the effects are significant in both the discovery and replication dataset. Next, we compared different brain regions and observed high allelic concordance and R_b estimates between different brain regions overall ($R_b > 0.76$, **Figure 3D**; allelic concordance: $> 91\%$, **Supplementary Figure 12**), and that π_1 estimates were dependent on sample size (0.39-0.95, **Supplementary Figure 12**).

3. Cell-type interacting QTLs -- The cell-type specificity or predominance of QTLs is of interest to the field. However, the implementation in this manuscript is quite bizarre. They use an supervised method (NNLS) with gene expression signatures derived from the CellMap project -- an unpublished resource that can therefore not be fully evaluated. There are now literally dozens of well powered human brain sc/sn-RNAseq datasets defining cell-type-specific gene expression signatures that could have been used (many of which are actually cited in this paper) but were not for unknown reasons. Furthermore, the same signatures were used across brain regions that are known to harbor VERY different cell types -- For example, the cerebellum is comprised of very distinct cell types (Purkinje cells, dentate granule cells, etc) that are not present in cortex and were not included in the reference panel. The proportions estimated are not physiologically reliable -- e.g., the number of astrocytes are much smaller than they should be, the number of macrophages are much larger than they should be (even if substituting microglia for macrophage). Other brain regions here also have very different cell-types (e.g. basal ganglia -- medium spiny neurons; hippocampus has CA1/CA2/CA3 pyramidal cells, etc...). As such, I don't consider any of the cell type analyses to be particularly reliable. There doesn't appear to be much consideration for excitatory vs interneuron neuron effects even though these are very distinct lineages. Finally, the proportions as estimated add up to 1 and therefore have one less degree of freedom than number of cell-types assessed. It doesn't appear that this is accounted for in the deconvolution method, and if so, this should be explicitly addressed in the paper as a limitation.

We thank the reviewer for these helpful comments, because we believe the cell type proportion estimates are very important for one of the main messages of this paper, i.e., the identification of cell type specific eQTLs in brain tissue.

We agree with the reviewer that different brain regions may be constituted of different proportions and types of cells, especially when comparing distinct brain regions such as the cerebral cortex and the cerebellum. We also agree that a signature matrix was derived from cortical tissue only and may not properly reflect the cell composition in cerebellum. We have removed the cell type prediction results for the non-cerebral cortical tissues from our manuscript because there is no corresponding reference dataset available.

We also thank the reviewer for the great suggestion to include more brain sub cell types to better reflect physiological proportions. In this revision, we have: 1) updated the cell type prediction signature matrix to also distinguish between excitatory and inhibitory neurons, 2) slightly altered our cell type prediction method to accommodate the additional cell types and potential dataset differences, and 3) altered our ieQTL analysis method and added additional single nucleus RNA-seq datasets for replication purposes.

First, we replaced the CellMap signature matrix with the signature matrix as supplied by the PsychENCODE consortium (Wang *et al.*; Science, 2018). We note that the CellMap signature matrix in the initial submission was constructed from public datasets (GSE126836, GSE97930, GSE103723, GSE104276, PRJNA544731, PRJNA434002, phs000424, phs001836 and the Allen Brain Atlas) and is now available as a preprint by Ouyang *et al.*, bioRxiv, 2021. The new signature matrix is able to further classify cell types, including excitatory and inhibitory neurons. While updating the cell type reference, we contacted the first author to confirm our cell type prediction method aligns with the method used in the PsychENCODE manuscript. Our method discards the developmental cell types obtained from fetal cells included in the reference as we expect that these cell types are very rare or not present in post mortem adult human brain.

Second, we altered our cell type prediction method by correcting the bulk expression matrix for dataset differences and RNA-seq alignment metrics prior to cell type prediction, since we observed that these covariates introduced biases in the downstream ieQTL analyses. Given these changes, we repeated the cell proportion prediction and observed averages which are more in line with physiological expectations as shown in **Rebuttal Figure 1**. We report a reconstruction accuracy of 87% (SD = 4%) which is very comparable with the 88% reported by Wang *et al.*; Science, 2018.

We decided to sum all the sub-cell types for excitatory neurons, inhibitory neurons and oligodendrocytes (OPC and oligodendrocytes) because most of these sub cell types had very low (<1%) average proportions and would be insufficiently powered to detect interaction eQTLs. After this aggregation step, the estimated average cell proportions were 28% excitatory neurons, 21% astrocytes, 16% other neurons, 15% oligodendrocytes, 12% endothelial cells, 6% microglia and 1% inhibitory cells. These proportions closely resemble the physiological proportions as well as the proportions reported by Wang *et al.*; Science, 2018. The aggregated distributions are shown in **Rebuttal Figure 1**, and **Supplementary Figure 16**.

Rebuttal Figure 1: Left: Predicted cell type proportions in MetaBrain cortex European using the PsychENCODE reference profile. Developmental cell types were discarded in the prediction of these cell types. The value above each violin denotes the average cell fraction in percentages over all samples. Right: Aggregated cell type proportions in MetaBrain Cortex-EUR. Sub-cell types for excitatory neurons, inhibitory neurons and oligodendrocytes (OPC and oligodendrocytes) are summed together. The value above each violin denotes the average cell fraction in percentages over all samples.

To accommodate these changes, we have updated **Figures 4, 5C, 5D, Supplementary Figures 16, 20, 29, 36, and Supplementary Tables 19, 20, 21**. Furthermore, we have removed the Supplementary Figures referencing cell type predictions in brain regions other than cortex. Additionally, we have reperfomed downstream analyses, and have updated the relevant display items. We have adjusted the main text as follows:

*In our Cortex-EUR subset, we predicted cell type proportions using single cell RNA-seq derived signature profiles⁵, including various subsets of inhibitory and excitatory neurons, astrocytes, microglia, oligodendrocytes and endothelial cells (**Supplementary Note, Supplementary Figure 16A**). We performed a correction for RNA-seq quality metrics and dataset indicator variables to prevent confounding effects and enable the joint analysis of all Cortex-EUR samples. The reconstruction accuracy of our prediction (87%) was comparable to previous studies using this reference profile⁵. To increase power to detect ieQTLs, we aggregated sub-cell types if they had a low average predicted frequency (<1%), after which excitatory neurons were the most abundant cell type (average cell proportion: 28.1%), followed by astrocyte (21.4%), other neuron (16.4%), oligodendrocytes (14.5%), endothelial cells (12.2%), microglia (6.4%), and inhibitory neuron (1.1%; **Supplementary Figure 16B**). These averages were very comparable to previous cell type estimations on a subset of these samples⁵. We observed low to moderate correlations between predicted cell types ($0.01 < r < 0.55$; **Figure 4A**), with neuronal cell types showing opposite correlations when compared to other cell types.*

Additionally, we changed the methods text as follows:

In the deconvolution of the MetaBrain bulk expression data we used a single-cell derived signature matrix including the major cell types in the brain: neurons (excitatory / inhibitory / other), oligodendrocytes, astrocytes, microglia, and endothelial cells. We used the TPM single-cell signature matrix published by the PsychENCODE consortium⁵. We decided to discard the developmental cell types because we expect that these cell types are very rare or not present in post mortem adult human brain, and because their signatures were obtained from fetal cells. We TPM normalized the gene expression of the MetaBrain bulk RNA-seq cortex samples and extracted the 418 unique signature genes included in the PsychENCODE single-cell signature profile. We then applied \log_2 transformation on both the signature matrix as well as the bulk gene count matrix. Then, to enable the joint analysis of samples, we corrected the gene counts for 20 RNA-seq quality metrics and dataset indicator variables using OLS. To maintain the information captured by relative expression differences between genes required for deconvolution, we rescaled the residuals to the original \log_2 transformed mean and standard deviation and replaced negative values with zero.

[...]

*Since the average proportions of sub cell types were often very low (**Supplementary Figure 16**), we opted to sum all the sub-cell types for excitatory neurons, inhibitory neurons and oligodendrocytes (OPC and oligodendrocytes). We observed that the first ± 10 expression PCs correlated strongly with predicted cell type proportions for specific cell types (**Supplementary Figure 36**).*

We have made several changes to our cell type interaction eQTL method to minimize potential false positives. This includes additional filters for potential outliers, application of a forced-normalization to the expression data, a more stringent MAF cutoff, an evaluation of a permutation based multiple testing strategy (see **Reviewer 2, question 8**), and exhaustively testing all possible combinations of allele encodings in Decon-QTL (whereas previously, this was limited to a single interaction term). After applying all these changes, we now find 88 inhibitory neuron, 212 endothelial cell, 433 microglia, 660 excitatory neuron, 879 other neuron, 887 astrocyte and 936 oligodendrocyte interaction eQTLs (BH-FDR<0.05). We believe these changes have greatly improved the robustness of the interaction eQTLs we report in our manuscript.

These changes are now described in the methods text as follows:

We excluded 50 samples with a cell-fraction z-score >4 on one or more cell types and limited the analysis to eQTLs with <95% missingness per dataset, a joint MAF >5%, and a joint Hardy-Weinberg equilibrium p-value <0.0001. The remaining 25,497 eQTLs and 2,633 samples were then used to identify cell type mediated eQTL effects.

[...]

Whereas the original Decon-QTL limits the number of genotypes that have an opposite genotypic encoding to one interaction term, we enabled testing of all possible allele encodings to increase accuracy. As expression matrix input, we followed the steps as described in the original Decon-QTL manuscript²³. In short, we took the TMM expression counts, applied a \log_2 transformation and subsequently corrected for dataset indicator variables, 20 RNA-seq alignment metrics, and 4 genotype

MDS components using OLS. As an additional step, we forced the data to the normal distribution per gene to reduce outliers. Finally, we inverted the \log_2 transformation.

Finally, we also assessed the impact of removing or adding an additional degree of freedom from the distribution when calculating the p-values. While we acknowledge that this could potentially have a consequence if the sample size is low (i.e., below 100 individuals), we expected that this should not have a consequence in our analysis. Indeed, we observed no noticeable differences. **Rebuttal Figure 2** compares the $-\log_{10}(\text{p-value})$ for each of the respective cell types.

Rebuttal Figure 2: Pairwise comparison per cell type of the $-\log_{10}(\text{p-value})$ interaction p-values resulting from Decon-QTL when using degrees of freedom equal to samples minus the number of terms in the model (x-axis) or removing one additional degree of freedom (y-axis). Each point is an ieQTL and is colored as follows: green denotes significant in both analyses, blue denotes only significant on the x-axis, orange denotes only significant on the y-axis, and grey is not significant. The horizontal and vertical dashed lines show the significance threshold of $\text{p-value} = 0.05$.

We describe this possible limitation in the methods text as follows:

In our study, Decon-eQTL uses a degrees of freedom equal to the sample size minus numbers of terms in the model (i.e. number of cell types times two). Since the predicted cell type proportions add up to 1, an argument could be made that one less degree of freedom should be used. We observed that this difference of one degree of freedom did not make a difference in the observed p-values per eQTL interaction (average correlation between p-values, $r = 1$), potentially due to the sample size of our dataset. However, we note that this difference may become more significant in studies with smaller sample sizes (e.g. < 100 samples).

We next updated our agreement analysis for the cell type interaction effects using allelic concordance, π_1 and R_b statistics. We compared our findings in Cortex-EUR with three independent datasets: MetaBrain Cortex-AFR (n=319), single nucleus RNA-seq (snRNA-seq) eQTL results from ROSMAP (n=39) and snRNA-seq eQTL results from Bryois *et al.* (n=196, medRxiv, 2021). Overall, for the overlapping eQTLs we found moderate to good agreement between our results. However, replication rates were often dependent on average predicted cell type proportion.

We note that, even with the availability of the larger Bryois study, the sample size of the replication datasets is a fraction of our discovery dataset in Cortex-EUR. Even in this large study, there were eQTL summary statistics available for only 54% of the Cortex-EUR ieQTLs, potentially because single-cell expression levels were too low to test almost half of all cell type specific eQTLs detected in bulk. This affects the agreement between the cell type interaction eQTLs we identified in our study and the replication datasets, especially with respect to the number of overlapping eQTLs and π_1 estimates. Since we observe a good R_b and allelic concordance for the eQTLs that could be tested in single-cell data, we expect that the additional ieQTLs we identified in bulk will also be identified in single nucleus data once sample sizes increase. This further highlights the potential of our work and the value of performing interaction analysis in bulk RNA-seq eQTL studies.

Updates to **Supplementary Figures 18, 20, 30** and **Supplementary Tables 8, 9, 23** reflect these changes. Moreover, we have added **Supplementary Tables 22** and **Supplementary Figures 17 and 19**. The adjusted results text is as follows:

*Next, we attempted replication of our findings in three independent datasets: MetaBrain Cortex-AFR, a single-nucleus RNA-seq (snRNA-seq) dataset from ROSMAP²⁷ (n=39) and a recent snRNA-seq study from Bryois *et al.*²⁸ (n=196). In the Cortex-AFR dataset we observed moderate agreement ($-0.17 < R_b < 0.98$, $0.00 < \pi_1 < 0.06$, $43\% < \text{allelic concordance} < 72\%$; **Supplementary Table 9, Supplementary Figure 17B**), which was highly dependent on the average predicted cell type proportion. Especially the π_1 estimates suggest that the analysis in the Cortex-AFR dataset is somewhat underpowered. When focusing on the 177 ieQTLs that were significant in both datasets, all cell types had an allelic concordance >97% except inhibitory neurons, which had an allelic concordance of 62%. In the ROSMAP snRNA-seq dataset we tested all cis-eQTLs from Cortex-EUR that had one or more interactions and compared the allelic direction of the ieQTL in Cortex-EUR with the eQTL direction in matching cell types. We observed good agreement for most cell types ($0.13 < R_b < 0.82$, $0 < \pi_1 < 0.63$, $48\% < \text{allelic concordance} < 81\%$; **Supplementary Table 9, Supplementary Figure 18B**). A total of 173 cis-eQTLs were significant in the*

Cortex-EUR and the ROSMAP snRNA-seq datasets of which 84 eQTLs (43 for oligodendrocyte and 41 for excitatory neuron) were cell type mediated by the corresponding cell type in bulk with a 100% allelic concordance (**Supplementary Figure 18B**). Last, we compared the Cortex-EUR ieQTLs with the snRNA-seq eQTLs identified in Bryois et al. Only 54% (1,734 out of 3,209 not considering other neuron ieQTLs) of Cortex-EUR ieQTLs were overlapping with the Bryois et al. snRNA-seq study. For the overlapping eQTLs we observed a high agreement for all cell types ($0.78 < R_b < 0.86$, $0.43 < \pi_1 < 0.83$, $81\% < \text{allelic concordance} < 90\%$; **Supplementary Table 9, Supplementary Figure 19B**), except inhibitory neurons, which had lower agreement ($R_b = 0.23$, $\pi_1 = 0.71$, allelic concordance=63%). We observed 1,070 ieQTLs that were significant in the Cortex-EUR ieQTL and the Bryois et al. snRNA-seq eQTL datasets for which all cell types had an allelic concordance >94% except inhibitory neurons, which had an allelic concordance of 71%.

We have added the following to the discussion:

Even with the availability of relatively larger population-based single-cell RNA-seq datasets^{28,78,79}, the sample size of these replication datasets remains low in comparison to our discovery dataset in Cortex-EUR. Furthermore, eQTL discovery in single cells is generally done with small datasets and is dependent on multiple factors including zero-rate of the gene, sequencing protocol, and number of cells measured per individual, causing a low eQTL overlap. This affects the agreement between the cell type interaction eQTLs we identified in our study and the replication datasets, especially with respect to the π_1 estimates. We expect that, since we observed good R_b and allelic concordance values for the interaction eQTLs that do overlap, the additional ieQTLs we identified have a good chance of being replicated in single nucleus eQTL datasets of the brain once the sample sizes of these studies increase. This highlights the potential of our work and performing interaction analysis in bulk RNA-seq eQTL studies in general.

4. Trans-QTLs. The trans-QTL analysis also does not pass the ‘smell’ test. That ~80% of trans-QTLs are derived from the single 7p21 locus is VERY concerning for an underlying confound or hidden bias in the data. I think this most likely represents a cell proportion shift – it seems that all the trans-eGenes are very specific to neuronal (rather than glial) lineages. As such, any SNP that is associated with even slight loss of neurons would then be confoundingly associated with top marker genes of that cell-type, irrespective of whether there was any actual genomic regulatory relationship. Given that a large number of the samples included here come from individuals with a neurodegenerative disorder like Alzheimer’s, and these samples do not appear to be excluded, it is not inconceivable that a neurodegenerative process is inducing such trans-QTLs. Other possibilities that need to be excluded but were not addressed including CNVs local to the 7p21 region and potential population stratification effects. QQ plots for nominal p-values for this locus need to be shown and evaluated for genome-wide inflation as I suspect is the case. Finally, the lack of inclusion of expression PCs as covariates in the trans-QTL analysis is problematic. The authors write that trans-QTLs may be associated with cell-proportion which is not true. While trans-QTLs may have very cell-type specific roles, such a

relationship would be elucidated via the cell-type INTERACTING QTLs, rather than omitting cell type proportion covariates (as captured by the 10 gene expression PCs).

We acknowledge the reservations of the reviewer about the *trans*-eQTL results in our initial submission. The reason why we presented the results is that there are multiple publications that have previously associated the 7p21.3 locus with changes in gene expression (Ren et al, Mol. Neurodegener. 2018, Yang et al, Neuron, 2020) and neuronal cell type composition (Li et al, Acta Neuropathol. 2020), suggesting that this locus could harbor multiple *trans*-eQTL effects.

We have now more carefully assessed the *trans*-eQTL effects in this region in multiple ways: we have performed *trans*-eQTL analysis with and without AMP-AD datasets, and while including and excluding a correction for PCs. This analysis shows that the majority of the 7p21.3 *trans*-eQTLs disappear when removing AMP-AD datasets, or when correcting for 100PCs. Analyzing heterogeneity of effect sizes in the meta-analysis (I^2 ; **Reviewer 2, Question 14**) indicated that the 7p21.3 *trans*-eQTLs show higher heterogeneity across datasets than the other *trans*-eQTLs, regardless of the analysis performed. Furthermore, we have investigated whether the 7p21.3 *trans*-eQTLs are dependent on neuronal proportions (**Supplementary Table 19**) and Alzheimer's Disease status by applying interaction analysis (**Supplementary Table 24**). These analyses show that the 7p21.3 *trans*-eQTL effects are likely specific to AMP-AD datasets, but we do not find evidence that they are due to Alzheimer's disease status. Additionally, we have looked in this region to see if copy number variation (CNV) is contributing to the effects observed at this locus. This is an unlikely scenario because CNVs in this region are extremely rare in gnomAD SVs v2.1. Furthermore, if a CNV would be the cause of the observed effects, we would expect the CNV to also have an effect in the datasets other than AMP-AD, unless the CNV would have been associated with AD. However, to our knowledge, such an association has not been identified in this locus so far. Finally, considering the results described above and the fact we corrected for potential population stratification by correcting for MDS components calculated over the genotype data, we believe it is unlikely that the observed effects are due to population stratification. Finally, we have added QQ-plots to **Supplementary Figure 27**, which show a λ -inflation of 2.1 when no PCs are corrected, which decreases to 1.05 when 100PCs are corrected.

We have updated the "Trans-eQTLs in the 7p21.3 locus" section in the **Supplementary Note** and included these analyses in the main text as follows:

*First, we determined the optimal number of PC correction (Supplementary Figure 7). We observed the largest number ($n=4,226$) of significant trans-eQTLs when no PCs were removed, followed by a decrease in significant effects, which stabilized after removing 100PCs ($n=1,639$). We removed trans-eQTLs for which the gene partially mapped within 5Mb of the trans-eQTL SNP, after which 2,971 (70%; 0PCs) and 737 (44%; 100 PCs removed) significant trans-eQTLs remained ($FDR<0.05$; **Supplementary Note; Figure 6A; Supplementary Table 17**). Without PC correction, we observed high heterogeneity (69% having $I^2>0.5$), which decreased after correcting 100 PCs (12.8% having $I^2>0.5$; **Supplementary Figure 26**). The majority (85%) of the trans-eQTLs observed without PC correction were located in a 7p21.3 locus, previously associated previously been associated with frontotemporal lobar degeneration⁵⁰, Alzheimer's*

disease⁵¹, changes in predicted neuron proportions⁵² and gene expression levels^{53,54}. However, we did not find evidence that they were dependent on Alzheimer's disease status or cell type proportions (**Supplementary Note; Supplementary Figures 26-29 and Supplementary Tables 17-24**). These trans-eQTLs were not significant when correcting for 100 PCs.

We therefore concentrated on 737 trans-eQTLs, detected after correcting for 100 PCs, which reflect 526 unique SNPs, and 108 unique genes. 127 SNPs had trans-eQTL effects on multiple genes. 461 (88%) of the trans-eQTL SNPs were associated with a significant cis-eQTL in Cortex-EUR, of which 150 (33%) were also a cis-eQTL index SNP, compared to 15,745 index variants out of 109,172 SNPs associated with a significant cis-eQTL among the list of other tested SNPs, representing an enrichment (Fisher exact test p -value=1.2x10⁻²⁸). This indicates that cis-eQTL index SNPs more often yield trans-eQTL effects in brain as compared to other cis-eQTL variants. 29 were also cis-eQTL SNPs in tissues other than cortex, suggesting that trans-eQTLs can also be observed for cis-eQTLs index SNPs identified in other tissues (**Supplementary Table 17**).

Since the trans-eQTLs that remain after correcting for 100 PCs show low heterogeneity, this suggests that these effects are more stable across datasets. We have therefore updated the main text using these results. Using this set of results, the *HBG* convergence example is still present, but the *KCNAS* convergence example is not. We have replaced this example with converging effects on the *ZNF311* gene.

5. Along the lines as in #4, it is very much unclear why standard factor analysis approaches such PEER were not used to calculate QTLs -- These have the benefit of removing hidden confounds like batch and other technical artifacts, as well as controlling for large variance component effect such as cell-type proportional differences across samples. Nearly all standard QTL papers that I have encountered perform some sort of search for optimal number of PEER factors or expression PCs to be used as covariates. Here, the use of 10 is both insufficient in terms of number (GTEx uses many more with a much smaller sample size) and in terms of documentation of how this was chosen. At a minimum, some search for the optimal number needs to be shown and described in more detail. These PCs that capture cell proportions should also be included as covariates for trans-QTL analyses or a lack of genome-wide QQ plot inflation needs to be shown.

We used PCs to remove hidden confounders as PEER factors and PCs generally capture similar variation from the gene expression data. To illustrate this, we have now performed PEER analysis on the Cortex-EUR dataset, and correlated the resulting PEER factors with the PC eigenvectors obtained from that dataset (**Rebuttal Figure 3**). This shows that PEER factors and PCs show a very high degree of similarity for the first 15 PCs and PEER factors. Later PCs and PEER factors correlate less strongly, but the variance explained per factor is low for those. Consequently, correcting for PCs or PEER factors should produce similar residual gene expression values.

Rebuttal Figure 3: PEER factors, especially those which explain the most gene expression variance (i.e. the first 20), are highly correlated with (nearly) identically numbered PCs.

As suggested by the reviewer, we have now performed an explicit scan to determine the optimal number of principal components to remove. These results are now summarized in **Supplementary Figure 7**. We have also added the following to the methods section:

*To account for any residual technical variation in the cis-eQTL analysis, we additionally corrected the gene expression data for PCs calculated over the RNA-seq sample correlation matrix. We determined the optimum number of PCs to remove, by performing a set of eQTL meta-analyses, removing up to 100 PCs, in steps of 10 PCs, for each eQTL dataset (**Supplementary Figure 7**). [...]*

To determine the optimal number of PCs to remove for the cis-eQTL analysis, we used 250 permutations, and concluded that the optimum number of PCs to remove was 30 PCs for Basalganglia-EUR and Hippocampus-EUR, 60 PCs for Cerebellum-EUR, 40 PCs for Cortex-AFR, 80 PCs for Cortex-EUR and 20 PCs for Spinalcord-EUR. We then repeated the cis-eQTL analyses in each of the eQTL datasets after correcting the optimal number of PCs using 1,000 permutations.

Similarly, we have performed this analysis for the *trans*-eQTLs. The relevant methods section now reads:

*As with the cis-eQTLs, we performed a scan for the optimal number of PCs to remove in the trans-eQTL analysis and observed that the number of trans-eQTLs stabilized after 20 PCs were removed (**Supplementary Figure 7**). For consistency with the cis-eQTL analysis with these datasets, we therefore chose to remove 100PCs when AMP-AD samples were included, and 80PCs when AMP-AD samples were excluded.*

MINOR

- Star alignment cites Gencode v32 as a reference but Gencode is an annotation not a reference. I assume this is hg38 but that should be explicitly documented

We thank the reviewer for pointing us to this error. The text now reads:

RNAseq data was processed using a pipeline built with molgenis-compute⁹². FASTQ files were aligned against the GRCh38.p13 primary assembly using the GENCODE⁹³ v32 annotation with STAR⁹⁴ (version 2.6.1c)

Reviewer #2

The authors present a large adult brain bulk eQTL meta-analysis of 14 publicly available datasets. Though multiple brain regions are sampled, the results are largely driven by samples from the cortex in European individuals (n=2970). The dataset was used to identify 27k primary cis (local) eQTLs as well as secondary or more signals from an additional 16k loci. In addition, trans (distal) eQTLs and interaction (cell type proportion) eQTLs were also discovered. The dataset was then used for subsequent co-localization, Mendelian randomization, and novel co-regulation work to help explain the mechanism of brain-related GWAS loci. This is a fantastic study that makes available a large dataset that will be immensely useful to the community and explains new mechanisms underlying common variant risk for multiple disorders. Nevertheless, I have some comments that I hope will be helpful in revising the manuscript.

We thank the reviewer for reviewing our manuscript.

1. From the supplementary note, only 10% of trait associated GWAS loci are in LD with an eQTL in this the most powerful brain eQTL study to date. Given a recent review asking where are the disease-associated eQTLs (PMID: 32912663/), this seems like something especially worthy of mentioning in the main text and not hidden in supp note. This may suggest that further increases in sample size of “standard” eQTLs will not lead to an increase in explanation of disease associated loci. I would be curious to hear the authors’ thoughts on this idea.

We agree this is an interesting point. The observation of low LD overlap with GWAS loci has been shown in our earlier observation in our larger whole blood-based eQTL meta-analysis that included 31,684 individuals (eQTLgen; Võsa *et al*, Nat Genet, 2021; Supplementary Results p111), where we explored 12,911 disease associated loci and observed that 29.3% were in LD with a *cis*-eQTL SNP. Although the set-up of that analysis was different than our analysis in brain cortex, it seems supportive that a larger sample size could still help explain more disease associated loci. For instance, it could help in more accurate fine-mapping of eQTL signals, which has recently been shown in a study by Gazal *et al*. (Nat. Genet, 2022) to more accurately link disease variants to genes. We expect this to be particularly the case for eQTLs with smaller effect sizes, which may be more relevant to disease.

A recent preprint by Mostafavi *et al*. (bioRxiv, 2022) suggests that GWAS and eQTL variants have different distances towards the TSS and overlap different kinds of regulatory elements. It could be that such GWAS associations are better captured by non-primary eQTLs. Our results show that the additional non-primary eQTLs, which are generally located further away from the TSS, also do not appear to greatly improve LD overlap with GWAS SNPs. Adding additional samples would lead to the identification of additional non-primary eQTLs, but we expect that this will not resolve most GWAS associations, since these non-primary associations are generally located close to the TSS as well. Nevertheless, disentangling independent eQTL associations could further aid future colocalization (e.g. Wallace, PLOS Genet. 2020) and mendelian randomization approaches.

Another possibility is that studies in bulk RNA-seq studies generally capture strong eQTL effects mainly for genes that are not dosage sensitive, and do not cause disruptive downstream consequences (Wang

et al., Am J Hum Genet, 2020). As outlined in the review by Umans *et al.* (Trends Genet, 2021), it might very well be that the GWAS relevant eQTLs only exist for specific gene isoforms, in specific tissues under certain contexts or in specific cell subpopulations, or combinations thereof. We believe that large single cell eQTL studies in multiple adult tissues as well as developmental tissues or iPSCs, such as sc-eQTLgen, GTEx, or dGTEx, should shed light into many of these factors.

Since we have updated the eQTL results on the basis of the reviewers' suggestions, we have now repeated the LD overlap, MR and colocalization analyses and have updated the relevant **Supplementary Tables 10**, and **12-16**. While the updated set of results has caused a slight increase in the LD overlap with *cis*-eQTL SNPs compared to our initial submission, the majority of brain-related GWAS variants cannot be linked to bulk brain eQTLs.

We have added the following to the discussion:

Our LD overlap analysis indicates that up to 15% of brain-related GWAS loci can be linked to a cis-eQTL variant when using 2,683 samples. While this low overlap could be caused by incomplete power in the GWAS and eQTL studies, we believe this to be unlikely: in our recent study in whole blood using 31,684 individuals⁴⁹, the proportion of overlapping variants was 29.3%, indicating that a ten-fold increase in sample size would yield at most a two-fold increase in LD overlap. Nevertheless, increasing the sample size of bulk eQTL studies could improve future fine-mapping efforts, which may help efforts to link disease SNPs to genes⁸³. We note that LD overlap has known limitations that may cause biased overlap estimates, including coincidental sharing of the association with the GWAS trait due to surrounding LD patterns in the genomic region, or lack of sharing due to uncertainty about the location of the causal variant. To circumvent some of the limitations of LD overlap, we focused our approach on a combination of Mendelian randomization and colocalization. [...]

It has recently become clear that many eQTLs can only be detected for specific splice isoforms or under specific contexts, such as specific cell types, disease status, or stimulation⁸⁹. We believe that current and future single-cell RNA-seq and CRISPR-based perturbation methods will shed light into many of these contexts. When sample sizes of these studies increase this will likely increase the proportion of GWAS loci that can be linked to eQTLs. Nevertheless, we believe that our study provides a step forward into the interpretation of brain-related disease GWAS signals.

2. The dataset was largely composed of cortical samples but it was not clear from which cortical regions the samples were derived. The authors use 10 PCs to control for global differences in transcriptomics during the QTL, nevertheless it would be beneficial to explicitly control for cortical region or minimally show that the PCs directly adjusts for this.

The MetaBrain dataset contains cortical samples derived from several regions of the brain (**Supplementary Table 1**). The cortical region annotation was not available for all the samples, especially the ENA cohort. Thus, we are unable to explicitly control for the cortical region.

To ensure we have corrected for these regional differences, we updated our PC correction from 10PCs to 80 PCs for Cortex-EUR and from 10 PCs to 40 PCs for Cortex-AFR due to **Reviewer 1, Question 5**. We have correlated these 80 PCs to the different cortical subsets, excluding cortex samples from ENA for

which we do not have a subregion annotated and excluding the one sample sampled from Subgenual anterior cingulate cortex. In total this included 2,435 samples (see figure). None of these correlations were significant after FDR multiple testing correction (all FDR corrected p-values = 1), suggesting that there are no large differences in gene expression due to cortical region in our expression data. After correction of the expression data for the first 80 PCs, we recalculated 50 PCs and correlated these to the different cortical regions. Again, all the correlation p-values were 1 after multiple testing correction (**Rebuttal Figure 4**).

Rebuttal Figure 4: Left: Z-scores from correlation (spearman) between cortex sub regions and the first 80 PCs calculated on Cortex-EUR samples for which sub regions were available. None of the correlations pass multiple testing correction at $p < 0.05$. Right: Z-scores from correlation (spearman) between cortex sub regions and the first 50 PCs calculated on Cortex-EUR samples after PC correction for which sub regions were available. None of the correlations pass multiple testing correction at $p < 0.05$.

Consequently, it is unlikely that cortical subregions largely affect the eQTL results. Additionally, when comparing eQTLs between different eQTL datasets, we observed high agreement between ethnicities and cerebral tissues in both our initial submission as well as this version of the manuscript, suggesting that many eQTLs are shared within the brain when eQTLs are assessed using bulk RNA-seq. We expect

that the major differences between cortical regions will only be found using methods like single cell RNA-sequencing or spatial transcriptomics.

As we did not find any difference in expression between the more specific brain regions after correction for technical covariates and hidden effects with PCs, we decided to not explicitly correct for these more specific brain regions.

3. It would be beneficial to use more precise definitions of cis/trans vs local/distal (PMID: 25707927).

We acknowledge that the definition of *cis* and *trans* used in the eQTL literature, and by extension in our work, does not adhere to the literal definitions of these words. For ease of discussing the results and comparisons to previously published work, we have chosen to use the *cis*- and *trans*- terminology that is commonly used in eQTL manuscripts. However, to accommodate the valid point of the reviewer, we have added the following section to the introduction:

While classically the definitions for cis- and trans are dependent on the biological mechanism³, in this manuscript, we define cis-eQTLs as variants affecting nearby genes (1Mb), and trans-eQTLs as variants affecting distal (>5Mb) genes, with no clear distinction in mode of action.

And the following to the discussion:

Throughout this manuscript, we used the term cis-eQTLs to describe local eQTLs, and trans-eQTLs to describe distal effects. However, we acknowledge that cis and trans do not necessarily implicate local and distal, respectively, but rather the mode of action of the effect alleles³. Distinguishing between local, distal, cis and trans would require additional analyses that determine the potential mode of action of the allele, such as integration of QTLs from different molecular phenotypes such as transcription factor binding, protein expression and epigenetic signals to identify true trans-acting alleles, and allele specific expression to identify true cis-acting alleles.

4. This statement doesn't seem to match with colocalization or pLI results (Figure 3B): "Thus, secondary, tertiary or quaternary cis-eQTL SNPs could potentially be even more interesting to follow-up than certain primary cis-eQTL SNPs to link association signals to function. "

We agree and have removed the sentence from the manuscript.

5. It's not clear if summary statistics from the whole genome (not just significant loci) from each of the different eQTL analyses run here are available on the metabrain.nl website. There is a "Download summary stats" link but not sure what it does yet. This will be extremely valuable to the research community.

The website that the reviewer refers to might have been changed during the review process. Currently, the summary statistics can be downloaded after registering with an e-mail address. We understand that this may compromise the anonymity of the reviewer. We apologize for this oversight. The direct link to the download page is here: <http://download.metabrain.nl/files.html>. We have temporarily stored the updated cis-eQTL summary statistics here: <https://data.harmjanwestra.nl/metabrain/2021-07-23->

SummaryStats/. The updated summary statistics will be moved to the main MetaBrain website in the near future.

6. Were sample swap checks/matching of RNA with DNA performed with tools like verifyBAMID?

We performed sample swap analysis using MixupMapper, which uses eQTL information to determine the best matching RNA-seq per genotyped sample. We apologize it was omitted in the previous version of our manuscript. Our MixupMapper analysis identified a number of sample mix-ups or mismatching samples, which we have summarized now in **Supplementary Table 1**.

We have added the following to the methods section:

In each of these eQTL datasets, we next applied MixupMapper¹⁰⁵ per cohort to identify mismatched samples: MixupMapper uses eQTLs identified in a cohort to determine if the RNA-seq sample is the best match for its assigned genotype sample. Using this procedure, we detected a single mismatch in Braineac for Basal ganglia-EUR, and 4 mismatches in AMP-AD ROSMAP, 34 in BrainGVEX and 121 in the ENA cohort in Cortex-EUR (Supplementary Table 1).

7. The methods did not provide enough description for me to understand how conditional tests were conducted in a meta-analysis framework? Was this done within each site and then meta-analyzed across sites?

We apologize for not having been clear here; the conditional analysis was done within each site, and then meta-analyzed across sites. We have now updated the description of this analysis in the methods section to provide more details:

Since cis-eQTL loci are known to often harbor multiple independent associations, we performed an iterative conditional analysis per eQTL dataset. Since our eQTL analyses were performed in a meta-analysis context, our conditional analysis consisted of several steps. From our initial meta-analysis (i.e., the first iteration), we had identified genes with a significant association, and the SNP with the strongest association per gene. For the second iteration of meta-analysis, we first removed the effect of the SNP having the strongest association in each included cohort using linear regression. We then repeated the meta-analysis on the residuals of that regression, focusing on genes that had a significant association in the first iteration. To determine significance of the results, we used the beta-distribution estimates for each gene that were obtained in the first iteration. We used these beta-distributions to adjust the nominal p-values of strongest SNP association per gene, and used calculated the q-value to correct for multiple testing. Genes with a q-value < 0.05 were considered significant. We repeated this procedure in consecutive iterations, each time adding the SNP with the strongest association per gene to the linear regression model for that gene, until no significant associations were found. We note that because of the setup of our analysis it could be that variants identified in the second iteration could be in LD with variants of the first iteration, and consequently should not be considered as true secondary eQTL effects. This can happen for example if a variant is present in only a part of the included cohorts. Therefore, after completing all iterations for an eQTL dataset, for each gene, we determined the LD between the variant of each iteration and the next iteration, and excluded variants when they had an $r^2 > 0.8$, or when no LD

could be calculated because of mutually exclusive missingness between included cohorts. We finally re-ranked the remaining variants to determine secondary, tertiary, quaternary, etc. variants.

8. It wasn't clear to me if the multiple corrections permutations based method used controlled for multiple SNPs tested for each gene and multiple genes separately. This is especially a question for the cell type proportions QTLs, where there was no description of a permutation approach. Without such a hierarchical multiple testing correction method, there can be an increase in false positive findings (PMID: 30189032).

The EMP method we used to map eQTLs does not explicitly correct for the number of SNPs tested per gene, but rather applies a genome-wide multiple testing correction. We have now implemented a multiple testing correction method that employs a local gene level multiple testing correction, followed by a multiple testing correction over all genes (FastQTL approach). We have compared these strategies in the **Supplementary Note** and **Supplementary Figure 35**. In short, the novel approach provides slightly less stringent FDR cut-offs compared to our previous approach (see **Reviewer 1 Question 1** for more details) and yields additional significant eQTLs. We found that the new set of eQTLs replicate well in independent datasets (see **Reviewer 1 Question 2** for more details).

For cell type proportion ieQTLs we initially did not perform a permutation-based approach, but limited multiple testing to a Benjamini-Hochberg (BH-FDR) approach for three reasons. Firstly, because we limited the analysis to only testing the top associated SNP per significant eQTL gene, greatly reducing the multiple testing burden. Secondly, because permuting interaction terms is not trivial due to the correlation structure between the terms of an interaction model. Finally, Decon-QTL employs an NNLS model that constrains the beta estimates to be positive, a requirement that cannot always be met during permutation. We have now described these considerations in the **Supplementary Note**.

Keeping in mind the limitations of a permutation-based approach, we tested if such an approach was feasible. For this, we adjusted the Decon-QTL method to permute the genotype component of one interaction term and to determine the empirical null distribution per cell type using 1,000 permutations. As expected, the resulting null distribution was not uniformly distributed, but inflated near p-values of 1, probably due to the NNLS nature of the model. Nevertheless, the vast majority (87.8%) of the BH significant findings were also significant using a permuted based approach (**Supplementary Figure 38**). Moreover, we found that the BH significant cell type interaction eQTLs replicate well in independent datasets (see **Reviewer 1 Question 3** for more details).

Given the limitations of the permutation approach, we decided to maintain the BH-FDR as our significance threshold but also to report the permutation-based FDR statistics in **Supplementary Table 8**.

We have added a reference to this in the methods section as follows:

We evaluated if the Benjamini-Hochberg properly reflects the null distribution by applying a permutation-based method in conjunction and found that the vast majority (87.76%) of Benjamini-

Hochberg significant interactions were also significant using permutations (Supplementary Note; Supplementary Figures 37 and 38).

9. Using only one SNP (one instrument, one point) is in my opinion not sufficient to confidently call MR. Instead, allelic heterogeneity is needed where multiple independent loci are associated with both the gene expression and the downstream trait. Without this they are effectively fitting a line through the origin and one point. I would minimally request that the number of independent loci for each MR analysis be stated.

We agree that multi-SNP MR would be preferable to single SNP MR, but only providing enough SNPs in the cis region were available to fit such a regression line reliably. It was for this reason, due to the lack of independent eQTL instruments available within genes to pool across in the MetaBrain dataset, that we did not conduct multi-SNP MR analysis. 97% of the genes in MetaBrain (10,932 of 11,270 genes) could only be instrumented with a single cis-eQTL. Of the genes that could be instrumented with multiple eQTLs, 98% of these could only be instrumented with two eQTLs (2,276 of 2,331 genes). We did plan to conduct multi-SNP MR on the subset of genes which had enough eQTLs available, but none of them did (maximum number of instruments available within a gene was 5). However, it should be noted that the top MR findings were followed up with colocalization analysis which does increase confidence in the finding being genuine, albeit it does not eliminate the possibility of confounding of the MR estimate due to horizontal pleiotropy.

We added a summary of these numbers to the discussion:

[...] we focused our approach on a combination of Mendelian randomization and colocalization. However, these approaches currently have some limitations as well. For instance, we opted to perform single SNP MR because other approaches, such as inverse variance weighted⁸⁴ (IVW) MR, pool the estimates across many SNP instruments, which for many genes were not available. For example, less than 5% of the genes in MetaBrain could be instrumented with more than one independent eQTL, and no genes had more than 5 eQTLs available which would not provide for reliable IVW estimation.

10. Downstreamer: “genes within loci” seem to be implicated by proximity alone, not using the eQTL resource generated in this paper. This seems like a wasted opportunity to appropriately assign GWAS locus to gene.

The first step in the Downstreamer methodology is to calculate gene p-values by summarizing GWAS summary statistics within a 25kb window of each gene. To accommodate the suggestion of the reviewer, we have now incorporated eQTL information in this first step. To incorporate the eQTL information we used the --variantGene option in Downstreamer. Using the variantGene option it is possible to link genes to more distal variants based on eQTL information. For this comparison we used all primary and non-primary eQTLs found in MetaBrain Cortex-EUR with the aim to improve the gene p-values. The effect on the eventual key gene scores calculated by Downstreamer is minimal as the key-genes score when including the eQTL information are highly correlated to those when eQTL information is not included (Pearson $r \geq 0.95$; **Rebuttal Figure 5** and **Supplementary Figure 34**).

We have added the following to the results section:

*Adding primary and non-primary eQTL variants from Cortex-EUR to the Downstreamer analysis using the variantGene option did not markedly affect the key gene prediction scores (correlation in Z-scores ≥ 0.95 ; **Supplementary Figure 34**).*

Rebuttal Figure 5: Downstreamer key gene Z-scores when not including eQTL information (x-axis) and when including eQTL information (y-axis) for 6 different diseases.

11. I did not see a description in methods of concordance calculation (Figures 3C-E; assuming it is π_1 or is it sign of effect concordance)? It wasn't clear to me if Figure 3E, Supp Fig 11 comparing concordance with GTEx (across tissues including brain) excludes GTEx? Otherwise it could be overfit.

We apologize for having been unclear here. We realize that concordance and replication can mean different things to different researchers. We have changed the word 'concordance' to 'agreement', have added additional measures of agreement (Rb and π_1) and have clarified how we calculate agreement across tissues and ethnicities in our response to **Reviewer 1, Question 2**.

We confirm that the GTEx dataset was excluded when comparing the eQTL overlap between the GTEx tissue and our Cortex-EUR meta-analysis. This is clarified in the main text:

Next, we repeated Cortex-EUR eQTL discovery while excluding GTEx and compared the results with cis-eQTLs from different tissues in the GTEx project (Figure 3E; Supplementary Figure 14, Supplementary Table 6).

12. It would be interesting to see concordance/sharing from fetal brain to adult brain to assess temporal specificity of effects (PMIDs: 31626773, 32268104, 30419947)

We thank the reviewer for pointing us to these eQTL publications of fetal and developing brain. We have compared the top eQTLs of these datasets to MetaBrain eQTLs. All top eQTL SNP-gene combinations were significant in MetaBrain, and >94% share the same direction of effect (**Rebuttal Figure 6**). The genes that do not share the same direction of effect (*ATP2B4*, *MCM4*, *LOC105371397*, *ALAD*, *OPTN*, *GNB1L*, *DRAM2*, *NTPCR*, *ZNF665*, *SNORA3A*, *RFNG*, *LOC101930655*, *PM20D1*, *LINC00672*, *DDTL*, *LOC100130207*, *SLC35G2*, *NMB*, *TTC19*) are not shared across studies, except for *NTPCR*. ToppGene enrichment (<https://toppgene.cchmc.org/enrichment.jsp>) for these genes yields no results related to brain development.

Rebuttal Figure 6: Agreement of effects in fetal and developing brain eQTL datasets. Shown are top eQTL effects for Brien et al. (left, Genome Biol. 2018), Werling et al. (middle, Cell Rep, 2020), and Walker et al. (right, Cell, 2019), and matching SNP-gene combination in MetaBrain. All top eQTL SNP-gene combinations from these eQTL studies are significant in MetaBrain.

13. The discussion on lack of eQTL discovered at the MAPT/chr17 inversion region locus was interesting. I was also confused because chr17 does show an MR for intracranial volume rs199441 in the chr17 inversion region (Supp Fig 19). Can the authors discuss why the contig/multiple annotation issue only affects MAPT and not other genes within the region?

There are 29 protein coding genes within a 1MB window of *MAPT*. Of these, 23 are also located on scaffolding chromosomes. We compared counts per gene with and without patch chromosome within

GTEX samples. As shown in **Rebuttal Figure 7** the problem of lower or 0 counts when aligning to a genome build including patch chromosomes is not limited to only *MAPT* in that locus (although about half of the genes on patch chromosomes are unaffected).

Rebuttal Figure 7: Gene counts aligning against reference genome including (x-axis) and excluding (y-axis) patch chromosomes. Alignments were done using 1,457 GTEx samples. Red dots show *MAPT* and genes within 1MB window of *MAPT* which are present on one of the patch chromosomes.

Additionally, after our adjustments to the QTL method and removal of more PCs as detailed above, we now do identify a significant eQTL effect for *MAPT* (top SNP: rs2732600, $p=5.72 \times 10^{-26}$, Z-score 10.54). We have changed the sentence

We observed that significant cis-eQTL findings were sensitive to RNA-seq alignment strategies, and it is difficult to confidently ascertain cis-eQTLs in regions with multiple haplotypes represented on patch chromosomes, like the MAPT locus on 17q21

to

We observed that significant *cis*-eQTL findings were sensitive to RNA-seq alignment strategies, highlighting the importance of harmonizing the data prior to eQTL calling (**Supplementary Note, Supplementary Figures 9 and 10**).

but have left in our comments about *MAPT* in the supplemental text.

14. Was any kind of heterogeneity test applied to the meta-analysis results (from either *cis* or trans analyses)? In the forest plots in Supp Figure 25 most trans-eQTLs seems to be driven by ROS-MAP, which does seem strange in that CMC is almost of equivalent sample size yet not driving any of the results.

We have now tested each eQTL for heterogeneity using the I^2 measure and updated **Supplementary Tables 2 and 17** with this information. We observed that the majority (between 68% and 76%) of the *cis*-eQTLs had an $I^2 < 50\%$, which indicates low to moderate heterogeneity (**Rebuttal Figure 8 and Supplementary Figure 8E**).

Rebuttal Figure 8: Distribution of heterogeneity measure I^2 per *cis*-eQTL discovery dataset indicates that the majority of *cis*-eQTLs do not show a large amount of heterogeneity.

We have added the following sentence to the main text to reflect these results:

Cis-eQTL effect directions were highly concordant between datasets included in the Cortex-EUR meta-analysis (median Spearman $r=0.77$; median allelic concordance=83%; **Supplementary Figure 8A-D**), and the majority (between 68% and 76%) showed low to moderate heterogeneity ($I^2 < 50\%$ **Supplementary Table 2; Supplementary Figure 8E**), indicating robustness of the identified effects across datasets.

We have also calculated I^2 measures for *trans*-eQTLs, which show high heterogeneity specifically for 7p21q3 *trans*-eQTLs when not correcting for PCs, but low heterogeneity when these *trans*-eQTLs are excluded (**Rebuttal Figure 9**). A detailed discussion of the *trans*-eQTL results can be found as a response to **Reviewer 1 Question 4**.

Rebuttal Figure 9: Distribution of heterogeneity measure I^2 for *trans*-eQTLs indicates that the majority of heterogeneity in *trans*-eQTLs is attributed to chr7 *trans*-eQTLs that disappear when data is corrected for PCs or when the AMP-AD dataset is excluded from analysis.

15. For ENA samples, were the data checked against multiple samples from the same donor within ENA and across all the other cohorts participating by comparing genotypes to ensure donors were not double counted?

For ENA samples, heterozygous sites with strong allelic bias in expression may falsely call genotypes. How is this corrected? How are intergenic variants far away from genes called?

In this revision, we detected several potential duplicates in the Cortex-EUR and Cortex-AFR datasets, which were not limited to the ENA dataset. The numbers of detected duplicates are now listed in **Supplementary Table 1**. As a consequence, the sample size of the Cortex-EUR dataset decreased to 2,683 and for Cortex-AFR to 319.

We have added the following section to the methods:

Next, to detect potential duplicate genotype samples, for each of the eQTL discovery datasets, we merged the genotype datasets, filtered out variants with MAF < 10% and Hardy-Weinberg p-value < 0.001, and calculated a genetic pairwise relationship matrix after pruning the data using the `-indep-pairwise 1000 50 0.8` option in PLINK2.0¹⁰⁴. Using the values in the GRM, we selected individual pairs with a value > 0.2 as potential duplicates. From this set of pairs, we selected samples to keep using the following procedure: if one, but not both of the samples were from the ENA dataset, we opted to select

the non-ENA sample, and otherwise, we selected the sample that had the highest 'PCT_USABLE_BASES' from their matching RNA-seq sample. Using this procedure, we detected multiple potential genotype duplicates across multiple datasets in the Cortex-EUR and Cortex-AFR eQTL discovery datasets (**Supplementary Table 1**). This resulted in the following final eQTL datasets: Basal ganglia-EUR (n=208), Cerebellum-EUR (n=492), Cortex-EUR (n=2,683), Cortex-AFR (n=319), Hippocampus-EUR (n=208) and Spinal cord-EUR (n=108; **Supplementary Table 1, Figure 2C**).

We did not explicitly correct for allelic imbalance. We agree that this may lead to more false positives, but we observed strong agreement between the ENA cohort eQTLs and the meta-analysis, with the majority of the opposite effects being limited to smaller effect size eQTLs. We also investigated the average allelic balance per variant and observed that there were some heterozygous calls with low AB values. While we expect the ENA samples to have less confident genotype calls, we do not expect this to have large consequences for eQTL analysis, except for perhaps a loss of power for some eQTL effects in this dataset.

With regards to genotypes called far away from genes: we call any variant found in the RNA-seq data. We note that transcription of RNAs can also occur outside of genes, as has been recently demonstrated by Ye *et al.* (Essays Biochem., 2020). To assess the quality of genotype calls far away from genes, we have annotated the location of ENA variants before and after the imputation and compared the allele frequencies between variants from the ENA dataset and the other datasets. The majority of variants are found in exons and introns, although we do find that a large number of variants are found in intergenic regions. The allele frequencies between ENA and other datasets are highly comparable, as are the eQTLs that we identified, giving us further confidence that the majority of these genotype calls are correct.

We have added the following to the **Supplementary Note** regarding these issues:

*We note that we did not explicitly filter for genotype calls with aberrant allelic bias (AB), this can lead to false positive genotype calls. It has been shown that this may affect downstream analyses, for example when performing allele specific expression analysis⁵. Therefore, we investigated the distribution of average AB across all genotyped variants in the ENA dataset (**Supplementary Figure 3A**). We observed that homozygous reference and alternate calls generally had expected average AB values (around 0 for homozygous reference, and 1 for homozygous alternate). For heterozygote calls, we observed that 83% of variants had an average AB between 0.2 and 0.8. Furthermore, 1% of variants had an average AB value < 0.1 or > 0.9, indicating that for the majority of the heterozygous calls have AB values that fall within the expected range. As such, we note that indeed a number of variants may contain false-positive genotype calls, but this is likely true for the minority of included variants.*

We also note that we did not explicitly limit the genotype calling to exonic regions, because RNA-seq reads can also map to regions outside of genes, such as enhancer regions⁶. However, these calls can potentially be of poorer quality than those called in exonic regions due to lower coverage. We annotated variants using SNPEff⁷. Of the 101,499 variants in the ENA genotype call set before imputation, 13,212 (13.02%) are located in exons, 55,095 in introns (54.28%), 10,114 (9.96%) close to genes (5k up or

downstream from gene), 20,797 intergenic (20.49%) and 170 other (0.17%), e.g. 3' and 5' UTR). Of the 1,488,359 variants in the ENA genotype call set after imputation, 141,448 (10.18%) are located in exons, 919,978 in introns (61.68%), 174,783 (11.74%) close to genes (5k up or downstream from gene), 241,720 intergenic (16.24%) and 1,932 other (0.1%, e.g. 3' and 5' UTR). We compared allele frequency estimates of the called and imputed variants with a MAF > 1% between ENA and the AMP-AD datasets for EUR individuals and observed a correlation of 0.98 for variants in exonic regions and a correlation of 0.96 for variants located in intergenic regions, indicating that variant calls are highly correlated in terms of allele frequency regardless of their position to genes (**Supplementary Figure 3B**).

Regardless of these observations, we cannot exclude the possibility of false positive genotype calls in the ENA dataset. However, when comparing the eQTL Z-scores for ENA with those from the meta-analysis, we observed that 92% of the 8,610 eQTL genes had the same allelic direction of effect with the meta-analysis (**Supplementary Figure 3C**), suggesting that generally the lack of AB filtering or filtering of variants outside of exonic regions did not greatly affect the eQTL outcome.

Typos/small things:

We thank the reviewer for notifying use of these typos and small mistakes.

1. “Thus, there is a level of uncertainty for the expected proportion fr each cell type”

We have changed the text.

2. “The majority 700 of included samples was of EUR descent” -> “were”

We have changed the text.

3. GTEx appears twice in data availability section.

We have merged the two GTEx sections.

4. Supplementary Figure 1D missing

We have relabeled figure 1E-1G to 1D-1F.

5. Supplementary Figure 9, can you please label the expression bins from highest to lowest.

We have added in the figure that 1 is low bin and 10 is high bin.

6. Supp Figure 10 legend for part B doesn't seem to make sense. The y-axis is density not cerebellum expression and not sure what the dotted line represents.

We have changed the legend.

7. Supplementary Figure 12, would be useful to have correlation coefficients and p-values

We have added the Rb correlation coefficients and p-values.

8. In Figure 1 bottom left hand cartoon, I think it's more useful to give N in terms of number of individuals vs number of samples as this is most important for the eQTL analysis.

We have adjusted this figure to include eQTL dataset sample sizes.

9. Figure 2D looks overplotted to me, consider making dot sizes smaller so that points are not occluded.

We have adjusted this figure.

Updates to MR and colocalization analyses

Consequences of updated eQTL calling pipeline to AD colocalization results

Schwartzentruber *et al.* (Nat Genet., 2021) recently published a large GWAS of Alzheimer's disease and conducted colocalization analyses with previously published brain and microglia eQTL datasets including BrainSeq (Jaffe *et al.*, Nat Neurosci, 2018), ROSMAP (Ng *et al.*, Nat. Neurosci., 2017), xQTL-eQTL (Ng *et al.*, Nat Neurosci, 2017), AMP-AD and CMC (Sieberts *et al.*, Sci. Data, 2020). To compare MetaBrain Cortex-EUR eQTLs with these previously published brain eQTL datasets, we also ran a systematic colocalization between the significant AD GWAS loci with all nearby MetaBrain Cortex-EUR eQTLs. Details of this analysis were described in the **Supplementary Note**. In general, we replicated the majority of colocalization findings from Schwartzentruber *et al.* (Nat Genet., 2021) but also identified several genes with novel colocalization signals including *SLTM*, *ADAMTS4*, *TAS2R41*, *TAS2R60*, *INPP5D*, *MS4A4E*, *CLU*, *PRSS8*. Given that we have updated the pipeline for eQTL calling, we also assessed how the AD colocalization results changed from the previous pipeline. Two genes (*CASS4* and *ZNF646*) for which we replicated colocalization signal ($PP4 > 0.7$) from Schwartzentruber *et al.* (Nat. Genet. 2021) with eQTLs identified using the previous pipeline didn't show colocalization using the updated pipeline. Furthermore, several novel MetaBrain Cortex-EUR colocalization signals identified using previous pipeline also disappeared after pipeline updates including *KLHDC9*, *EPHA1*, *EED*, *TREML2*. However, a larger number of novel colocalization signals emerged with the new eQTL calling pipeline such as *SLTM* at the *ADAM10* locus ($PP4=0.76$), *ADAMTS4* at the *ADAMTS4* locus ($PP4=0.82$), *TAS2R41* ($PP4=0.79$) and *TAS2R60* ($PP4=0.98$) at the *EPHA1* locus, *INPP5D* at the *INPP5D* locus ($PP4=0.99$), *MS4A4E* at the *MS4A4A* locus ($PP4=0.75$), *CLU* at the *PTK2B-CLU* locus ($PP4=0.91$) and *PRSS8* at the *VKORC1* locus ($PP4=0.98$).

Other consequences of updated eQTL calling pipeline to MR and colocalization results

As we re-performed the eQTL analysis using the mbQTL method, we also needed to repeat the MR and colocalization step in the publication which used this data as an input. To examine consistency of the findings, we compared these results against our previous MR from the eQTL mapping pipeline (EMP). As

we anticipated, the MR estimates were very similar, with the major difference being additional genes picked up in the mbQTL MR which were not detected in the EMP MR.

For the MR, we LD clumped the eQTL summary stats and selected significant effects at $p < 5 \times 10^{-8}$ cut-off, harmonized the eQTL and GWAS SNP effects across our outcomes of interests, and then performed single SNP MR which derives a Wald ratio (WR) estimate for each eQTL-outcome SNP pairs available. We then selected out the top WR findings and performed Bayesian colocalization (coloc) analysis on these to confirm sharing of the genetic signal in the genomic region. In the EMP MR, a total of 268,030 WR tests were performed, of which 474 passed the Bonferroni correction of $p < 1.87 \times 10^{-7}$, and 159 passed coloc (coloc probability $< 70\%$). In the re-run with mbQTL eQTLs, a total of 359,763 WR tests were performed, of which 549 passed the Bonferroni correction of $p < 1.43 \times 10^{-7}$ and 186 passed coloc. For comparison purposes, a full join was performed on the top WR result (lowest p-value) for each gene in the mbQTL and EMP WR datasets to capture the WRs for both the shared genes and for the genes which were only instrumented in one of the datasets. This total combined dataset consisted of 310,074 WRs.

By using the mbQTL method it was possible to instrument more gene trait pairs. 72,232 more gene trait pairs (4,442 unique genes, 23% of the WRs) could be instrumented by mbQTL method and therefore had WR results available. In contrast, 7,697 gene trait pairs (1,449 unique genes, 2% of the WRs) were instrumented by the EMP but not by the mbQTL method. 1,147 WRs were identified as top hits (WR p-value $< 5 \times 10^{-5}$) in the EMP MR and 1,473 in the mbQTL MR. Gene trait pairs not instrumented in EMP accounted for 280 (19%) of the top WR findings in the mbQTL MR, which accounted for 57% of the 493 WR findings in total not detected in the EMP MR (**Rebuttal Figure 10**).

Rebuttal Figure 10: Number of different significant WR findings (p -value $< 5 \times 10^{-5}$) discovered by the mbQTL and EMP methods. Grey dots represent genes which were instrumented in the mbQTL method but not in EMP, and the blue dots are instrumented in the EMP and not the mbQTL. The orange and grey dots represent genes which are instrumented by both methods, but do not share significance (orange is significant in EMP but not mbQTL and yellow is significant in mbQTL but not EMP).

Next, we examined the agreement between the mbQTL and EMP MR results for the shared genes. Pearson's correlation (r) between WR estimates was high: max $r=0.92$, min $r=0.86$ across the 31 traits tested (**Rebuttal Figure 11**). The vast majority of WRs (91%) had the same directionality, with this proportion increasing for the top WR findings: 1,156 (97%) of the 1,193 top hit WRs for the mbQTL had the same direction as the EMP method. Furthermore, the vast majority of the WRs with opposite directionality between the methods are due to the GWAS SNP rather than the eQTL changing direction. In a follow-up analysis, where we were able to reliably orientate the SNP effects to be on the minor allele from 1000 Genomes, we found that of the 3,419 WRs with opposite directions, 3,361 (98%) had an opposite directionality for the GWAS SNP and not the eQTL effect. 150 of the 1,130 top hit WRs (13%) for EMP were not top hits in mbQTL MR and 213 of the 1,193 top hit WRs (18%) in mbQTL were not top

hits in EMP MR. Most of these WRs still had evidence of nominal significance (WR p-value < 0.05): 112 of the mbQTL WRs (75%) and 147 of the EMP WRs (69%).

Rebuttal Figure 11. Pearson correlation between WR estimates derived using the mbQTL and EMP method.

Decision Letter, first revision:

27th Jul 2022

Dear Dr Westra,

Your Article, "Brain expression quantitative trait locus and network analysis reveals downstream effects and putative drivers for brain-related diseases" has now been seen by 2 referees. You will see from their comments below that while they find your work of interest, some important points are raised. We are interested in the possibility of publishing your study in Nature Genetics, but would like to consider your response to these concerns in the form of a revised manuscript before we make a final decision on publication.

We therefore invite you to revise your manuscript taking into account all reviewer comments. Please highlight all changes in the manuscript text file. At this stage we will need you to upload a copy of the manuscript in MS Word .docx or similar editable format.

*2) If you have not done so already please begin to revise your manuscript so that it conforms to our Article format instructions, available

here.

*3) Include a revised version of any required Reporting Summary:

Please be aware of our guidelines on digital image standards.

[redacted]

We hope to receive your revised manuscript within about eight weeks. If you cannot send it within this time, please let us know.

Sincerely,
Wei

Wei Li, PhD
Senior Editor
Nature Genetics
New York, NY 10004, USA
www.nature.com/ng

Reviewers' Comments:

Reviewer #1:

Remarks to the Author:

The authors have address most of my concerns and substantially improved the manuscript. I have a few remaining questions that require some clarification:

1. eQTL agreement Rb: The authors used three approaches to investigate eQTL agreement between populations and tissues: allelic concordance, ρ_1 , and Rb. It wasn't clear in the results or in the methods section that if Rb is the correlation between eQTL beta (i.e., linear regression slope). Also in the methods section, the authors mentioned that eQTL agreement analysis "taking into account the effect direction, size and standard error in both datasets". How these are factors addressed by the three approaches should be explained. I suggest in addition to the correlation between betas, the authors calculate allelic fold change (aFC) as a measure of eQTL effect size and compare with different tissues from GTEx.
2. eQTL agreement ρ_1 : In the methods, I suggest that the authors explain for discovery gene-eQTL pairs that are absent in the replication dataset, are these omitted, or is any distribution used to

generate missing p-values.

3. Terms: Are these terms different or exchangeable: in lines 223-224, "cell type dependent eQTLs can be identified in bulk RNA-seq data by performing cell type deconvolution and determining cell type interaction eQTLs (ieQTLs)", and in methods section (line 991), "cell type mediated eQTLs".

4. MR and colocalization: I suggest the authors explain the motivation of restricting colocalization to MR genes in results or in methods section.

5. Trans-eQTL: Motivation of restricting trans-eQTL discovery to disease associated variants and lead cis-eQTL variants. The authors argued that identified trans-eQTLs are enriched in lead cis-eQTL variants, but isn't the analysis limited to those variants in the first place?

Minor:

6. Fig 3B "Genes without significant eQTL" in non-primary eQTL ranks. Are these the same set of genes as shown in primary group?

7. Line 258 median of agreement. I suggest report the median of Rb and pi1 in addition to the range.

In the abstract, I believe macrophage actually refers to microglia now

Reviewer #2:

Remarks to the Author:

The authors have addressed all my concerns in detail. I think this is an exciting resource that will be highly valuable to the neurogenetics community and has interesting, well-powered findings. I support the publication of this manuscript.

Author Rebuttal, first revision:

Response to referees

Reviewer #1:

The authors have address most of my concerns and substantially improved the manuscript. I have a few remaining questions that require some clarification:

We thank the reviewer for their careful consideration of our revised manuscript.

1. eQTL agreement Rb: The authors used three approaches to investigate eQTL agreement between populations and tissues: allelic concordance, pi1, and Rb. It wasn't clear in the results or in the methods section that if Rb is the correlation between eQTL beta (i.e., linear regression slope). Also in the methods section, the authors mentioned that eQTL agreement analysis "taking into account the effect direction, size and standard error in both datasets". How these are factors addressed by the three approaches should be explained. I suggest in addition to the correlation between betas, the authors calculate allelic fold change (aFC) as a measure of eQTL effect size and compare with different

tissues from GTEx.

We thank the reviewer for this suggestion. In principle, correlation of allelic fold changes (abbreviated to caFC) and R_b should both capture correlation of effect sizes, but both methods have slightly different assumptions, and while allelic fold change (aFC) requires genotype and expression data to be available, R_b can be calculated using summary statistics only. As the reviewer suggests, R_b captures the correlations of eQTL effect size slopes (e.g.: beta's), while taking into account potential covariance in the standard errors of those slopes (see eq. 1-4 of Qi et al, Nature Communications, 2018). R_b calculations have been performed for this purpose on GTEx brain tissues before by Qi *et al.*, Nature Communications 2018 and more recently by us to compare blood based eQTLgen eQTLs with the various GTEx tissues (Vösa *et al.*, Nature Genetics 2021). The aFC method estimates the unit difference in expression between alleles, which when correlated, should be very comparable with correlations of eQTL effect size betas if the expression data is properly normalized. We observed a high correlation between R_b and caFC estimates for comparisons within MetaBrain and between MetaBrain Cortex-EUR and GTEx datasets.

We have added the following to the main text and have updated **Supplementary Table 5** and **Supplementary Figure 12** to include the aFC results:

We used four metrics for eQTL agreement. We calculated allelic concordance (AC) as the proportion of shared significant eQTLs that have the same direction of effect in both discovery and replication dataset, π_1 as the proportion of eQTLs that are true positives in the replication dataset and the R_b and correlation of allelic fold change (caFC) as a measure of the correlation of effect sizes between two datasets (see Methods).

[...]

We observed high allelic concordance, R_b and caFC estimates between the different ethnicities ($R_b > 0.78$, Figure 3C; caFC > 0.55, AC > 92.95%, Supplementary Figure 12; Supplementary Table 5).

Interestingly, π_1 and caFC estimates between ethnicities were low when the discovery set had more samples than the replication (e.g.: Cortex-EUR vs Cortex-EAS caFC: 0.55, π_1 : 0.29, Supplementary Figure 12, but not vice versa (e.g.: Cortex-EAS vs Cortex-EUR caFC: 0.85, π_1 : 0.95, Supplementary Figure 12).

[...]

Next, we compared different brain regions and observed high AC, R_b and caFC estimates between different brain regions overall ($R_b > 0.76$, Figure 3D; caFC > 0.65, AC > 91%, Supplementary Figure 12)

We have now also clarified the distinctions between the methods in the methods text:

We have used four different measurements of agreement of eQTL effects when comparing between different brain regions or tissues: allelic concordance (AC), π_1 , R_b , and correlation of allelic fold changes (caFC).

[...]

R_b ¹¹³ effectively estimates the correlation between eQTL effect slopes (e.g.: betas from linear regression), while controlling for potential covariance in standard errors of those slopes, and caFC measures the correlation between estimates of the foldchange in expression values between alleles. We note that while AC, π_1 , and R_b can be calculated from summary statistics, caFC requires access to genotype and expression data. We therefore limited the caFC analysis to comparisons within the MetaBrain datasets, and to comparisons with the GTEx tissues.

[...]

As an additional analysis, we performed R_b analysis while focusing on eQTL SNPs identified in the GTEx muscle tissue, to prevent potential biases caused by detection of eQTL SNPs in the same type of tissue, as described in the original R_b manuscript. For AC, π_1 and R_b comparisons with GTEx, we used the summary statistics for GTEx-v8 that were downloaded from the GTEx portal website. Finally, we applied a similar procedure to calculate caFC, by taking the SNP-gene combination with the lowest p-value in the discovery cohort for each gene if it was significant, matching those to the replication dataset, regardless of significance in the replication dataset. In MetaBrain datasets, we calculated allelic fold changes (aFC) using gene expression data that was log2 transformed and corrected for technical covariates and the suitable number of principal components for that dataset. While for GTEx, we downloaded the GTEx-v8 normalized expression matrices and covariate matrices for each tissue from the GTEx portal website, after which we log2-transformed and corrected the gene expression matrix for each tissue using the covariates by OLS. aFC calculations per eQTL were performed with the original aFC script by Mohammadi et al. using the settings `--log_xform 1` and `--log_base 2`. For all comparisons with GTEx tissues, we performed discovery in Cortex-EUR, while excluding the GTEx cohort.

2. eQTL agreement pi1: In the methods, I suggest that the authors explain for discovery gene-eQTL pairs that are absent in the replication dataset, are these omitted, or is any distribution used to generate missing p-values.

We have added the following sentence to the pi1 calculation explanation to make clear we do not impute missing p-values:

SNP-gene combinations not tested in the replication cohort were removed.

3. Terms: Are these terms different or exchangeable: in lines 223-224, “cell type dependent eQTLs can be identified in bulk RNA-seq data by performing cell type deconvolution and determining cell type interaction eQTLs (ieQTLs)”, and in methods section (line 991), “cell type mediated eQTLs”.

We agree with the reviewer that it is important for clarity to harmonize the terms. We have updated “cell type mediated eQTLs” to “cell type dependent eQTLs”, and “eQTLs mediated by cell type” to “eQTLs dependent on cell type” throughout the manuscript.

4. MR and colocalization: I suggest the authors explain the motivation of restricting colocalization to MR genes in results or in methods section.

We thank the reviewer for his suggestion. We have added the following sentence to the colocalization methods section:

Colocalization analysis was restricted to the top MR findings as it is not necessary to colocalize genes not identified by MR. For such genomic regions there will only be evidence that they are associated with gene expression and not also with the disease outcome of interest.

5. Trans-eQTL: Motivation of restricting trans-eQTL discovery to disease associated variants and lead cis-eQTL variants. The authors argued that identified trans-eQTLs are enriched in lead cis-eQTL variants, but isn't the analysis limited to those variants in the first place?

We apologize for the confusion here. The analysis the reviewer mentions was performed within the set of SNPs tested for *trans*-eQTLs that also overlapped *cis*-eQTLs SNPs in Cortex-EUR. We have clarified this enrichment test as follows in the main text:

We therefore concentrated on 737 trans-eQTLs, detected after correcting for 100 PCs, which reflect 526 unique SNPs, and 108 unique genes. 127 SNPs had trans-eQTL effects on multiple genes. 461 (88%) of the trans-eQTL SNPs were associated with a significant cis-eQTL in Cortex-EUR. However, this overlap also included SNPs that were not necessarily the cis-eQTL index SNPs (i.e.: those with the strongest association with a gene), but could rather be variants tagging the index SNP because of LD. Within the set of SNPs we tested for trans-eQTLs that also overlapped with cis-eQTL SNPs in Cortex-EUR, we therefore determined how many SNPs were a cis-eQTL index SNP. 150 (33%) of the 461 significant trans-eQTL SNPs overlapping a cis-eQTL SNP were also the cis-eQTL index SNP, whereas of the 109,172 SNPs

that did overlap a cis-eQTL SNP but that were not a trans-eQTL SNP, only 15,745 (14.42%) were index SNPs, which represents an enrichment (Fisher exact test p -value= 1.2×10^{-28}).

Minor:

6. Fig 3B “Genes without significant eQTL” in non-primary eQTL ranks. Are these the same set of genes as shown in primary group?

No, but this was not clearly stated in the figure legend. To clarify this, we have added the following line to figure 3B description:

For mean expression and pLI score, each row compares the eQTL genes for that rank with eQTL genes from the previous rank (e.g. for tertiary eQTLs, the non-significant (grey) distribution is from genes that do have secondary, but don't have tertiary eQTLs).

7. Line 258 median of agreement. I suggest report the median of Rb and pi1 in addition to the range.

We thank the reviewer for the suggestion. We have now added median R_b , π_1 and AC values to the main text:

In the Cortex-AFR dataset we observed moderate agreement ($-0.17 < R_b < 0.98$, median R_b : 0.77, $0.00 < \pi_1 < 0.06$, median π_1 : 0.0, $43\% < AC < 72\%$, median AC: 0.67; Supplementary Table 9, Supplementary Figure 17B)

[...]

We observed good agreement for most cell types ($0.13 < R_b < 0.82$, median R_b : 0.68, $0 < \pi_1 < 0.63$, median π_1 : 0.36, $48\% < AC < 81\%$, median AC: 0.65; Supplementary Table 9, Supplementary Figure 18B).

[...]

For the overlapping eQTLs we observed a high agreement for all cell types when excluding inhibitory neurons ($0.78 < R_b < 0.86$, median R_b : 0.84, $0.43 < \pi_1 < 0.83$, median π_1 : 0.69, $81\% < AC < 90\%$, median AC: 0.9; Supplementary Table 9, Supplementary Figure 19B).

8. In the abstract, I believe macrophage actually refers to microglia now

We thank the reviewer for pointing this out, this has been updated to microglia.

Reviewer #2:

The authors have addressed all my concerns in detail. I think this is an exciting resource that will be highly valuable to the neurogenetics community and has interesting, well-powered findings. I support the publication of this manuscript.

We thank the reviewer for their careful consideration of our revised manuscript.

References:

Qi, T., Wu, Y., Zeng, J. *et al.* Identifying gene targets for brain-related traits using transcriptomic and methylomic data from blood. *Nat Commun* **9**, 2282 (2018). <https://doi.org/10.1038/s41467-018-04558-1>

Decision Letter, second revision:

30th Sep 2022

Dear Dr. Westra,

Thank you for submitting your revised manuscript "Brain expression quantitative trait locus and network analysis reveals downstream effects and putative drivers for brain-related diseases" (NG-A57137R1). It has now been seen by the original referees and their comments are below. The reviewers find that the paper has improved in revision, and therefore we'll be happy in principle to publish it in Nature Genetics, pending minor revisions to comply with our editorial and formatting guidelines.

Sincerely,
Wei

Wei Li, PhD
Senior Editor
Nature Genetics
New York, NY 10004, USA
www.nature.com/ng

Reviewer #1 (Remarks to the Author):

The authors have addressed all my concerns.

Final Decision Letter:

17th Jan 2023

Dear Dr. Westra,

I am delighted to say that your manuscript "Brain expression quantitative trait locus and network analysis reveals downstream effects and putative drivers for brain-related diseases" has been accepted for publication in an upcoming issue of Nature Genetics.

Your paper will be published online after we receive your corrections and will appear in print in the next available issue. You can find out your date of online publication by contacting the Nature Press Office (press@nature.com) after sending your e-proof corrections. Now is the time to inform your Public Relations or Press Office about your paper, as they might be interested in promoting its publication. This will allow them time to prepare an accurate and satisfactory press release. Include your manuscript tracking number (NG-A57137R2) and the name of the journal, which they will need when they contact our Press Office.

Please note that *Nature Genetics* is a Transformative Journal (TJ). Authors may publish their research with us through the traditional subscription access route or make their paper immediately open access through payment of an article-processing charge (APC). Authors will not be required to make a final decision about access to their article until it has been accepted. [Find out more](https://www.springernature.com/gp/open-research/transformative-journals)

about Transformative Journals

Authors may need to take specific actions to achieve [compliance](https://www.springernature.com/gp/open-research/funding/policy-compliance-faqs) with funder and institutional open access mandates. If your research is supported by a funder that requires immediate open access (e.g. according to [Plan S principles](https://www.springernature.com/gp/open-research/plan-s-compliance)) then you should select the gold OA route, and we will direct you to the compliant route where possible. For authors selecting the subscription publication route, the journal's standard licensing terms will need to be accepted, including <https://www.nature.com/nature-portfolio/editorial-policies/self-archiving-and-license-to-publish>. Those licensing terms will supersede any other terms that the author or any third party may assert apply to any version of the manuscript.

Please note that Nature Portfolio offers an immediate open access option only for papers that were first submitted after 1 January, 2021.

If you have not already done so, we invite you to upload the step-by-step protocols used in this manuscript to the Protocols Exchange, part of our on-line web resource, natureprotocols.com. If you complete the upload by the time you receive your manuscript proofs, we can insert links in your article that lead directly to the protocol details. Your protocol will be made freely available upon publication of your paper. By participating in natureprotocols.com, you are enabling researchers to more readily reproduce or adapt the methodology you use. [Natureprotocols.com](https://natureprotocols.com) is fully searchable, providing your protocols and paper with increased utility and visibility. Please submit your protocol to <https://protocolexchange.researchsquare.com/>. After entering your nature.com username and

password you will need to enter your manuscript number (NG-A57137R2). Further information can be found at <https://www.nature.com/nature-portfolio/editorial-policies/reporting-standards#protocols>

Sincerely,
Wei

Wei Li, PhD
Senior Editor
Nature Genetics
New York, NY 10004, USA
www.nature.com/ng